# In-Context Learning Strategies Emerge Rationally

**Daniel Wurgaft**[1,2,3*]  **Ekdeep Singh Lubana**[2,3*]  **Core Francisco Park**[2]
**Hidenori Tanaka**[2,3]  **Gautam Reddy**[4]  **Noah D. Goodman**[1,5]

[1]Department of Psychology, Stanford University
[2]CBS-NTT Program in Physics of Intelligence, Harvard University
[3]Physics of Artificial Intelligence Group, NTT Research, Inc., Sunnyvale, CA, USA
[4]Joseph Henry Laboratories of Physics, Princeton University
[5]Department of Computer Science, Stanford University

## Abstract

Recent work analyzing in-context learning (ICL) has identified a broad set of strategies that describe model behavior in different experimental conditions. We aim to unify these findings by asking *why* a model learns these disparate strategies in the first place. Specifically, we start with the observation that when trained to learn a mixture of tasks, as is popular in the literature, the strategies learned by a model for performing ICL can be captured by a family of Bayesian predictors: a memorizing predictor, which assumes a discrete prior on the set of seen tasks, and a generalizing predictor, where the prior matches the underlying task distribution. Adopting the normative lens of *rational analysis*, where a learner's behavior is explained as an *optimal* adaptation to data given *computational constraints*, we develop a hierarchical Bayesian framework that *almost perfectly* predicts Transformer next-token predictions throughout training—*without* assuming access to its weights. Under this framework, pretraining is viewed as a process of updating the *posterior probability* of different strategies, and inference-time behavior as a *posterior-weighted average* over these strategies' predictions. Our framework draws on common assumptions about neural network learning dynamics, which make explicit a tradeoff between loss and complexity among candidate strategies: beyond how well it explains the data, a model's preference towards implementing a strategy is dictated by its complexity. This helps explain well-known ICL phenomena, while offering novel predictions: e.g., we show a superlinear trend in the timescale for transitioning from generalization to memorization as task diversity increases. Overall, our work advances an explanatory and predictive account of ICL grounded in tradeoffs between strategy loss and complexity.

## 1 Introduction

In-Context Learning (ICL) has significantly expanded the open-ended nature of large language models (LLMs) [1–5], allowing them to learn novel behaviors from merely the provided context [6–12]. This has motivated a large body of work that analyzes controlled experimental settings to better understand ICL [13–17], leading to (i) behavioral accounts of *what* strategies are followed by a model to learn from its context [18–23], e.g., the ridge estimator in linear regression tasks [14, 17]; (ii) developmental accounts identifying *when*, i.e., under what training [24] and data [25] conditions, a particular strategy is used by the model [24–30]; or (iii) mechanistic accounts characterizing *how* such strategies get implemented [31–33], e.g., the use of induction heads [34]. While some have attempted characterizing ICL as a single procedure [22, 35, 36], more recent work has argued that the broader phenomenology of ICL stems from a model learning different strategies under varying experimental conditions [26, 30, 37].

---

[*]Equal contribution. Email: `wurgaft@stanford.edu`, `ekdeeplubana@fas.harvard.edu`.

39th Conference on Neural Information Processing Systems (NeurIPS 2025).

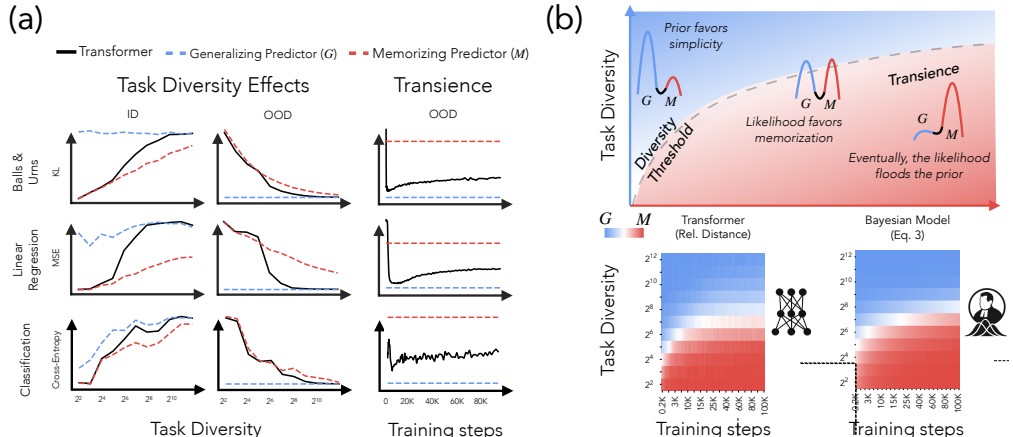

Figure 1: **Why Does a Model Learn Different Strategies for Performing ICL?** To answer this question, we analyze three distinct settings where a model is trained to learn a mixture of tasks. **(a) Model Behavior Transitions Between Memorizing and Generalizing Predictors.** We first make the observation that across settings, as diversity of data distribution and amount of training are increased, model behavior transitions between two Bayesian predictors: a memorizing predictor, $M$, which assumes a discrete prior over seen tasks, and a generalizing predictor, $G$, which assumes a continuous prior over the true distribution. This recapitulates prior results on task-diversity thresholds [25] and transient generalization [24] in a unifying language. **(b) A Hierarchical Bayesian Framework Provides an Explanatory and Predictive Account of ICL.** The consistency of these transitions motivates a hierarchical Bayesian model of ICL, where a model's inference-time behavior is framed as a posterior-weighted interpolation in the Bayesian predictors $M$ and $G$, while pretraining is seen as a process of updating the preference (posterior probability) toward different predictors. We find that under reasonable assumptions regarding neural network learning dynamics, our framework is highly predictive of model behavior *throughout* training, *without* access to model weights (bottom panel). Additionally, our framework provides an explanatory account of ICL phenomena via a tradeoff between the *loss* and the *complexity* of learned solutions (top panel).

These results suggest that a remaining hurdle for developing a unified account of ICL is understanding *why*, in the first place, models learn different strategies with disparate generalization abilities. Indeed, given that memorizing the training distribution almost always leads to better performance, why does the model learn an 'underfitting', out-of-distribution generalizing solution at all [24, 25]? Moreover, if capacity limitations prevent memorization, why does the model, among all underfitting solutions, learn the one that captures the *true* generative process [25, 26]? Finally, why does it *first* learn such a solution, only to eventually give way to one that does not generalize to novel tasks [24, 27]? In addressing the questions above, we make the following **contributions**.

- **Model Behavior Transitions Between Memorizing and Generalizing Predictors.** We first make the observation that in popularly studied ICL settings, where a model is trained to learn a mixture of tasks, previously identified setting-specific solutions can be unified in the language of Bayesian inference. Across *three distinct settings*, we replicate well-known ICL phenomena [24, 25] and show models primarily transition between two ICL phases, determined by behaviorally matching one of two Bayesian predictors: 1) a *memorizing predictor* with a discrete prior over *seen* tasks or 2) a *generalizing predictor* with a continuous prior over the *true* data-generating distribution.

- **A Hierarchical Bayesian Model of ICL Grounded in Rational Analysis.** This observation then motivates our core contribution: we propose to understand ICL by invoking the approach of **rational analysis** from cognitive science [38–42], where a learner's behavior is explained as an *optimal* adaptation to data, given a learning objective and *computational constraints*. In our case, building on the finding that Transformers transition between memorizing and generalizing predictors, we examine how a Bayes-optimal learner would trade off between these solutions across training and data conditions, given a *simplicity* bias [43–48] and power-law scaling of loss with dataset size [49, 50]. This yields a *hierarchical* Bayesian framework that casts pretraining as a process of weighing preference (*posterior probability*) toward different solutions based on their *loss* and *complexity*. At inference, model behavior is framed as a posterior-weighted average of these solutions (which themselves are Bayesian—hence the term "hierarchical"). This is in contrast

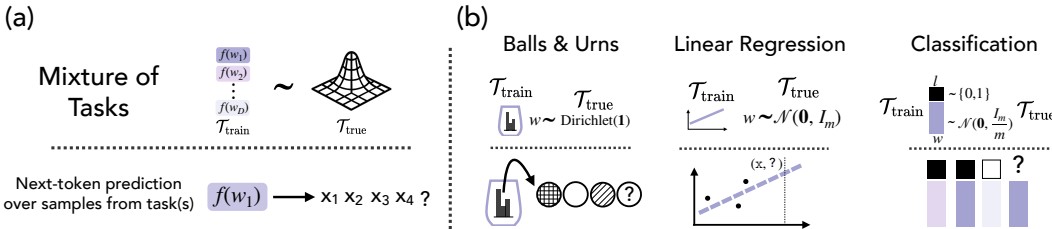

Figure 2: **Experimental Settings: Learning a Finite Mixture of Tasks. (a) General Formulation.**
Popularly studied experimental settings in the literature on ICL can be seen as training a model to
learn a distribution defined using a mixture of tasks (denoted $\mathcal{T}_{\text{train}}$), where each task is a parameterized
latent function whose parameters are sampled from a distribution $\mathcal{T}_{\text{true}}$. **(b) Considered Settings.** We
analyze three distinct instantiations of this general formulation: *Balls & Urns*, which captures the
belief update interpretation of ICL and is a simplification of the Markov modeling setting from prior
work [26, 52], and two popularly studied settings from the literature that capture the few-shot learning
interpretation of ICL, i.e., in-context *linear regression* [14, 25] and *Classification* [20, 24, 28, 29].

to several previous Bayesian accounts of ICL [35, 51], which frame ICL as implementing a single
Bayesian predictor. We derive a closed-form expression for our model, which *almost perfectly*
predicts Transformer next-token predictions throughout training *without* access to weights, as well
as captures varied ICL phenomena including task diversity effects and transience.

- **A Loss-Complexity Tradeoff Underlies ICL Phenomena and Yields Novel Predictions.** Our
framework makes explicit a tradeoff driving model preference towards a predictor as a function of
its *loss* and *complexity*. This tradeoff helps explain *transitions in model behavior* due to changes in
data-diversity [25] or training time [24] as artifacts *implicitly* induced by changes in a predictor's
complexity or loss, *relative* to other predictors. This interpretation further helps us identify several
novel predictions, e.g., a superlinear trend between task diversity and transience.

Overall, we argue our work advances a unifying explanatory and predictive account of ICL. More
broadly, our results suggest the value of taking a *normative* perspective towards explaining neural
networks, viewing learning behavior (both in-context and throughout training) as a *rational* adaptation
to data properties and computational constraints.

## 2 Preliminaries: Learning a Finite Mixture of Tasks

To capture both few-shot learning [6, 14, 20, 25, 53] and belief update formulations [26, 34, 54, 55]
of ICL, we analyze three distinct experimental settings in this paper. To this end, we first provide
a general formulation of the learning setup, which allows us to formalize a unified language for
examining model strategies for ICL in the next section.

**General Formulation.**  We cast prior settings used for studying ICL [20, 25, 26] as learning a finite
mixture of tasks. Specifically, settings analyzed in this work involve learning a mixture distribution
$\mathcal{T}_{\text{train}}$ defined over $D$ parametrized functions $\{f(\boldsymbol{w}_1, \cdot), \ldots, f(\boldsymbol{w}_D, \cdot)\}$, or 'tasks'. For each function
$f(\boldsymbol{w})$, the parameters $\boldsymbol{w} \in \mathbb{R}^m$ are sampled from a predefined distribution $\mathcal{T}_{\text{true}}$ (see Fig. 2). We
call $D$ the *task diversity* of the mixture. Every training iteration, we randomly select a function
$f(\boldsymbol{w}, \cdot) \in \mathcal{T}_{\text{train}}$, and use it to generate a sequence of length $C$ (details vary by setting, see below).
Batches consist of independently generated sequences. We autoregressively train Transformers
(GPT-NeoX architecture [56, 57]) for a predefined number of iterations $N$. Model performance is
evaluated either on in-distribution (ID) sequences drawn from $\mathcal{T}_{\text{train}}$, or on out-of-distribution (OOD)
sequences drawn from the underlying distribution $\mathcal{T}_{\text{true}}$ (see App. F for further details on architecture
and method). We analyze three specific instantiations of this general formulation, detailed next.

**Balls & Urns.** Related to the belief update formulation of ICL, this setting is inspired by the classic
'Urn Problem' from the probability literature [58]. Specifically, one draws (with replacement) balls
from an urn containing balls of $m$ types, and the goal is to estimate the distribution of ball types. Since
solving this task only requires inferring unigram statistics of the input (a histogram), this setting simpli-
fies the Markov modeling setup proposed by Park et al. [26]. A task $f(\boldsymbol{w}) = \text{Categorical}(\boldsymbol{w})$ denotes
a stochastic map ('urn') from which states ('balls') are sampled, with $\boldsymbol{w} \sim \mathcal{T}_{\text{true}} = \text{Dirichlet}(\mathbf{1})$.
Thus, the distribution $\mathcal{T}_{\text{train}}$ consists of $D$ 'urns' $\{\text{Categorical}(\boldsymbol{w}_1), \ldots, \text{Categorical}(\boldsymbol{w}_D)\}$. To
generate data, we sample $C$ states from a randomly selected function $f(\boldsymbol{w})$, yielding a sequence
$\boldsymbol{s} := [\boldsymbol{x}_1 \ldots \boldsymbol{x}_C]$, where $\boldsymbol{x} \sim \text{Categorical}(\boldsymbol{w})$.

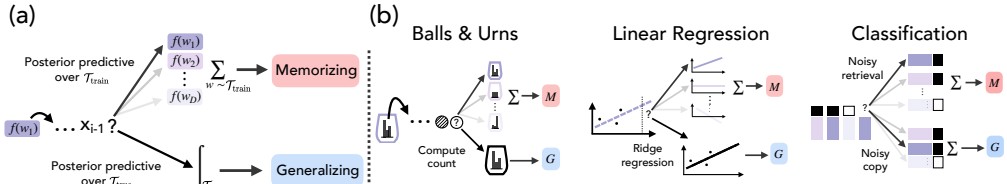

Figure 3: **Predictors in Different Experimental Settings. (a) Memorizing and Generalizing Predictors.** We compare model behavior to two idealized Bayesian predictors: (i) *Memorizing predictor* ($M$), which assumes a discrete prior over the mixture distribution $\mathcal{T}_{\text{train}}$, and (ii) *Generalizing predictor* ($G$), which assumes a prior over $\mathcal{T}_{\text{true}}$, the distribution from which tasks are sampled. **(b) Task-Specific Instantiations.** These predictors yield closed-form solutions (App. G); e.g., in Balls & Urns, the memorizing predictor computes a posterior-weighted average over urns seen in training, whereas the generalizing predictor uses empirical unigram statistics with pseudo-counts.

**Linear Regression.** A standard problem setting in literature on understanding the few-shot learning formulation of ICL [14, 25, 53]. Here, the goal is to learn to in-context solve linear regression problems. A task $f(\boldsymbol{w}) = \boldsymbol{w}^\intercal \boldsymbol{x} + \epsilon$ denotes a noisy linear map that transforms a continuous input via a linear map $\boldsymbol{x} \sim \mathcal{N}(\boldsymbol{0}, \boldsymbol{I}_m)$ and introduces additive noise $\epsilon \sim \mathcal{N}(0, \sigma^2)$, where $\boldsymbol{w} \sim \mathcal{T}_{\text{true}} = \mathcal{N}(\boldsymbol{0}, \boldsymbol{I}_m)$ and $\boldsymbol{I}_m$ denotes the $m \times m$ identity matrix. Thus, the training distribution $\mathcal{T}_{\text{train}}$ consists of $D$ linear mappings $\{f(\boldsymbol{w}_1, \cdot), \ldots, f(\boldsymbol{w}_D, \cdot)\}$. To generate data, we sample $C$ inputs and transform them with a randomly selected function $f(\boldsymbol{w}, \boldsymbol{x})$, yielding a sequence $\boldsymbol{s} := [\boldsymbol{x}_1, \boldsymbol{w}^\intercal \boldsymbol{x}_1 + \epsilon_1, \ldots, \boldsymbol{x}_C, \boldsymbol{w}^\intercal \boldsymbol{x}_C + \epsilon_C]$.

**Binary Classification.** Another popularly studied setting in literature on understanding the few-shot formulation of ICL [20, 28]. Our parameterization is inspired by Nguyen and Reddy [29], who define a task $f(\boldsymbol{w}, l) = \boldsymbol{w} \oplus l$ to denote an item-label pair, with $\boldsymbol{w} \sim \mathcal{N}(0, \boldsymbol{I}_m/m), l \in \{0, 1\}$ defining $\mathcal{T}_{\text{true}}$. Thus, the training task distribution $\mathcal{T}_{\text{train}}$ consists of $D$ item-label pairs $\{\boldsymbol{w}_1 \oplus l_1, \ldots, \boldsymbol{w}_D \oplus l_D\}$. Unlike other settings, multiple functions $f(\boldsymbol{w}, l)$ are used to generate data: first, $C - 1$ item-label pairs $\boldsymbol{w} \oplus l$ are randomly sampled (with replacement) from $\mathcal{T}_{\text{train}}$. A pair is chosen from the sequence at random to be the query pair—it is appended to the end of the sequence, and its label is corrupted to be $-1$. We noise these items via $\tilde{\boldsymbol{w}} = \frac{\boldsymbol{w} + \sigma \boldsymbol{\epsilon}}{\sqrt{1 + \sigma^2}}$, with $\sigma \in \mathbb{R}$ acting as the within-class variance, and $\boldsymbol{\epsilon} \sim \mathcal{N}(0, \boldsymbol{I}_m/m)$. This process yields a sequence $\boldsymbol{s} := [\boldsymbol{x}_1, \ldots, \boldsymbol{x}_{C-1}, \boldsymbol{x}_{\text{query}}] = [\tilde{\boldsymbol{w}}_1 \oplus l_1, \ldots, \tilde{\boldsymbol{w}}_{C-1} \oplus l_{C-1}, \tilde{\boldsymbol{w}}_{\text{query}} \oplus -1]$. Models are only trained to predict the label of $\tilde{\boldsymbol{w}}_{\text{query}}$.

## 3  What Strategies: Memorizing and Generalizing Predictors

Our goal in this work is to understand *why* a model learns different strategies for performing ICL. We must thus first establish *what* these strategies are, allowing us to then characterize the dynamics driving changes in a model's preferred strategy. To this end, we build on the idea that autoregressive training with the next-token prediction objective corresponds to maximizing the likelihood of the data and learning the distribution underlying it [59]. We thus consider the two distributions forming the basis of our general formulation—i.e., $\mathcal{T}_{\text{train}}$ and $\mathcal{T}_{\text{true}}$—and consider *optimal* strategies a learner can be expected to implement if it learns these distributions. This generalizes the approach of Raventós et al. [25], and yields the following two Bayesian predictors.

- **Memorizing Predictor ($M$).** The memorizing predictor assumes a discrete prior over the distribution of seen tasks ($\mathcal{T}_{\text{train}}$) and implements a posterior predictive of the form:

$$M(\boldsymbol{s}_i|\boldsymbol{s}_{1:i-1}) = \sum_{\boldsymbol{w} \sim \mathcal{T}_{\text{train}}} p(\boldsymbol{w}|\boldsymbol{s}_{1:i-1}) f_{\boldsymbol{w}}(\boldsymbol{s}_i|\boldsymbol{s}_{1:i-1}) = \mathbb{E}_{\boldsymbol{w}|\boldsymbol{s}_{1:i-1}, \mathcal{T}_{\text{train}}}[f_{\boldsymbol{w}}(\boldsymbol{s}_i|\boldsymbol{s}_{1:i-1})].$$

- **Generalizing Predictor ($G$).** The generalizing predictor assumes a continuous prior over the distribution from which tasks are sampled ($\mathcal{T}_{\text{true}}$), implementing a posterior predictive of the form:

$$G(\boldsymbol{s}_i|\boldsymbol{s}_{1:i-1}) = \int_{\boldsymbol{w} \sim \mathcal{T}_{\text{true}}} p(\boldsymbol{w}|\boldsymbol{s}_{1:i-1}) f_{\boldsymbol{w}}(\boldsymbol{s}_i|\boldsymbol{s}_{1:i-1}) \, d\boldsymbol{w} = \mathbb{E}_{\boldsymbol{w}|\boldsymbol{s}_{1:i-1}, \mathcal{T}_{\text{true}}}[f_{\boldsymbol{w}}(\boldsymbol{s}_i|\boldsymbol{s}_{1:i-1})].$$

For all tasks we analyze, the predictors above can be defined in a closed-form manner (see App. G), mapping onto task-specific strategies defined in prior work: e.g., what has been called 'dMMSE' vs. ridge estimator in linear regression [25], 'in-weights learning' vs. 'in-context learning' in classification [20, 33], and 'retrieval' vs. 'inference' in sequence modeling [26].

## 3.1 Validating the Memorizing and Generalizing Predictors

We next demonstrate the validity of the memorizing and generalizing predictors for the purpose of our analysis. Specifically, we show that as experimental conditions are varied, a model's behavior primarily transitions between these predictors. To this end, we consider two core phenomena associated with ICL—an increase in models' OOD performance with increasing *task diversity* $D$ [25, 60, 61], and the 'forgetting' of this ability with increasing training steps $N$—a phenomenon known as *transient generalization* [24, 30, 62]. We first replicate these results behaviorally in Fig. 1(a), finding that, across settings, Transformers transition between performing like the memorizing predictor vs. like the generalizing predictor (see App. H for full results). Then, we make a direct comparison by computing the distance between our trained model's next-token predictions and the predictions of the memorizing and generalizing predictors. Specifically, let $d(.,.)$ denote a distance measure (symmetrized KL-divergence or Euclidean distance), and denote the Transformer model trained from scratch via $h(.)$. Then, as a function of $D$ and $N$, we can plot a heatmap of the *relative distance* between the trained model and the memorizing and generalizing predictors, defined as $d_{\mathrm{rel}} = {}^{(r+1)}/_2$,

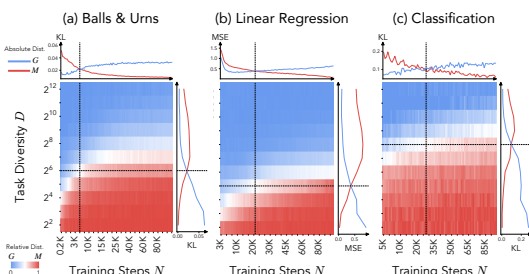

Figure 4: **Relative Distance Captures Transitions in Model Behavior.** We show the relative distance between model outputs and the two predictors. Marginals report the absolute distance values (e.g., symmetrized KL between model and predictor outputs for the Balls & Urns setting), holding $N$ constant (for the right plot), or $D$ constant (for the top plot), and varying the other variable (denoted with the dotted line). Across all settings, we see model behavior decomposes into two phases, explained by either the memorizing or generalizing predictor. In this figure, we use context length of 128, task dimensionality of 8, and MLP width of 256 for balls and urns. Linear regression has similar parameters except context length of 32, and classification has similar parameters other than MLP width of 512.

where $r := \frac{d(h,G)-d(h,M)}{d(G,M)}$. This metric evaluates to 0 vs. 1 if the model is closer to the generalizing vs. the memorizing predictor. Results are shown in Fig. 4; see App. H.3 for absolute distances. We clearly see that increasing $D$ for fixed $N$, the model first behaves like a memorizing predictor (in red), only to eventually transition to behaving like a generalizing predictor (in blue)—illustrating task-diversity effects [25]. Meanwhile, when increasing $N$ for middle values of $D$, the model starts closer to a generalizing predictor, only to eventually give way to a memorizing predictor—illustrating transient generalization [24]. More broadly, across all tasks, we see there is a clear *delineation* of the model behavior into two phases of $(N, D)$, such that model behavior is best explained by either the memorizing or the generalizing predictor in a given phase. Given the optimality of these predictors on the distribution of seen tasks ($\mathcal{T}_{\mathrm{train}}$) or the underlying distribution ($\mathcal{T}_{\mathrm{true}}$), our analysis provides an explanation for *why* these predictors were observed in prior work. However, several questions remain, including why a generalizing strategy is learned *even when it leads to worse ID performance*, and *why does varying experimental conditions change which strategy, among the memorizing and generalizing predictors, is implemented by a model*. We address these questions next.

## 4 Answering the Why: A Hierarchical Bayesian Account of ICL

Our analysis above shows that, except for intermediate values of $N$ and $D$, model behavior is primarily explained by Bayes-optimal predictors capturing the distributions $\mathcal{T}_{\mathrm{train}}$ and $\mathcal{T}_{\mathrm{true}}$. Motivated by these findings, we adopt the lens of *rational analysis* [38–42], a framework in cognitive science that aims to explain a learner's behavior as *optimal*, under *computational constraints*. What might be considered optimal in our case? Recall the fact that ICL is an *inductive* problem, i.e., a problem of predicting the next observation given past ones. Specifically, a predictor $h^{\mathrm{pred}}$ performing ICL predicts the $i^{\mathrm{th}}$ token $\boldsymbol{s}_i$ given previous elements in the sequence $\boldsymbol{s}_{1:i-1}$, using mechanisms it may have learned for this purpose based on sequences $S_{\mathcal{T}_{\mathrm{train}}}(N, D)$ seen in training (denoted $S_{\mathcal{T}_{\mathrm{train}}}$ from hereon for brevity). Then, given a hypothesis space of possible solutions the model has learned, Bayesian inference prescribes an optimal way to solve this problem via the *posterior predictive distribution*: compute a weighted average of predictions from each solution, with weights defined by

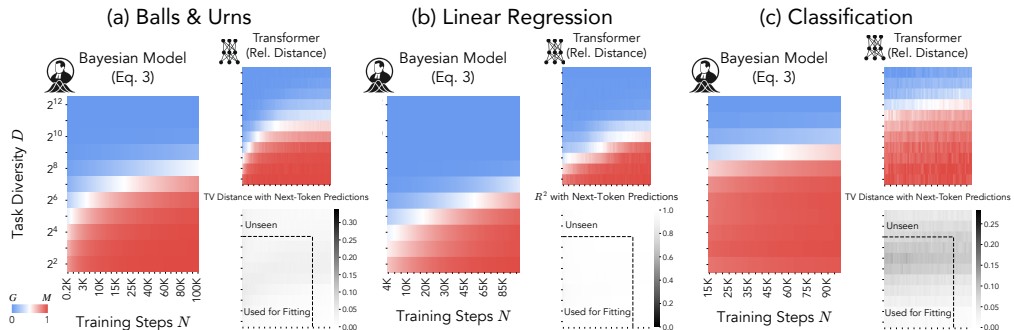

Figure 5: **Our Bayesian Model Captures Transitions Between Strategies Explaining Model Behavior.** We plot the posterior probability of the memorizing predictor given by our theoretical model (Eq. 2). Across three broad experimental settings—**(a)** Balls & Urns, **(b)** Linear Regression, and **(c)** Classification—we find our model identifies the phases best explained by a given predictor and the boundary between them, hence capturing the transition between solutions seen in a Transformer's training (as shown by the relative distance maps). Importantly, our model is highly predictive of the pretrained Transformer's behavior (next-token predictions) across conditions used for fitting the three free parameters of our model and unseen ones. Max color bar value for Balls & Urns and Classification is determined by the performance of a baseline predictor that always outputs the mean of the distribution $\mathcal{T}_{\text{true}}$. In this figure, we use context length of 256, task dimensionality of 8 and MLP width of 256 for balls and urns. Linear regression and classification have similar parameters except context length of 64 and 384, respectively.

a solution's posterior probability (i.e., a *posterior-weighted average*). Relying on the results of Sec. 3, we can assume our hypothesis space simply consists of the memorizing and generalizing predictors[2]. Thus, in our case, each solution itself corresponds to a Bayesian predictor—specifically, predictors $M$ and $G$—hence resulting in the following *hierarchical Bayesian model*.

$$h^{\text{pred}}(\boldsymbol{s}_i|\boldsymbol{s}_{1:i-1}, S_{\mathcal{T}_{\text{train}}}) = \sum_{Q \in \{M,G\}} p(Q|S_{\mathcal{T}_{\text{train}}})\, Q(\boldsymbol{s}_i|\boldsymbol{s}_{1:i-1}). \tag{1}$$

The mathematical form of the predictor above frames in-context behavior as a *linear interpolation* in $M$ and $G$, with posterior probabilities $p(M|S_{\mathcal{T}_{\text{train}}}), p(G|S_{\mathcal{T}_{\text{train}}})$, estimated from training, determining the interpolation weights. Then, in order to use this model to explain how a neural network performs ICL, we must estimate how posterior probabilities vary across training and data conditions.

**Modeling the Posterior Probabilities.** The posterior probability for a predictor $Q$, i.e., $p(Q|S_{\mathcal{T}_{\text{train}}}) \propto p(S_{\mathcal{T}_{\text{train}}}|Q)p(Q)$, is comprised of a likelihood term and a prior term—thus, these are the two terms we must estimate. In line with the perspective of rational analysis, in modeling these terms, we consider the following two well-known *computational constraints* of neural networks.

- **A1:** Loss scales in a power-law manner with dataset-size $N$, i.e., $L(N) \approx L(\infty) + {}^A\!/\!{}_{N^\alpha}$, where $L(N)$ denotes the loss after $N$ updates. This leads the log-likelihood to scale with effective sample size $\gamma N^{1-\alpha}$ rather than $N$ (see App. D.1): $\log p(S_{\mathcal{T}_{\text{train}}}|Q) = \gamma N^{1-\alpha}\, \mathbb{E}_{\boldsymbol{s} \sim \mathcal{T}_{\text{train}}}[\log p(\boldsymbol{s}|Q)]$.
- **A2:** Neural networks exhibit a bias toward simpler solutions. Specifically, using $K(Q)$ to denote the Kolmogorov complexity for predictor $Q$, we accommodate the Transformer-specific implementation cost by defining $K_{\text{T}}(Q) = K(Q)^\beta$. Then, taking the form of a universal prior, the prior probability of learning a predictor $Q$ is $p(Q) \propto 2^{-K_{\text{T}}(Q)} = 2^{-K(Q)^\beta}$.

**A1** is merely a paraphrased version of well-known power-law scaling behaviors seen in neural network training [49, 50] and dictates how quickly observed data updates model behavior. That is, it offers a functional form for the rate at which likelihood in a posterior calculation grows, i.e., the rate of evidence accumulation. Meanwhile, **A2** is a well-known inductive bias of neural networks [43–48]. Our specific functional form for the prior is grounded in algorithmic information theory: Kolmogorov complexity $K(Q)$ is the length of the shortest program on a universal Turing machine that implements $Q$, and the coding theorem relates it to probability via $p(Q) \propto 2^{-K(Q)}$ [63–65]. Following common

---

[2]While in principle other predictors may be needed to capture model behavior, in the settings we analyze, we find that *very quickly* into training, other predictors perform poorly compared to $M$ and $G$ in predicting model behavior, and thus focus on these predictors as our primary hypotheses (see 4.2, App. E for further discussion).

practice [44, 45, 66–69], we estimate Kolmogorov complexity (which is uncomputable) as $\tilde{K}$ via lossless compression: we apply several lossless compressors to the code and data for $Q$, and take the smallest resulting size (App. F.2). For brevity, we write $\tilde{K}$ as $K$ below.

Returning to our goal, we now consider a model trained for $N$ iterations on a task-mixture $\mathcal{T}_{\text{train}}$ of diversity $D$. The log-posterior odds of the two predictors can be defined as follows.

$$\eta(N, D) := \log \underbrace{\frac{P(M|S_{\mathcal{T}_{\text{train}}})}{P(G|S_{\mathcal{T}_{\text{train}}})}}_{\text{Posterior odds}} = \log \underbrace{\frac{P(S_{\mathcal{T}_{\text{train}}}|M)}{P(S_{\mathcal{T}_{\text{train}}}|G)}}_{\text{Bayes factor}} + \log \underbrace{\frac{P(M)}{P(G)}}_{\text{Prior odds}}.$$

Under constraints **A1**, **A2**, this simplifies as follows (see App. D.1).

$$\eta(N, D) = \underbrace{\gamma N^{1-\alpha} \Delta L(D)}_{\text{Loss term}} - \underbrace{\Delta K(D)^{\beta}}_{\text{Complexity term}}, \tag{2}$$

where $\Delta K(D)^{\beta} := K(M_D)^{\beta} - K(G)^{\beta}$ is the difference between the exponentiated Kolmogorov complexity of the two predictors (with $M_D$ denoting the memorizing solution defined for $D$ tasks); $\Delta L(D) := \overline{L}_G(\mathcal{T}_{\text{train}}(D)) - \overline{L}_{M_D}(\mathcal{T}_{\text{train}}(D)) = -(\mathbb{E}_{\boldsymbol{s} \sim \mathcal{T}_{\text{train}}}[\log p(\boldsymbol{s}|G)] - \mathbb{E}_{\boldsymbol{s} \sim \mathcal{T}_{\text{train}}}[\log p(\boldsymbol{s}|M_D)])$ is the difference between the average loss of the two predictors for sequences sampled from $\mathcal{T}_{\text{train}}$; Finally, $\gamma$ is a constant related to the term $A$ from constraint **A1**. To get the posterior probabilities for $M$ and $G$, we simply convert $\eta$ via the sigmoid function, denoted $\sigma(\cdot)$, yielding:

$$h^{\text{pred}}(\boldsymbol{s}_i|\boldsymbol{s}_{1:i-1}, S_{\mathcal{T}_{\text{train}}}) = \sigma(\eta(N, D)) M(\boldsymbol{s}_i|\boldsymbol{s}_{1:i-1}) + (1 - \sigma(\eta(N, D))) G(\boldsymbol{s}_i|\boldsymbol{s}_{1:i-1}). \tag{3}$$

Note that the free parameters of this Bayesian model, i.e., $(\alpha, \beta, \gamma)$, depend on the problem setting and the Transformer's learning dynamics on it. To identify their values, we simply fit the Bayesian model's predictions to the pretrained Transformer $h(.)$'s next-token predictions on inputs retrieved from a subset of values $(N, D)$. We emphasize that we *only fit three free parameters* across model checkpoints in 11 different training runs to get our results.

**Validating the Model.** We now check whether our model accurately captures the behavior of the pretrained Transformer and reproduces ICL's phenomenology. As shown in Fig. 5, our Bayesian model yields an *almost perfect prediction* of Transformer next-token predictions for both seen / unseen settings. Moreover, without fitting to the relative distance maps, we find an *almost perfect match* between the posterior probabilities of the memorizing predictor given by our model and relative distance values. These results are replicated across *72 different maps* in App. H.4 with varying MLP width (see Fig. 7(a)), context length, and task dimensionality, yielding robust support for our model. Across all maps, we find our model is highly predictive of Transformer next-token predictions, with a mean $R^2$ of $0.97 \pm 0.004$ (SE) in Linear Regression, a mean agreement of $0.92 \pm 0.007$ in Classification, and a mean Spearman rank correlation of $0.97 \pm 0.001$ in Balls & Urns. Additionally, we find strong correlations of $0.99, 0.98, 0.99$ between our model's posterior probabilities and the relative distance values given by the Transformer in the Linear Regression, Classification, and Balls & Urns settings, respectively. Finally, we also examine ablations of our functional form in App. I, showing the computational constraints we assume are necessary for the success of our framework.

## 4.1 Predictions

We next analyze our model to make informative qualitative predictions. Unless stated otherwise, we use Balls & Urns setting with context length 128, task dimensionality of 8, and MLP width of 256.

- **Sub-linear sigmoidal growth from generalizing to memorizing.** We examine whether relative distance, which acts as an estimate for the posterior of the memorizing solution, shows the functional form predicted by our model: sublinear growth with $N$, and sigmoidal growth with $N^{1-\alpha}$. This is strongly validated in Fig. 6(a) by fitting a parameterized logistic to predict relative distance from $N^{1-\alpha}$ for each curve (i.e., holding $D$ constant). Fig. 6(a), bottom panel, also shows that even at high $D$, relative distance slowly increases towards the memorizing predictor in a sigmoidal manner, contrasting with Raventós et al. [25]'s claim that at high $D$ the Transformer only becomes *closer* to the generalizing predictor with increasing $N$ (see App. J).

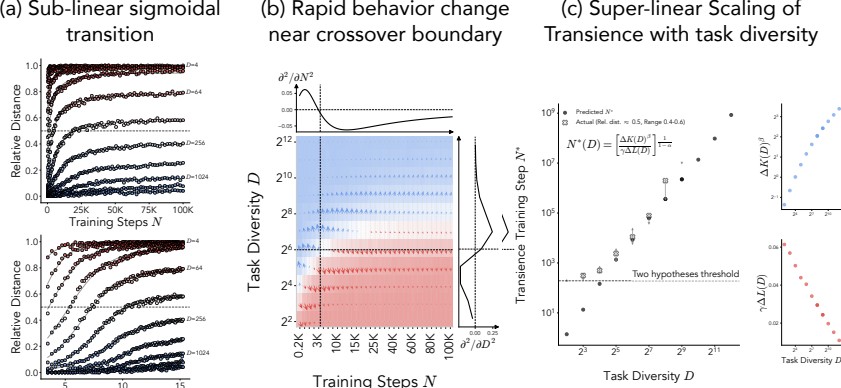

Figure 6: **Predictions from Our Model.** **(a)** Relative distance is predicted to show sub-linear scaling with $N$, and sigmoidal growth with $N^{1-\alpha}$. This is validated by a strong match between the empirical curves and parameterized logistic fits (dark lines). Note, however, that some fitted plateau parameters fall below 1, implying *never* reaching full memorization, whereas Bayesian trajectories should reach it *eventually* (see App. K). **(b)** We predict a rapid change in model behavior for intermediate values of $N$ and $D$, yielding a crossover-like boundary between phases dominated by each predictor. This can be seen via the magnitude of the second derivative of the relative distance near the boundary. Marginal plots show the second derivative of the relative distance with respect to $N$ or $D$. **(c)** Our analysis predicts super-linear scaling of the time of transience $N^*$ as diversity $D$ increases.

- **Rapid behavior change near crossover.** At the point where the two hypotheses have equal log-posterior odds, our model predicts there will be a crossover in which predictor dominates the posterior and explains model behavior. Given our sigmoidal functional form for $\eta(N, D)$, we can expect this change to be rapid—small variations in experimental conditions (e.g., task diversity) will yield large changes in model behavior. To confirm this, we plot the second derivative of the relative distance in Fig. 6(b), finding its magnitude is indeed greatest near the boundary $\eta(N, D) = 0$.

- **Scaling of time to Transience with Task-Diversity.** For a given value of $D$, we can predict the critical training time at which a crossover from generalizing to memorizing will occur by solving for $N^*$ in $\eta(N^*, D) = 0$ (i.e., relative distance $d_{\text{rel}} = 0.5$): $N^*(D) = \left(\frac{\Delta K(D)^\beta}{\gamma \Delta L(D)}\right)^{\frac{1}{1-\alpha}}$. Observing Fig. 6(c), we find that beyond the two-hypotheses threshold (which determines the minimum $N$ from which our model holds, App. E), our predictions for $N^*$ hold well for several orders of magnitude, before observations diverge at higher $D$, as transience is *slower* than predicted[3]. These results indicate *super-linear growth of time to transience with task diversity*. Importantly, according to our expression, at high $D$ the denominator is very small and hence *time to transience can approach infinity*. In such a condition, generalization persists *regardless* of training time.

## 4.2 Extending the Model

We additionally examined two extensions of our model: **(1) Multiple Strategies:** we add a constant mean-predictor solution, which has been shown to emerge very early in training [70, 71], to modeling the linear regression setting. Denoting this *optimal constant* solution as $C$, our model becomes: $h^{\text{pred}}(s_i|s_{1:i-1}, S_{\mathcal{T}_{\text{train}}}) = \sum_{Q \in \{M, G, C\}} p(Q|S_{\mathcal{T}_{\text{train}}}) Q(s_i|s_{1:i-1})$. As Fig. 7(b) and App. H.5 show, our model successfully generalizes to capturing complex dynamics with multiple predictors, including a sharp transition from a context-insensitive solution to ICL, which has been observed in other settings [29, 72]. Our model explains this "rapid emergence" constant solution is preferred early due to its simplicity, yet eventually, low likelihood leads to its posterior dropping sharply (Fig. 7(b), bottom).

**(2) In-Context Strategy Selection:** We find that when evaluated on OOD data, models shift their behavior toward the generalizing solution, thereby moving the transition boundary between memorizing and generalizing (Fig. 7(c)). This makes sense from a Bayesian perspective: Given novel data, a Bayesian learner should continue updating its posterior in accordance with the likelihoods of different hypotheses. The generalizing solution outperforms memorization on OOD data, thus its

---

[3]Interestingly, we find that learning rate annealing enhances the Transformer's adherence to Bayes-optimal trajectories, and so we use it in this experiment. Yet, the effect appears to decay throughout training, yielding a slower rate of advancing toward memorization, which can be seen by the last two observed transience points veering from our model's predictions (see Fig. 6(c) and App. K).

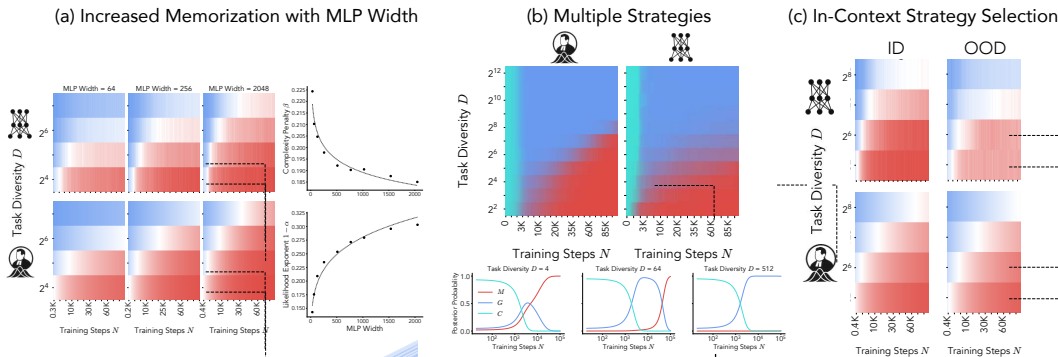

Figure 7: **Our framework captures diverse ICL phenomenology. (a)** Scaling MLP width raises the transition boundary between memorization and generalization, and our model captures this effect via decreasing complexity penalty and increasing sample efficiency. **(b)** Incorporating a mean-predictor solution for linear regression in our model allows capturing complex transition dynamics. **(c)** When evaluated in OOD settings, models increase their resemblance to the generalizing solution, exhibiting *in-context strategy selection*, which we model as in-context posterior updates (see App. D.4).

posterior should increase. To enable in-context posterior updates, we write: $h^{\text{pred}}(\boldsymbol{s}_i|\boldsymbol{s}_{1:i-1}, S_{\mathcal{T}_{\text{train}}}) = \sum_{Q \in \{M,G\}} p(Q|S_{\mathcal{T}_{\text{train}}}, \boldsymbol{s}_{1:i-1}) \, Q(\boldsymbol{s}_i|\boldsymbol{s}_{1:i-1})$. Our final form includes an added term for in-context loss differences, without additional free parameters (App. D.4). As shown in Fig. 7(c), our model qualitatively captures the shift towards generalization in OOD evaluation. Yet, it struggles to explain OOD behavior around the transition boundary (App. H.6), which could mean other predictors are needed to explain OOD behavior. Moreover, for the current model, we make the simplifying assumption that in-context posterior updates follow similar power-law parameters as pretraining, yet this is unlikely, and further work is needed to distinguish in-context and parametric learning [73].

## 4.3 The Loss-Complexity Tradeoff

Having formalized and demonstrated the empirical validity of our hierarchical Bayesian framework, we now discuss the intuitive interpretation it offers us. Specifically, Eq. 2 suggests that the tradeoff between posterior-odds corresponding to different predictors is driven by the *loss* a predictor achieves on the training data and its *complexity* (see Fig. 8): early in training, the prior dominates, therefore a less complex solution—the generalizing predictor in our case—will be strongly favored as per the posterior calculation. However, the memorizing predictor will almost always have a lower loss than the generalizing predictor on training data. Thus, in low-to-medium task diversity settings, as training proceeds and $N$ increases, the loss term in Eq. 2 will overtake the complexity term, i.e., the likelihood eventually dominates the posterior and 'floods' the prior. This will lead to the memorizing predictor becoming favored—explaining the transient generalization phenomenon from prior work [24]. In contrast, in high task diversity settings, the prior strongly disfavors the memorizing predictor due to its complexity, and given the sub-linear accumulation of likelihood (i.e., the $N^{1-\alpha}$ term), the time for transience to occur grows superlinearly—giving rise to the phenomenon of task diversity threshold seen in

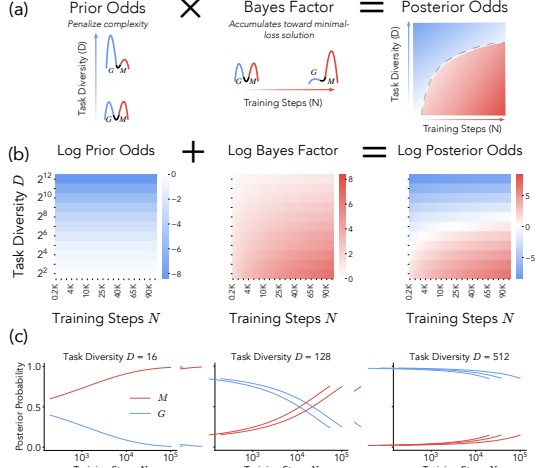

Figure 8: **Intuition Elicited by The Bayesian Model. (a)** Our framework suggests Transformers have a prior preference for learning simpler solutions. However, throughout training, preference is updated towards solutions that better explain the data (i.e., have greater likelihood). This happens even at the expense of higher complexity, which in our case, yields a transition toward a memorizing predictor. **(b, c)** Our framework captures the tradeoff between solutions, showing that the *boundary* between the two phases corresponds to *equal posterior probabilities* of the two predictors.

prior work [25]. The importance of a loss-complexity tradeoff for ICL was also arrived at by two recent papers [30, 74], which we discuss further in Sec. 5.

# 5 Discussion

In this work, we aim to unify findings from the ICL literature by asking *why* Transformers learn disparate strategies for performing ICL across varying training conditions. To do so, we take a *normative* perspective [38] which aims to explain Transformer learning as *optimal* under *computational constraints*. This lens yields a hierarchical Bayesian framework, which, assuming simplicity bias and scaling laws as computational constraints, offers a highly predictive account of model behavior: by fitting merely *three variables* to Transformer next-token predictions, we can *almost perfectly* predict model behavior across a spectrum of experimental settings *without* access to its weights. Our account also implies a fundamental trade-off that occurs throughout training between the *loss* and *complexity* of potential solutions learned by a model, with simpler, generalizing solutions learned early in training, while more complex, better-fitting solutions eventually becoming preferred. This tradeoff helps explain prior findings in the ICL literature, and provides novel predictions regarding training dynamics. Thus, we argue for our hierarchical Bayesian framework as an explanatory and predictive account of ICL, and a step towards a unified understanding of its phenomenology.

**Relation to previous Bayesian models of ICL.** Here, we discuss the relation between our work and other Bayesian or normative accounts of ICL. We provide an extended review of additional related works in App. B. Several prior works framed ICL as **Bayesian at inference-time**. Other than a few cases [17, 21], most such works view ICL as a *single* Bayesian predictor [25, 35, 51, 62, 75, 76]. The focus on a single predictor in these works has often led them to view ICL as a single strategy, with some focusing solely on a memorizing solution [35], while others refer mainly to a generalizing solution as "true" ICL [25]. While there are cases in which in-context behavior is well approximated by a single predictor (e.g., the memorizing solution for low $N, D$), our results robustly show that a mixture of predictors is required to fully capture model behavior, and that the extent to which a predictor explains model behavior *varies* across training. Thus, a posterior-weighted average over *different* predictors, which considers a bias towards different predictors coming from training, rather than only inference-time, is required to *fully* capture model behavior across conditions. In contrast with studies focusing on inference-time, two recent works, Carroll et al. [30] and Elmoznino et al. [74], offer **Bayesian or normative views of pretraining to perform ICL**. Importantly, while these papers take different theoretical perspectives from ours, they arrive at a similar conclusion as us: the existence of a tradeoff between loss and solution complexity in pretraining. Elmoznino et al. [74] offer a normative theoretical analysis of training to perform ICL via next-token prediction loss, showing it yields an Occam's razor objective which minimizes both loss and solution complexity. Carroll et al. [30] study task diversity effects and transient generalization in the linear regression setting of Raventós et al. [25]. Their Bayesian account of pretraining, which is rooted in theory of singular models [77] and makes different assumptions from ours, interestingly yields a relatively similar functional form for the posterior odds (though their form does not take into account neural scaling laws). However, their measure of complexity is architecture-dependent [78], thus they only provide a qualitative analysis as they cannot directly estimate the complexity of Bayesian predictors. In contrast, our hierarchical Bayesian framework provides a *quantitative*, predictive account of pretraining phenomena, in addition to capturing inference-time behavior as a posterior-weighted average of solutions, which is not addressed by Carroll et al. [30] or Elmoznino et al. [74]. Despite that, we view these works as valuable contributions with complementary insights to ours, in particular regarding potential explanations for the source of neural networks' simplicity bias.

**Limitations.** While we rely on a specific theoretical abstraction to arrive at our results, i.e., a hierarchical Bayesian model, we believe the predictive power of this abstraction corroborates its faithfulness. However, one limitation of our analysis is use of toy settings where model behavior is largely explained by only two predictors (with the addition of the optimal constant solution in linear regression). Accommodating more complex settings that LLMs encounter in pretraining or inference-time will be an important future test for the framework. Finally, a crucial limitation of our analysis comes from the simple relation we assume between algorithmic complexity and complexity of implementation by a Transformer—while the assumption of simplicity bias is well-backed by theoretical and empirical claims [43, 44, 46], we believe our assumed relation could be improved by building on recent advances defining architecture-dependent measures of how many effective parameters are used by a model to implement a solution [71, 78].

## Acknowledgments

We thank the Computation and Cognition Lab, in particular Ben Prystawski and Michael Li; Jay McClelland, Satchel Grant, Jerome Han and the PDP Lab; the Physics of Intelligence group at Harvard, especially Eric Bigelow and Sonia Murthy; the CRISP Lab at Harvard; Jesse Hoogland and Matthew Farrugia-Roberts; Surya Ganguli and the Neural Dynamics and Computation Lab; and Navin Goyal and the theory group at Microsoft Research India for useful discussions.

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

# Appendix

## Table of Contents

# A    Glossary of Useful Terms

**In-Context Learning (ICL).**    In this paper, we use a broad construal of ICL advanced by Lampinen et al. [37] and Olsson et al. [72]: any way in which models use their context to adapt their predictions and reduce loss can be considered as ICL. This notion of ICL encapsulates many forms of sequence modeling and instruction following, as well as the more traditional view of ICL as few-shot learning [6]. Hence, in this work, we use tasks that capture both sequence modeling [26] and few-shot learning [25, 28] notions of ICL.

**Generalizing Predictor ($G$).**    A predictor defined by a posterior-weighted average with a *continuous prior* over the true data-generating distribution $\mathcal{T}_{\text{true}}$. Such a predictor does not depend on the tasks seen during training, and can hence generalize well to novel, unseen tasks. This predictor maps onto 'task-learning' or 'inference' notions of ICL [26]. See App. G for the form of generalizing predictors in the studied settings.

**Memorizing Predictor ($M$).**    A predictor defined by a posterior-weighted average with a *discrete prior* defined over the distribution of tasks seen by the model during training, $\mathcal{T}_{train}$. Such a predictor depends on the tasks seen during training, and hence generalizes primarily to those tasks. This predictor maps onto 'task-retrieval' notions of ICL, though it also captures a context-constrained 'in-weights learning' solution in the classification task [20, 33]. See App. G for the form of memorizing predictors in the studied settings.

**Task Diversity ($D$).**    For a mixture distribution $\mathcal{T}_{\text{train}}$ defined over $D$ tasks $\{f(\boldsymbol{w}_1, \cdot), \ldots, f(\boldsymbol{w}_D, \cdot)\}$, $D$ is referred to as the *task diversity* of the mixture.

**Relative Distance ($d_{rel}$).**    A measure we use to characterize trade-off between predictors in settings where there are primarily two predictors (though one can easily generalize this measure to a setting with more predictors). Specifically, given a model $h$, two predictors $Q_1, Q_2$, and a distance function $d$, we define the term $r := \frac{d(h,Q_1) - d(h,Q_2)}{d(Q_1,Q_2)}$ and then relative distance as $d_{\text{rel}} = {(r+1)}/{2}$. The latter operation rescales $r$ to a scale of 0 (if $h = Q_1$) to 1 (if $h = Q_2$), essentially assessing where the model $h$ lives when a line is drawn with endpoints ranging from $Q_1$ to $Q_2$.

**Posterior Odds, Prior Odds, Bayes Factor.**    Consider a set of observations $X$ and two hypotheses $H_1$ and $H_2$ that are being assessed as candidates to explain the data. *Prior odds* are defined as the ratio $\frac{P(H_1)}{P(H_2)}$, i.e., if no observations are seen yet, which hypothesis is apriori preferred. *Bayes factor* is defined as the ratio of likelihoods $\frac{P(X|H_1)}{P(X|H_2)}$, i.e., once the observations are received, we compare how likely individual hypotheses deem these observations. Finally, *posterior odds* are defined as the ratio $\frac{P(H_1|X)}{P(H_2|X)} = \frac{P(H_1)}{P(H_2)} \times \frac{P(X|H_1)}{P(X|H_2)} = \text{Prior odds} \times \text{Bayes factor}$, i.e., how do observations affect the prior to help assess which hypothesis is more likely.

**Rational Analysis.**    Rational analysis is a framework in cognitive science introduced by John R. Anderson [38], in which the behavior of a learner is explained as an optimal adaptation to its learning environment, given its goal and computational constraints. In the case of training a neural network, the learning environment is given by the training data distribution, the goal is given by the optimization objective, and the computational constraints can be seen as constraints and inductive biases of the architecture and optimization algorithm. The framework of rational analysis is characterized as *normative*, since it specifies how a learner *should* behave, in an optimal sense, given its environment, goals, and computational constraints. This approach has been successfully applied in cognitive science to explain a wide range of phenomena in humans [38, 39, 41, 79, 80].

**Transient Generalization ($N^*$).**    For a given task diversity $D$, when a model is trained for sufficiently long, its behavior transitions from more closely resembling a generalizing predictor to more closely resembling a memorizing one. We define this point as the point at which the relative distance between memorizing and generalizing crosses 0.5. This phenomenon is called transience, transient generalization, or the transient nature of in-context learning (see App. B for a longer discussion). The critical time to reach this point is denoted $N^*$ in the main paper.

# B Related Work

## B.1 Prior Work Studying Task-Diversity Effects and Transience

We first discuss prior work studying the phenomenology of ICL that forms the core of our paper: task diversity effects and transience. These works often focus on providing *mechanistic accounts* that are more bottom-up in nature, i.e., suitable for studying specific settings. For example, the induction head is a core mechanism employed by a model to perform the in-context classification task by Singh et al. [24], Reddy [28]. When transience occurs, the role of induction head is diminished, since it is not vital for implementing a memorizing predictor. However, as we show, transience occurs across a spectrum of settings, including ones where induction heads are never learned during training (e.g., Balls and Urns, which we analyze using a one-layer Transformer)—prior mechanistic accounts would not help justify transience in such settings. Our work thus takes a top-down account, i.e., we offer insights based on a *computational model* of ICL developed by capturing its phenomenology. This helps identify relevant control variables controlling a dynamic we argue lies at the heart of transience and task diversity effects: tradeoff between loss and complexity of a solution, which manifests itself into transitions between generalizing and memorizing predictors. Reconciling our computational model with mechanistic approaches undertaken in prior work can be an exciting avenue for progress.

- **Task diversity Threshold.** A phenomenon first popularized by Raventós et al. [25] in an in-context linear regression task, albeit originally demonstrated by Kirsch et al. [61] in a prior work on in-context classification of permuted MNIST images, and recently expanded to a Markov modeling setting by Park et al. [26], to another classification setting by Nguyen and Reddy [29], as well as to a modular arithmetic setting by He et al. [60]. Specifically, Raventós et al. [25] empirically show that if one trains Transformers on a mixture of linear regression tasks, there is a critical number of tasks below which the model behavior is well-characterized by a Bayesian predictor over the seen tasks (what we term the memorizing predictor), while above it the behavior is well-characterized by the standard solution of ridge regression. These results were expanded in a recent theoretical work by Lu et al. [81], who study the asymptotic dynamics of a linear attention Transformer and show the change in solution used by the model as a function of task diversity is a second-order phase transition. In contrast to their work, our empirical analysis covers a broad range of settings that involve sequence modeling, regression, and classification, while the analytical parts of our paper provide a predictive framework that identifies the relevant control variables and explains how they affect the behavior of standard, nonlinear Transformers across all studied settings.

- **Transience / Transient Nature of In-Context Learning.** Originally observed by Chan et al. [20] while investigating the effects of data-centric properties on ICL, the term was popularized by Singh et al. [24]. Specifically, focusing on an in-context classification task, Singh et al. [24] showed that a model's ability to perform the generalizing ICL solution (employing a copy mechanism via the induction head) goes away when trained long enough. This phenomenon was recently generalized to a Markov modeling task by Park et al. [26] and to simplified variants of the in-context classification setting by Chan et al. [27], Nguyen and Reddy [29]. We especially emphasize the work of Nguyen and Reddy [29], who empirically identify a competition dynamic between a memorizing and generalizing solution for their task, building on this observation to perform a gradient flow analysis of their task and developing an effective theory of how transient generalization occurs. Our work differs in the sense that we provide an account of the competition dynamic and a model for its origins, instead of a gradient flow account that operates under the assumption of competition.

## B.2 Hierarchical Bayesian Models of Learning to Learn

Hierarchical Bayesian models have been widely employed in cognitive science as models of 'learning to-learn', or meta-learning, in humans [82–87]. These models allow the agent to learn the prior distribution from the data, rather than the prior being predefined. This allows learning inductive biases that constrain hypothesis spaces (called over-hypotheses), which can instruct future learning [86]. Additionally, it entails estimating evidence for several different hypothesis spaces (like the continuous vs. discrete priors in our work) and using these hypothesis spaces for generalization [85]. Note also that much of the work on hierarchical Bayesian models in cognitive science takes a normative perspective. Finally, we want to highlight the work of Grant et al. [88], who explicitly

analyze a well-known meta-learning algorithm (MAML [89]) via a hierarchical Bayesian lens, as well as Binz et al. [90], who highlight connections between meta-learning and hierarchical Bayesian models. We have been broadly influenced by this literature in approaching the study of ICL, which we believe has many direct parallels with work on learning to learn.

### B.3 Broader Work on Understanding ICL

A crucial benefit of our Bayesian modeling framework is that one can retrieve posterior probabilities over the predictors the model predictions are being decomposed into. In other words, we can represent the model predictions as an interpolation of the predictors' outputs (since posterior probabilities sum to 1). Park et al. [26] propose a similar approach: "Linear Interpolation of Algorithms" (LIA). Specifically, LIA represents model predictions as an interpolation of the predictors' outputs, directly fitting the weights for interpolation. That is, for each condition of training time ($N$), task diversity ($D$), LIA requires fitting a set of parameters, summing up to a total of $N \times D$ parameters fit (across many of the maps in our work, this would mean over $900$ parameters per combination of context-length, task dimensionality, and MLP width). Meanwhile, our framework, by defining a precise functional form that explicates the role of $N$ and $D$, minimizes the number of terms needed to perform fitting to 3—these terms are primarily related to model family, its learning dynamics, and intrinsic randomness of the data, hence making them hard to explicate. Crucially, one can see our framework as providing grounding to LIA: under the hood, LIA is trying to identify the posterior probabilities for all experimental settings! Moreover, our framework offers an explanation for *why* change in predictor weights occurs during training, and importantly, can *predict* dynamics of $N, D$ conditions it was not trained on, which LIA cannot do since it has to be fit separately to each $N, D$ condition.

Beyond the papers discussed above, there have been several complementary efforts to understand ICL from different perspectives (see the recent summary by Lampinen et al. [37] for a longer review). For example, several works, some briefly discussed above, characterized the trade-off between memorizing and generalizing in a more mechanistic way, as a competition dynamic between circuits [24, 26, 29, 33]. We see our results as elucidating the forces driving this competition, as well as demonstrating its consistency across several settings. Additionally, several papers analogize ICL as implicitly performing optimization to learn novel tasks [22, 36, 53] or to meta-learning in general [91]; develop scaling laws for the sample efficiency of ICL [75]; identify limits of tasks that can be learned in-context [14, 17, 92]; demonstrate sudden learning curves exist in ICL [11, 54]; and characterize mechanisms employed by a model to perform ICL [13, 31, 34, 93]. Broadly, all these papers offer useful insights that are complementary to ours.

### B.4 A Note Regarding Complexity

Our setup in this work was quite specific in the sense that, in all cases, task diversity was the variable by which we titrated the complexity of implementing the memorizing solution. However, it should be noted that our theory and the simplicity bias we assume refer broadly to the complexity of implementing predictive solutions, which can arise from other sources in diverse data distributions. Furthermore, while in our setup, the generalizing solution was often simpler than the memorizing one given high enough task diversity, in many cases, the optimal generalizing solution can be quite complex, and other heuristic solutions (which still do not involve memorizing the training distribution) may be preferred. An excellent example of such a dynamic is provided by Qin et al. [94], who show that more complex training data can indeed drive learning of a hierarchical generalizing solution (rather than a linear, heuristic solution), whereas in simple-data conditions, models learn the heuristic solution.

## C  Takeaways and Future Work

**Takeaways.**   We see several main takeaways from our work for understanding ICL, training dynamics, and generalization behavior in neural networks more broadly.

- **Is ICL Bayesian? Depends on the Assumptions.** There exists debate regarding whether ICL can be viewed as Bayesian [25, 35, 62, 95]. Some studies that saw the emergence of a generalizing predictor have categorized it as 'non-Bayesian' [25, 62], since it is not the Bayes-optimal solution given the training distribution (which is the memorizing solution). In contrast, taking the perspective

of rational analysis, the right question is not *whether* ICL is Bayesian, but under *what assumptions* is it Bayesian. Clearly, the generalizing predictor *is* Bayes-optimal with respect to the true distribution $\mathcal{T}_{\text{true}}$. Moreover, as we show in this work, when taking into account a *simplicity bias* over predictors, learning a generalizing predictor *can* be considered Bayes-optimal in certain conditions even when it is not the optimal strategy for minimizing train loss. Thus, by extending a Bayesian perspective on ICL to both training *and* inference-time, we show that assessing the Bayes-optimality of ICL requires considering the model's bias towards different predictors coming from pretraining. Cautiously, given our highly predictive results, we conclude that ICL can be considered as approximately Bayesian *given* constraints (assumptions) of a simplicity bias and sublinear sample efficiency (though see App. K for discussion of a deviation from Bayes-optimality that is not accommodated by our current assumptions).

- **A Loss-Complexity Tradeoff is Fundamental to Understanding Training Dynamics.** We find that a tradeoff between the loss and complexity of solutions learned by models lies at the heart of ICL phenomenology. We believe the hierarchical view from which this tradeoff emerges, in which pretraining is a process of updating posterior probability for different solutions based on their complexity and loss, and ICL is a posterior-weighted average of solutions, can be a powerful explanatory perspective for understanding Transformer learning dynamics more broadly.

- **The Value of a Normative Perspective.** An important takeaway that we believe may be of independent interest to the community is that, to understand generalization behavior in Transformers and neural networks more broadly, it may be enough to observe the structure of the data, and assume the network is well-approximated by a Bayes-optimal density estimator with a simplicity bias and sublinear sample efficiency. We hope our work shows that such a top-down *normative* perspective can provide highly predictive accounts, as well as potential explanations for *why* models behave the way they do. We encourage a wider adoption of this approach for understanding neural network behavior.

Our work also opens up several exciting avenues for further progress.

- **Can we Explain In-Context Transitions?** First, we note the phase transition elicited in this work primarily assumes a biased optimization process, and the ability to overcome this bias by seeing data supporting another solution. Accordingly, since ICL can be viewed as an optimization process [22, 96], it may be possible to use our results to explain behavioral transitions seen in recent work on in-context learning of novel concepts or behaviors [11, 54, 97].

- **Does our Framework Explain other Pretraining Phenomena?** we believe the competition dynamic characterized in our work is similar to the one demonstrated by Qin et al. [94] in a language modeling task. There, the authors show that depending on the data diversity and complexity, a model can either learn a bag of heuristics or the underlying grammar to generate sentences from the language. It may be possible to explain the phenomenology elicited in that work via the hierarchical Bayes lens we take in this paper, since there likely exists a tradeoff between compressibility and loss of a heuristic vs. grammar-learning solution.

- **Connecting our top-down framework to a bottom-up mechanistic account.** Our work intentionally takes a top-down approach, and hence does not offer a mechanistic account of how Transformer learning dynamics implement the loss-complexity tradeoff, or how the model weights different solutions at inference time. Such a bottom-up analysis likely requires studying either the gradient flow dynamics of ICL [29] or using mechanistic interpretability tools to examine circuits [32, 33].

## D   Derivations

Below, we derive formal expressions for the functional form of log-posterior odds (Eq. 2) and show how one can convert it into a predictive model (Eq. 3). For completeness, we repeat below the constraints underlying our modeling framework.

- **A1:** Loss scales in a power-law manner with dataset-size $N$, i.e., $L(N) \approx L(\infty) + A/N^\alpha$, where $L(N)$ denotes the loss after $N$ updates, and $A$ is a constant that depends on model loss at initialization and training hyperparameters.

- **A2:** Neural networks exhibit a bias toward simpler solutions. Specifically, using $K(Q)$ to denote the Kolmogorov complexity for predictor $Q$, we accommodate the Transformer-specific

implementation cost by defining $K_\mathrm{T}(Q) = K(Q)^\beta$. Then, taking the form of a universal prior, the prior probability of learning a predictor $Q$ is $p(Q) \propto 2^{-K_\mathrm{T}(Q)} = 2^{-K(Q)^\beta}$.

## D.1 Log-Posterior Odds

We consider a parameterized model class $H(.)$ learning to implement a predictor $Q \in \{M, G\}$ when trained using a learner $T$ on a dataset $S_{\mathcal{T}_\mathrm{train}}(N, D)$ of $N$ sequences sampled from the distribution $\mathcal{T}_\mathrm{train}$ with diversity $D$. Note that, when required for clarity, we use $M_D$ to denote the memorizing solution defined for a dataset with task diversity $D$, but in most cases we use $M$ for brevity. We approximate this model's learning dynamics via a hierarchical Bayes framework, i.e., we assume learning happens via a posterior update by computing likelihood of the data under all considered hypotheses (which are themselves Bayesian predictors, hence the term 'hierarchical'). Each hypothesis has an associated prior that reflects the learning pipeline's proclivity towards implementing it. For brevity, we will use the notation $S_{\mathcal{T}_\mathrm{train}}$ to refer to the dataset, with sequences seen at update $N$ denoted via a superscript $S_{\mathcal{T}_\mathrm{train}}^{(n)}$; i.e., $S_{\mathcal{T}_\mathrm{train}} = \cup_n S_{\mathcal{T}_\mathrm{train}}^{(n)}$. We also use $\Theta_Q$ to denote the set of parameters in the landscape of model-class $H$, such that $H(\theta, \boldsymbol{s}) = Q(\boldsymbol{s})$ for any sequence $\boldsymbol{s}$ if $\theta \in \Theta_Q$.

We begin by analyzing the posterior of the predictor $Q$ learned by the model:

$$
\begin{aligned}
P(Q|S_{\mathcal{T}_\mathrm{train}}, T, H) &= P(\Theta_Q|S_{\mathcal{T}_\mathrm{train}}, T, H) \\
&= \int_{\theta \in \Theta_Q} P(\theta|S_{\mathcal{T}_\mathrm{train}}, T, H) \\
&\propto \int_{\theta \in \Theta_Q} P(S_{\mathcal{T}_\mathrm{train}}^{(1)}, \dots, S_{\mathcal{T}_\mathrm{train}}^{(N)}|\theta) \, P(\theta|T, H) \\
&\overset{\mathbf{A1}}{=} \int_{\theta \in \Theta_Q} \prod_{n=1}^{N_\mathrm{eff}} P(S_{\mathcal{T}_\mathrm{train}}^{(n)}|\theta) \, P(\theta|T, H) \\
&= \int_{\theta \in \Theta_Q} \prod_{n=1}^{N_\mathrm{eff}} P(S_{\mathcal{T}_\mathrm{train}}^{(n)}|Q) \, P(\theta|T, H) \\
&= \prod_{n=1}^{N_\mathrm{eff}} P(S_{\mathcal{T}_\mathrm{train}}^{(n)}|Q) \cdot \underbrace{\int_{\theta \in \Theta_Q} P(\theta|T, H)}_{\text{Prior of the learner and model-class towards learning } Q} \\
&\overset{\mathbf{A2}}{\propto} \prod_{n=1}^{N_\mathrm{eff}} P(S_{\mathcal{T}_\mathrm{train}}^{(n)}|Q) \cdot (2^{-K(Q)^\beta})
\end{aligned}
$$

We now take the log of this quantity, thereby examining the *unnormalized log-posterior* for $Q$:

$$
\begin{aligned}
&\log \prod_{n=1}^{N_\mathrm{eff}} P(S_{\mathcal{T}_\mathrm{train}}^{(n)}|Q) \cdot (2^{-K(Q)^\beta}) \\
&= \sum_{n=1}^{N_\mathrm{eff}} \log P(S_{\mathcal{T}_\mathrm{train}}^{(n)}|Q) - K(Q)^\beta \\
&= N_\mathrm{eff} \underbrace{\frac{1}{N_\mathrm{eff}} \sum_{n=1}^{N_\mathrm{eff}} \log P(S_{\mathcal{T}_\mathrm{train}}^{(n)}|Q)}_{\text{Average log likelihood for sequences drawn from } \mathcal{T}_\mathrm{train} \text{ under predictor } Q} - K(Q)^\beta \\
&= -N_\mathrm{eff} \overline{L}_Q(\mathcal{T}_\mathrm{train}(D)) - K(Q)^\beta.
\end{aligned}
$$

Overall, we have then

$$
\begin{aligned}
\eta(N, D) &:= \log \frac{P(M|S_{\mathcal{T}_{\text{train}}}, T, H)}{P(G|S_{\mathcal{T}_{\text{train}}}, T, H)} \\
&= N_{\text{eff}} \left( \overline{L}_G(\mathcal{T}_{\text{train}}(D)) - \overline{L}_{M_D}(\mathcal{T}_{\text{train}}(D)) \right) - \left( K(M_D)^\beta - K(G)^\beta \right) \\
&= N_{\text{eff}} \Delta L(D) - \Delta K(D)^\beta.
\end{aligned}
\tag{4}
$$

In the above, our constraints get operationalized as follows.

- **A1** helps us accommodate the fact that while a Bayesian learner would make optimal use of all samples shown to it, neural network training in fact makes suboptimal use of samples seen during training, which we model by defining $N_{\text{eff}}$, i.e., the effective number of samples a neural network learns from. We will estimate this value below in Eq. 6.

- **A2** provides a form for the prior the learning pipeline (includes the learner $T$ and model-class $H$) has towards implementing the predictor $Q$.

We next model $N_{\text{eff}}$. Specifically, we use the power-law scaling behavior of neural networks' learning dynamics to compute the loss reduced in $N$ updates by such a pipeline, identifying the number of updates an idealized Bayesian learner would have to make in order to reduce loss by this amount.

$$
\begin{aligned}
N_{\text{eff}} &:= \frac{\text{Loss under power-law scaling in } N \text{ updates}}{\text{Loss of a Bayesian learner in a single update}} \\
&= \frac{\sum_{n=1}^N (L(n) - L(\infty)) \delta n}{\overline{L}_Q} \\
&= \frac{1}{\overline{L}_Q} \sum_{n=1}^N \frac{A}{n^\alpha} \delta n \\
&= \frac{A}{\overline{L}_Q} N^{1-\alpha} \int_0^1 \frac{1}{\hat{n}^\alpha} \delta \hat{n} \\
&= \frac{A}{(1-\alpha) \overline{L}_Q} N^{1-\alpha} \\
&= \gamma N^{1-\alpha},
\end{aligned}
$$

where $\overline{L}_Q = \overline{L}_Q(\mathcal{T}_{\text{train}}(D)) = -\mathbb{E}_{s \sim \mathcal{T}_{\text{train}}}[\log p(s|Q)]$, $\hat{n} = {}^n/_N$ and $\gamma = \frac{A}{(1-\alpha) \overline{L}_Q}$ is a constant that subsumes the loss of the predictor $Q$ and the constant $A$ from our assumed form of power-law scaling, which depends on the random loss of a network and effects of hyperparameters like batch-size and sequence lengths used to define the train data.

Substituting $N_{\text{eff}}$ back into Eq. 4, we get our final model:

$$
\boxed{\eta(N, D) = \gamma N^{1-\alpha} \Delta L(D) - \Delta K(D)^\beta.}
\tag{5}
$$

### D.2 Converting from Posterior-Odds to a Predictive Model

At inference, the pretrained Transformer is shown a sequence $s$, for which it makes a next-token prediction. To simulate this process in our framework, we define a Bayesian predictor, denoted $h^{\text{pred}}$, as follows.

$$
\begin{aligned}
h^{\text{pred}}(s_i|s_{1:i-1}, S_{\mathcal{T}_{\text{train}}}) &:= \sum_{Q \in \{M, G\}} P(Q|S_{\mathcal{T}_{\text{train}}}, T, H) Q(s_i|s_{1:i-1}) \\
&= \sum_{Q \in \{M, G\}} \underbrace{P(Q|S_{\mathcal{T}_{\text{train}}}, T, H)}_{\text{Pretraining Prior}} \underbrace{Q(s_i|s_{1:i-1})}_{\text{Prediction}} \\
&= \frac{\exp(\eta(N, D))}{1 + \exp(\eta(N, D))} M(s_i|s_{1:i-1}) + \frac{1}{1 + \exp(\eta(N, D))} G(s_i|s_{1:i-1}).
\end{aligned}
\tag{6}
$$

Using $\sigma(.)$ to denote the sigmoid function, we have the final form from Eq. 3 as follows.

$$h^{\text{pred}}(\boldsymbol{s}_i|\boldsymbol{s}_{1:i-1}, S_{\mathcal{T}_{\text{train}}}) = \sigma(\eta(N, D))\, M(\boldsymbol{s}_i|\boldsymbol{s}_{1:i-1}) + (1 - \sigma(\eta(N, D)))\, G(\boldsymbol{s}_i|\boldsymbol{s}_{1:i-1}). \quad (7)$$

**Remark.** It is worth highlighting that $\eta(N, D)$ essentially serves the role of free-energy in the analysis above. Use of free-energy to model an interpolation between two states of a system is a common theoretical framework used in physics to study systems that undergo transitions between a disordered state to an ordered state: e.g., see in Landau theory, one considers interpolations between free energy at high temperature and low temperature to model continuous (second-order) phase transitions [98]. Our overall theoretical model, and the phenomenology it elicits, are very similar to models from physics, a parallel that we believe can be worth pursuing in future work, e.g., to uncover universality behavior beyond what we considered in this paper.

### D.3 Extending the Framework to Multiple Predictors

Our framework can be easily extended to multiple predictors. In the case of multiple predictors, the posterior predictive is written as:

$$h^{\text{pred}}(\boldsymbol{s}_i|\boldsymbol{s}_{1:i-1}, S_{\mathcal{T}_{\text{train}}}) = \sum_i P(Q_i|S_{\mathcal{T}_{\text{train}}}, T, H)\, Q_i(\boldsymbol{s}_i|\boldsymbol{s}_{1:i-1}).$$

To get the posterior, we first compute the unnormalized log-posterior (i.e., the logit) for each predictor $Q_i$:

$$\begin{aligned} \text{logit}(Q_i|S_{\mathcal{T}_{\text{train}}}, T, H) &= \log p(S_{\mathcal{T}_{\text{train}}}|Q_i) + \log p(Q|T, H) \\ &= -N_{\text{eff}}\, \overline{L}_Q(\mathcal{T}_{\text{train}}(D)) - K(Q)^\beta \quad \textbf{(By A1, A2)} \end{aligned}$$

Then, we simply apply softmax for normalization:

$$P(Q_i|S_{\mathcal{T}_{\text{train}}}, T, H) = \frac{\exp\left[\text{logit}(Q_i|S_{\mathcal{T}_{\text{train}}}, T, H)\right]}{\sum_j \exp\left[\text{logit}(Q_j|S_{\mathcal{T}_{\text{train}}}, T, H)\right]}$$

### D.4 Extending the Framework to Accommodate In-Context Strategy Selection

In most of our analyses, we assume that the posterior for a predictor $Q$ does not depend on the current sequence and only depends on sequences seen during training. Formally, $P(Q_i|S_{\mathcal{T}_{\text{train}}}, \boldsymbol{s}, T, H) = P(Q_i|S_{\mathcal{T}_{\text{train}}}, T, H)$. While we find that this assumption does not harm the predictive ability of our model for ID evaluation, it is known that Transformers perform in-context strategy selection [21, 99]. Furthermore, for OOD evaluation, we find that Transformers behave more like the generalizing solution (see Fig. 7), indicating in-context strategy selection, since the generalizing solution performs better OOD compared with the memorizing solution. Thus, to accommodate in-context strategy selection, we discard the assumption that the posterior only depends on sequences seen during training, and instead write it as:

$$P(Q|S_{\mathcal{T}_{\text{train}}}, \boldsymbol{s}, T, H) \propto P(S_{\mathcal{T}_{\text{train}}}|Q)\, P(\boldsymbol{s}|Q)\, P(Q|T, H)$$

Which, given predictors $M$ and $G$, yields the following form for the log-posterior odds:

$$\eta(N, D) := \log \underbrace{\frac{P(M|S_{\mathcal{T}_{\text{train}}}, \boldsymbol{s})}{P(G|S_{\mathcal{T}_{\text{train}}}, \boldsymbol{s})}}_{\text{Posterior odds}} = \log \underbrace{\frac{P(S_{\mathcal{T}_{\text{train}}}|M)}{P(S_{\mathcal{T}_{\text{train}}}|G)}}_{\text{Bayes factor (Pretraining)}} + \log \underbrace{\frac{P(\boldsymbol{s}|M)}{P(\boldsymbol{s}|G)}}_{\text{Bayes factor (Context)}} + \log \underbrace{\frac{P(M)}{P(G)}}_{\text{Prior odds}}.$$

Given previous work showing in-context learning updates tend to follow power laws such as those seen in pretraining [75, 97, 100], we can follow a similar argument to that detailed in D.1 to claim the effective context length scales sub-linearly as $|\boldsymbol{s}|_{\text{eff}} = \gamma'|s|^{1-\alpha'}$ for some $\gamma', \alpha'$. For simplicity, we assume here that $\gamma$ and $\alpha$ are shared between posterior updates in pretraining and during the context. Further research is required to understand the differences between parametric and in-context learning

updates [73]. Given that, and taking $N$ to be the number of total tokens seen by the Transformer, we write:

$$\eta(N, D) := \log \underbrace{\frac{P(M|S_{\mathcal{T}_{\text{train}}}, \boldsymbol{s})}{P(G|S_{\mathcal{T}_{\text{train}}}, \boldsymbol{s})}}_{\text{Posterior odds}}$$

$$= N_{\text{eff}}\, L_{\text{train}}(D) + |\boldsymbol{s}|_{\text{eff}}\, \Delta L_{\text{context}}(D) - \Delta K(D)^{\beta}$$

$$= \gamma N^{1-\alpha} \Delta L_{\text{train}}(D) + \gamma |\boldsymbol{s}|^{1-\alpha} \Delta L_{\text{context}}(D) - \Delta K(D)^{\beta}.$$

With $|\boldsymbol{s}|$ being the number of in-context examples in the context, and $\Delta L_{\text{context}}(D) := \overline{L}_G(\boldsymbol{s}) - \overline{L}_{M_D}(\boldsymbol{s})$ is the difference between the average loss of each predictor for the specific context. We note that this functional form is certainly idealized, as it assumes similar sample efficiency parameters for in-context learning and pretraining, and future research is needed to shed light on the similarities and differences of belief updating in-context vs. during training.

# E  Two-Hypotheses Threshold: Minimum amount of training to enable the Hierarchical Bayesian Model

One can reasonably expect our proposed Hierarchical Bayesian model to explain learning dynamics of in-context learning will not be predictive of a Transformer's behavior early-on in training, for otherwise we are saying even an untrained model perfectly generalizes. In actuality, the Transformer becomes amenable to approximation by our model after some minimal amount of training has occurred. To automatically calculate whether we have finished this regime of training, we calculate an "optimal" interpolation between the two predictors if they were capable of explaining the model behavior: specifically, we rely on the relative distance as an estimate of the optimal interpolation weighting, and use it as a baseline to compare model outputs with. In particular, we compute the loss between this optimal interpolation baseline and our trained Transformer model, and if this loss is below a threshold, we claim our theoretical model is applicable. We call this threshold the **two-hypotheses threshold** (see Fig. 6).

To define the **two-hypotheses threshold**, we make the observation that while the loss between Transformer and interpolating predictor can be large to begin with, it very quickly reduces to a small value. We can expect this latter regime is where our theoretical model is most likely to accurate at modeling the trained Transformer's behavior. Motivated by this, we heuristically choose the two-hypotheses threshold as a loss value 20% higher than minimum for Balls & Urns and Classification, and 10% higher than minimum for Linear Regression, on the scale defined from minimum to maximum loss (we find that a stricter threshold is required for linear regression to surpass the early high-loss regime, given the larger variance in interpolation loss values).

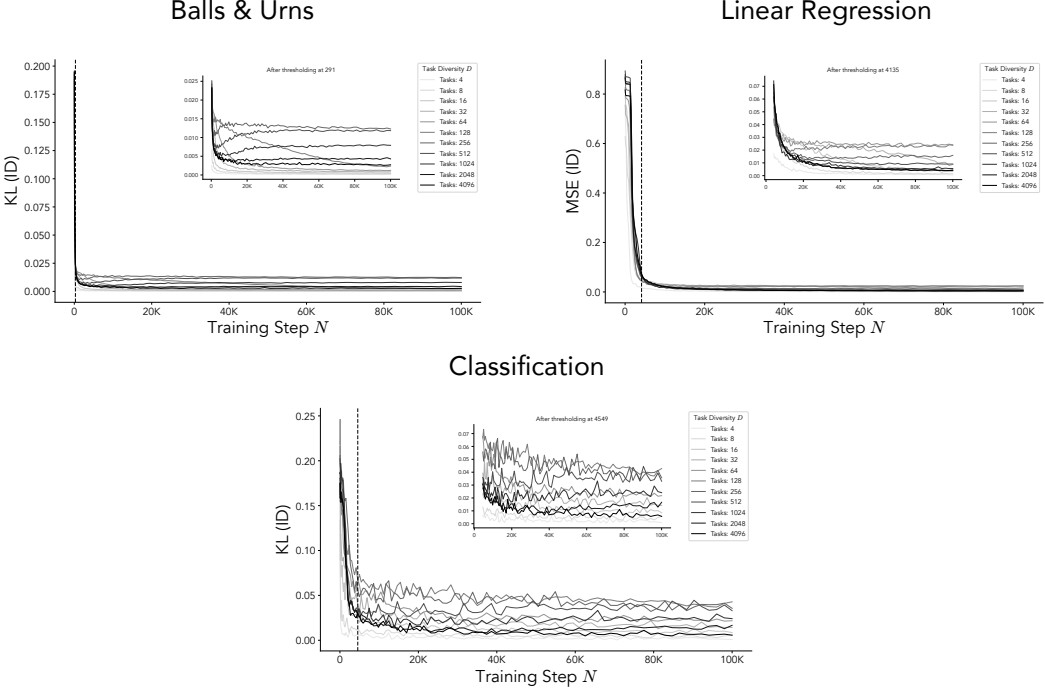

Figure 9: **Two-Hypotheses Threshold.** Defining an 'optimal' interpolation between the memorizing and generalizing predictors towards minimizing the Euclidean distance to the trained Transformer's predictions, we report the loss between this optimal interpolation and the Transformer's predictions. We observe a minimum amount of training is necessary for this loss to become sufficiently small such that our hierarchical Bayesian model, which implicitly assumes the Transformer can be functionally decomposed into the two predictors, will become applicable. The dotted lines demarcate this threshold, which we call the "two-hypotheses threshold". Mean-squared error (MSE) in the figure above is normalized by dimension. KL in this figure indicates forward KL from the Transformer's next token predictions to the interpolation of predictors.

# F Experimental Details

## F.1 Training and Model Details

**Model.** For all settings, we use the GPT-NeoX architecture sourced from Huggingface [56, 57]. While the number of layers / blocks in the model depend on the specific experimental setting (as reported below), we use only 1 attention head per layer and follow a sequential residual stream architecture across all settings.

**Training.** We use the Huggingface trainer with default parameters, changing only the learning rate, batch-size, total iterations, and warmup steps (reported below). Gradients are clipped to unit-norm. All models are trained on A100 GPUs, with maximum training budget reaching 2 days for all experiments encompassing the linear regression setting. We vary data-diversity $D$ from $\{2^2, 2^4, \ldots, 2^{12}\}$ across all settings.

**Settings-Specific Details.** For our three core settings, we report results covering the following hyperparameters. We note that similar to Carroll et al. [30], as we vary task-diversity $D$, we include tasks from the lower diversity-values in the setting involving the larger one—this allows us to assess effects of increasing diversity on the learning of a given task.

- **Balls and Urns.** Models of hidden dimension size $64$ are trained for 100K steps, with no warmup steps, at a constant learning rate of $5 \times 10^{-4}$ and batch-size of $64$. For our analysis, we derive experimental settings from combinations of task-dimensionality (equivalent to vocabulary-size), which varies in the set $\{8, 12, 16\}$; context length, which varies in the set $\{128, 256, 320\}$; and MLP expansion factor, which varies in the set $\{0.5, 4, 8\}$.
  - We conduct a separate experiment in which we attempt to elicit transience in higher task diversity settings ($D \in \{2^8, 2^9\}$). To do so on a reasonable compute budget (2M, 10M steps, respectively), we, similar to prior work [24, 29], have to intervene on the training pipeline. However, unlike prior work that often relies on weight decay for this purpose, we use learning rate annealing and find it to be sufficient. Specifically, we rely on an inverse square root schedule for decaying the learning rate with number of dimensions, context length, and MLP expansion fixed to 8, 128, and 4 respectively. We train up to $D = 2^7$ for 100K steps, and then train with $D = 2^8$ for 2M steps, and with $D = 2^9$ steps for 10M steps. Results of this experiment are shown in Fig. 6(c) and App. K.
- **Linear Regression.** Models of hidden dimension size $64$ are trained for 100K steps, with 5K warmup steps, at a constant learning rate of $5 \times 10^{-4}$ and batch-size of $128$. For our analysis, we derive experimental settings from combinations of task-dimensionality, which varies in the set $\{8, 12, 16\}$; context length, which varies in the set $\{16, 32, 64\}$; and MLP expansion factor, which varies in the set $\{0.5, 4, 8\}$. Like in Balls & Urns, we additionally train a setting using learning rate annealing (setting parameters are task dimensionality of 8, Context length of 16, and MLP expansion factor of 4. We train once with warmup of 500, and once with warmup of 5000. See results in app. K).
- **Classification.** Models of hidden dimension size $64$ are trained for 100K steps, with no warmup, at a constant learning rate of $5 \times 10^{-4}$ and batch-size of $64$. For our analysis, we derive experimental settings from combinations of task-dimensionality, which varies in the set $\{8, 16\}$; context length, which varies in the set $\{128, 256, 384\}$; and MLP expansion factor, which varies in the set $\{0.5, 4, 8\}$ for the 8 dimensions experiment, and is kept constant at 4 for the 16 dimensions experiment.

## F.2 Analysis Details

Next, we specify broad details of our analysis pipeline. These notes clarify design decisions made in our evaluation and motivations underlying them.

**General model and predictor evaluation.** For **OOD evaluation** of both the Transformer and our procedurally defined predictors, i.e., the memorizing predictor $M$ and generalizing predictor $G$, we draw 500 sequences from 500 *unseen* tasks (however, following still the same task distribution $\mathcal{T}_{\text{true}}$). In comparison, **ID evaluation** involves 500 sequences from *seen* tasks. If task-diversity $D$ is less than 500, sequences from the same task may be seen multiple times.

**Fig. 1 details.** In this figure, we use context length of 128, MLP expansion factor of 4, and task dimensionality of 8 for Balls & Urns. Linear Regression uses similar parameters, only with a context length of 32, and Classification uses similar parameters, only with MLP expansion factor of 0.5. In all plots displaying task diversity effects, we hold $N = 100$K. In the transience figures, we use $D = 256$ for Balls & Urns, $D = 64$ for Linear Regression, and $D = 512$ for Classification. For the comparison of relative distance maps, we show maps from the Balls & Urns setting with the parameters described above.

**Computing absolute distance between Transformer and predictors.** Given an input, we use both the Transformer model and the procedurally defined predictors to make next-token predictions. Then, we compare distance between these predictions using either the symmetrized KL (average of forward and backward KL) for the Balls and Urns and Classification settings, or the mean-squared error (MSE) for linear regression.

**Computing relative distance between Transformer and predictors.** Recall that relative distance, for a given distance measure $d(.,.)$ between two functions or distributions (symmetrized KL-divergence or Euclidean distance), is defined as $d_{\text{rel}} = (r+1)/2$, where $r := \frac{d(h,G)-d(h,M)}{d(G,M)}$ and $h(.)$ denotes the Transformer model trained from scratch. This metric implicitly makes the assumption that in some function space, the model $h(.)$ lies on a line between the predictors $M$ and $G$. Correspondingly, for the scenarios this assumption is violated, the value of $d_{\text{rel}}$ can go outside the range 0–1. This occurs relatively rarely, but nevertheless noticeably (e.g., if the model implements the optimal constant solution early on in training for linear regression). Accordingly, we clamp the metric between 0–1. For the multiple predictors extension of the model (App. H.5), we generalize the relative distance measure by fitting a convex combination of strategies to Transformer next-token predictions training and data diversity condition.

**Minimum amount of training to enable the Hierarchical Bayesian Model.** One can reasonably expect our proposed Hierarchical Bayesian model to explain learning dynamics of in-context learning will not be predictive of a Transformer's behavior early on in training, for otherwise, we are saying even an untrained model perfectly generalizes. In actuality, the Transformer becomes amenable to approximation by our model after some minimal amount of training has occurred. To automatically calculate whether we have finished this regime of training, we use the relative distance as an estimate of an "optimal" interpolation weight between the two predictors. We compute the loss between this optimal interpolation baseline and our trained Transformer model, and if this loss is below a threshold, we claim our theoretical model is applicable now (see details on our choice of threshold in App. E)

**Fitting the Bayesian Model.** We must perform the following three steps in order to fit our model.

- **Approximating Kolmogorov complexity.** Because true Kolmogorov complexity is not computable, we estimate an upper bound by compressing a self-contained bundle for each predictor: (i) the cleaned Python source that instantiates the predictor, and (ii) any numpy arrays it needs at inference time (e.g., the full table of urn distributions for the memorizing baseline in Balls & Urns). We remove comments, docstrings, and extraneous whitespace from the source code. For arrays, we first apply simple delta-encoding (store successive differences) to expose additional structure. The pre-processed bundle is compressed with four strong, off-the-shelf algorithms: `lzma` (preset=9 | PRESET_EXTREME), `bzip2` (level=9), `brotli` (quality=11, mode=TEXT), and `zstd` (level=22). We take the smallest compressed size (in bits) across the four algorithms as our estimate; this is a standard practice for obtaining a loose but practical upper bound. To keep estimates comparable, we exclude external libraries such as PyTorch from compression: all predictors call the same set of PyTorch primitives, so including them would add a large constant offset without altering relative complexities. This choice does, however, ignore the fact that some primitives might be cognitively "cheaper" for a Transformer to implement than others—an important caveat for future work.

- **Computation of average log likelihood per predictor.** For every experimental condition we first compute the token-level log-likelihood that each predictor assigns to the in-distribution sequences. Because the irreducible error term cancels when models are compared, we use KL-based evaluations to the Balls & Urns and classification tasks, treating the linear-regression setting separately. To summarize performance, we need the mean log-likelihood per token, yet

the empirical loss distribution is, at times, quite skewed: most tokens later in the context incur near-zero loss, whereas a small fraction of early tokens produce large spikes. Therefore, a naive arithmetic mean converges slowly and exhibits high variance. Thus, we instead use median of means, an estimator for the true mean that has better convergence under long-tailed distributions.

- **How fitting is done.** To fit the 3 free parameters of the Bayesian model, we minimize the mean KL divergence (or mean-squared error in the linear-regression setting) between the interpolated predictions and the Transformer outputs. Optimization is performed with `scipy.optimize.minimize` using the L-BFGS-B algorithm, capped at 1K iterations and 2K function evaluations, with gradient and function tolerances of $10^{-7}$. Exact gradients are supplied via PyTorch's automatic differentiation, ensuring stable convergence. For each task we fit on $80\%$ of the $(N, D)$ configuration grid and reserve the remaining $20\%$ for held-out validation and diagnostic checks. The process of fitting and evaluation takes around a minute or less to complete. In the case of the in-context strategy selection, we find that we require a parameter search using basin hopping with 50 iterations before beginning optimization. We optimize using the 5 best candidate initialization parameters found via basin hopping, and take the best result after optimization.

**Novel Predictions Analysis Details.** To show sub-linear sample efficiency and a sigmoidal curve in $N^{1-\alpha}$, we fit a parameterized logistic $\frac{a}{1+\exp(-b(N^{1-\alpha}-N_0))}$ with free parameters $a, b, N_0$ to each training run (constant $D$ value), via scipy curvefit function. We use the $\alpha$ value given by the Bayesian model. Curve fits are shown in Fig. 6(a). To compute the second derivative of the relative distance (Fig. 6), we simply use parameters for the logistic fits described above (which provide very close fits, as can be seen in Fig. 6(a)), then compute the second derivative based on the form for the second derivative of a parametrized logistic. To plot the vector field, we normalize both $U$ and $V$ directions by the larger value among the 90th percentile values for $U$ and $V$.

## G  Additional Details Regarding Settings and Predictors

We now give a more detailed discussion of the different settings analyzed in this work: (i) Balls & Urns, (ii) Linear Regression, and (iii) Classification. We also provide details of how the memorizing and generalizing predictors are implemented for these settings. Broadly, as also visualized in Fig. 10, all settings involve learning of a mixture of tasks $\mathcal{T}_{\text{train}}$ drawn from the true task distribution $\mathcal{T}_{\text{true}}$. The number of tasks involved in the mixture is called its task diversity (denoted $D$). For all settings, we find models learn predictors of two types: a *memorizing predictor*, which corresponds to the Bayesian posterior predictive distribution with a discrete prior over seen tasks $\mathcal{T}_{\text{train}}$, and a *generalizing predictor*, which corresponds to the Bayesian posterior predictive distribution with a prior over the true task distribution $\mathcal{T}_{\text{true}}$. The precise forms of these predictors, as well as how sequences are assembled into training batches in each setting, are provided in the following sections.

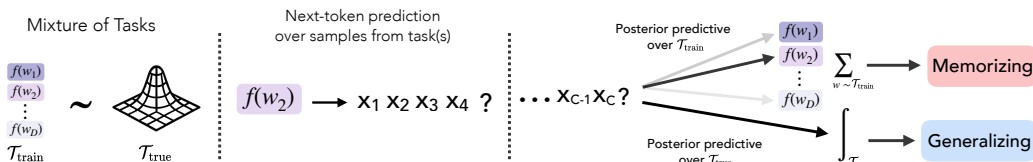

Figure 10: **General Abstraction Capturing our Experimental Settings and their Predictors.** Each setting involves a mixture of parameterized functions (called a "task"), with $D$ functions (the "task diversity"). Task consist of predicting the next element in a sequence, and vary based on whether models are trained in a standard auto-regressive fashion (like Balls & Urns) or whether they are only trained to predict some elements in the sequence (only function outputs in Linear Regression, and only the last label in Classification). Across settings, the solutions learned by Transformers can be characterized as *memorizing predictors* or *generalizing predictors*. A memorizing predictor is defined as the Bayesian posterior predictive distribution with $\mathcal{T}_{\text{train}}$, the distribution of seen tasks, as its prior. A generalizing predictor is defined as the Bayesian posterior predictive with the true task distribution $\mathcal{T}_{\text{true}}$ as its prior.

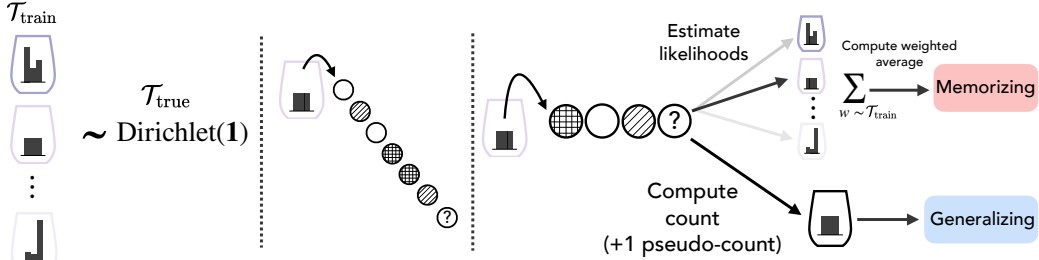

Figure 11: **Visualizing the setup for Balls and Urns.** Each task involves an "urn" that outputs a "ball" of a specific type every time it is sampled from. The task then involves seeing samples from an urn, concatenated to form a sequence. A memorizing predictor for this setting involves computing the sequence-level unigram statistics, i.e., the counts for each ball type, and comparing them with distributions from urns seen during training. Meanwhile, a generalizing predictor simply assumes the distribution of balls follows a uniform Dirichlet prior, thus predicting simply based on computing the unigram statistics from the sequence and adding a 1 pseudo-count for each ball type. Thus, this predictor generalizes to novel urns not seen by the model during training.

## G.1 Balls and urns

**Memorizing Predictor.** The memorizing predictor perform a Bayesian averaging operation and requires computing a weighted average of all urn distributions seen during training. The weight on each urn is derived from the likelihood of the current sequence of observations being generated by that urn. Formally, let $\boldsymbol{w}_d$ denote the parameters for an urn $d \in \{1, \ldots, D\}$, with each element $\boldsymbol{w}_d^{(k)}$ containing the probability for ball type $k \in \{1, \ldots, m\}$ under urn $D$. Then, the probability of a new ball being of type $k$ after seeing a sequence $\boldsymbol{s}$ is:

$$
p(k|\boldsymbol{s}) \propto \sum_{\boldsymbol{w}_d \in \mathcal{T}_{\text{train}}} \boldsymbol{w}_d^{(k)} \prod_{k' \in \{1,\ldots,m\}} (\boldsymbol{w}_d^{(k')})^{n_{k'}}.
$$

With $n_{k'}$ being the number of occurrences of ball of type $k'$ in the sequence.

**Generalizing Predictor.** Given that the true distribution is a uniform Dirichlet, and that the Dirichlet distribution is a conjugate prior of the categorical distribution (from which we draw our samples), the optimal way to estimate the probability of a ball of a particular type $k$ in a sequence $\boldsymbol{s}$ of length $C$ is: $p(k|\boldsymbol{s}) = \frac{n_k+1}{C+m}$, with $n_k$ being the number of occurrences of ball of type $k$ in the sequence. That is, the optimal strategy for the true distribution is simply computing a count for each type, adding a 1 pseudo-count, and dividing by the sequence length.

## G.2 Linear regression

**Additional Details Not Provided in Main Text.** To maintain a constant signal-to-noise ratio across tasks with different dimensionality $m$, we set $\epsilon_i \sim \mathcal{N}(0, \sigma^2)$, with $\sigma^2 = \frac{m}{256}$.

**Generalizing Predictor.** The *generalizing predictor* in this case simply performs ridge regression. Given $\boldsymbol{x} = (\boldsymbol{x}_1^\intercal, ..., \boldsymbol{x}_{C-1}^\intercal)$ and $\mathsf{y} = (\mathsf{y}_1, ..., \mathsf{y}_{C-1})$, the weight estimate is after seeing $C-1$ examples is:

$$
\hat{\boldsymbol{w}}_G^{(C)} = (\boldsymbol{x}^\intercal \boldsymbol{x} + \sigma^2 \boldsymbol{I}_m)^{-1} \boldsymbol{x}^\intercal \mathsf{y}
$$

**Memorizing Predictor.** The *memorizing predictor* in this case performs inference by Bayesian averaging: a weighted average across all $\boldsymbol{w}^{(d)}$s seen in the training distribution, with weights determined by the likelihood that the sequence was generated by the specific $\boldsymbol{w}^{(t)}$. After seeing $C-1$

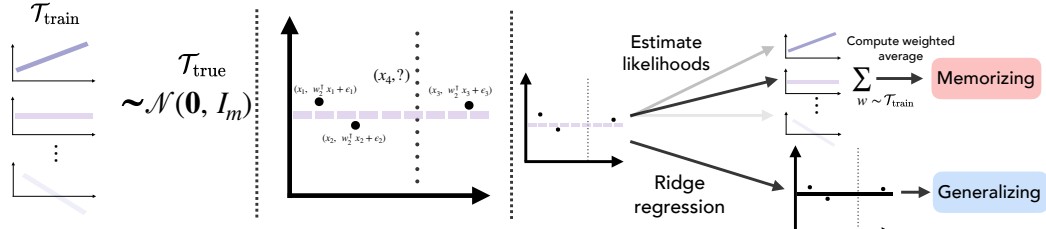

Figure 12: **Visualizing the setup for Linear Regression.** Each task involves a linear regression problem, defined by parameters $w$, that outputs a pair $(x, y)$, where $y = w^\intercal x + \epsilon$ is a noisy linear transformation of the vector $x$. The task then involves seeing a sequence of such pairs, concatenated to form a sequence. A memorizing predictor for this setting involves computing the likelihood of the pairs seen in context under the parameters of each task seen during training, using this result to compute a posterior over said tasks and a posterior-weighted average with a discrete prior over seen tasks. Meanwhile, the generalizing predictor is merely the ridge regression operation, which is equivalent to performing a Bayesian average operation assuming a continuous Gaussian prior. Correspondingly, this predictor generalizes to novel regression tasks that were not seen by the model during training.

examples, the weight estimate is:

$$\hat{w}_M^{(C)} = \sum_{w_d \in \mathcal{T}_{\text{train}}} \frac{\exp(-\frac{1}{2\sigma^2} \sum_{c=1}^{C-1} (y_c - w_d^\intercal \, x_c)^2)}{\sum_{w_{d'} \in \mathcal{T}_{\text{train}}} \exp(-\frac{1}{2\sigma^2} \sum_{c=1}^{C-1} (y_c - w_{d'}^\intercal \, x_c)^2)} w_d$$

### G.3 Classification

We use the classification setting with the formulation from [29] as well as inspiration from the noisy class centroids introduced by [28]. As Nguyen and Reddy [29] have shown, their simplified setting captures the phenomenology of other classification settings proposed by Chan et al. [20], Reddy [28]. For simplicity, we include only binary labels in our version.

**Additional Details Not Provided in Main Text.** When presented in context, items $w$ are noised and presented as $\tilde{w} = \frac{w + \sigma\epsilon}{\sqrt{1+\sigma^2}}$. We use within-class variance of $\sigma^2 = 0.5$ in all settings, and $\epsilon \in \mathcal{N}(0, I_m/m)$ is sampled separately for each item in the context.

**Memorizing Predictor.** The *memorizing predictor* in this setting performs inference by computing a posterior-weighted average over item-label pairs seen in the training distribution $w \oplus l \sim \mathcal{T}_{\text{train}}$. The posterior depends on the similarity of the latent $w$ to the query $\tilde{w}_{\text{query}}$ as well as to each context item $\tilde{w}_j$, given that the model assumes the generative process in which the query is taken from the same latent as one of the context items. The form for a noisy item used for defining the input sequence is $\tilde{w} = \frac{w + \sigma\epsilon}{\sqrt{1+\sigma^2}}$ for some $w \sim \mathcal{T}_{\text{train}}$. Thus, since $\epsilon$ has covariance $I_m/m$, we can write a noisy item sampled from a given $w$ will be distributed as: $(\tilde{w} \mid w = w_d) \sim \mathcal{N}(\frac{1}{\sqrt{1+\sigma^2}} w, \frac{\sigma^2}{1+\sigma^2} I_m/m)$. We then define $\mathcal{L}(\tilde{w}|w) = \exp\left(-\frac{m}{2\sigma^2}(1+\sigma^2) \left\| \tilde{w} - \frac{1}{\sqrt{1+\sigma^2}} w \right\|^2\right)$ as a value proportional to the likelihood of this distribution (disregarding constants). Then, the probability of the query label being 1 is:

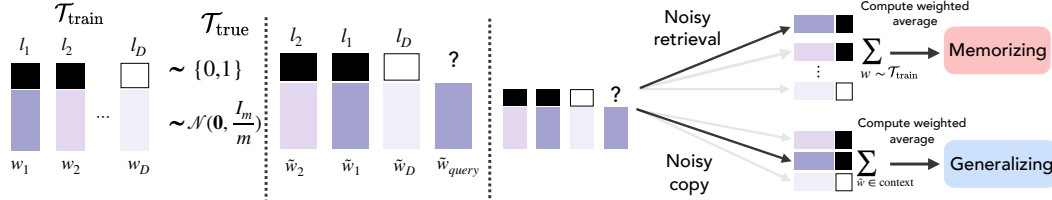

Figure 13: **Visualizing the setup for Classification.** Each task involves noisy item-label pairs $\tilde{\boldsymbol{w}} \oplus \boldsymbol{l}$, and ends with a noisy query item $\tilde{\boldsymbol{w}}_{\text{query}}$ which comes from the same true item $\boldsymbol{w}$ as one of the items in the sequence. Items are noised via $\tilde{\boldsymbol{w}} = \frac{\boldsymbol{w}+\sigma\epsilon}{\sqrt{1+\sigma^2}}$, with $\epsilon \in \mathcal{N}(0, \boldsymbol{I}_m/m)$ sampled from the same distribution as the true item $\boldsymbol{w}$. The memorizing predictor stores the true latent items of the training distribution and computes a posterior-weighted average using the labels for each true item. The posterior for a latent is determined by its similarity to the noisy query item, as well as its similarity to each context item, given that the generative process dictates that the query comes from the same latent as one of the context items. Therefore, this predictor implements a context-constrained 'in-weights learning' solution, discussed by Singh et al. [33]. In contrast, the generalizing predictor implements a noisy copy operation. It estimates the likelihood that the query head and each item seen in context come from the same true item. Then, it predicts via a posterior-weighted average according to the labels of each item seen in context. Therefore, this predictor works for OOD settings containing novel items that were not previously seen during training.

$$p(1|\boldsymbol{s}) = \sum_{\boldsymbol{w}_d \in \mathcal{T}_{\text{train}}} p(\boldsymbol{w}_d | \tilde{\boldsymbol{w}}_{\text{query}}, \{\tilde{\boldsymbol{w}}_1 \oplus \boldsymbol{l}_1, ..., \tilde{\boldsymbol{w}}_{C-1} \oplus \boldsymbol{l}_{C-1}\}) \, p(1|\boldsymbol{w}_d)$$

$$\propto \sum_{\boldsymbol{w}_d \in \mathcal{T}_{\text{train}}} \left[ p(\tilde{\boldsymbol{w}}_{\text{query}} | \boldsymbol{w}_d) \left( \sum_{\tilde{\boldsymbol{w}}_j \in \text{Context}} p(\tilde{\boldsymbol{w}}_j | \boldsymbol{w}_d) \mathbb{1}(\boldsymbol{l}_j = \boldsymbol{l}_d) \right) \mathbb{1}(\boldsymbol{l}_d = 1) \right]$$

$$\propto \sum_{\boldsymbol{w}_d \in \mathcal{T}_{\text{train}}} \left[ \mathcal{L}(\tilde{\boldsymbol{w}}_{\text{query}} | \boldsymbol{w}_d) \left( \sum_{\tilde{\boldsymbol{w}}_j \in \text{Context}} \frac{\mathcal{L}(\tilde{\boldsymbol{w}}_j | \boldsymbol{w}_d)}{\sum_{w_d \in \mathcal{T}_{\text{train}}} \mathcal{L}(\tilde{\boldsymbol{w}}_j | \boldsymbol{w}_d)} \mathbb{1}(\boldsymbol{l}_j = \boldsymbol{l}_d) \right) \mathbb{1}(\boldsymbol{l}_d = 1) \right]$$

**Generalizing Predictor.** The *generalizing predictor* in this setting performs inference by computing a posterior-weighted average over the noisy item-label pairs seen in the context.

This predictor assumes access to the true generative process for defining the context, in which items $\{\boldsymbol{w}_1, ..., \boldsymbol{w}_{C-1}\}$ are randomly sampled from $\mathcal{T}_{\text{true}}$, with one item chosen at random to be the query $\boldsymbol{w}_{\text{query}}$. Finally, all items are independently noised via $\tilde{\boldsymbol{w}} = \frac{\boldsymbol{w}+\sigma\epsilon}{\sqrt{1+\sigma^2}}$.

Because $\boldsymbol{w}_{\text{query}}$ is sampled from the specific set of latent items $\{\boldsymbol{w}_1, \ldots, \boldsymbol{w}_{C-1}\}$ underlying the context rather than drawn anew from the continuous distribution $\mathcal{T}_{\text{true}}$, the prior over $\boldsymbol{w}_{\text{query}}$ collapses to a uniform discrete distribution over these unobserved context latents. Since the model cannot access these true latents directly, it computes the conditional probability of the query given each noisy context item $\tilde{\boldsymbol{w}}_j$—treating $\tilde{\boldsymbol{w}}_{\text{query}}$ and $\tilde{\boldsymbol{w}}_j$ as noisy observations of the same latent parent.

The conditional distribution of the query item given the context item is Gaussian:
$$p(\tilde{\boldsymbol{w}}_{\text{query}} | \tilde{\boldsymbol{w}}_j) = \mathcal{N}\left(\tilde{\boldsymbol{w}}_{\text{query}}; \mu_G, \Sigma_G\right)$$
Using the standard Gaussian conditional formulas we get:
$$\mu_G = \frac{1}{1+\sigma^2} \tilde{\boldsymbol{w}}_j, \quad \Sigma_G^{-1} \propto \frac{(1+\sigma^2)^2}{\sigma^2(2+\sigma^2)} I_m$$
Disregarding normalization constants, the probability of the query label being 1 is:
$$p(1|s) \propto \sum_{\tilde{\boldsymbol{w}}_j \in \text{Context}} p(\tilde{\boldsymbol{w}}_{query} | \tilde{\boldsymbol{w}}_j) \cdot p(1|\tilde{\boldsymbol{w}}_j)$$

$$\propto \sum_{\tilde{\boldsymbol{w}}_j \in \text{Context}} \exp\left( -\frac{m(1+\sigma^2)^2}{2\sigma^2(2+\sigma^2)} \left\| \tilde{\boldsymbol{w}}_{\text{query}} - \frac{\tilde{\boldsymbol{w}}_j}{1+\sigma^2} \right\|^2 \right) \mathbb{1}(\boldsymbol{l}_j = 1).$$

# H   Main Results

In the following sections, we provide the results reported in the main paper across all settings and experiments.

## H.1   Task Diversity Effects

We find task diversity effects [25] to be very robust across settings and experimental conditions. More specifically, we consistently find that increasing task diversity yields a transition in Transformer behavior from behaving like a memorizing predictor to behaving like a generalizing predictor. In the following, we present evidence of this phenomenon for ID sequences. See results in following pages.

### H.1.1 Balls & Urns

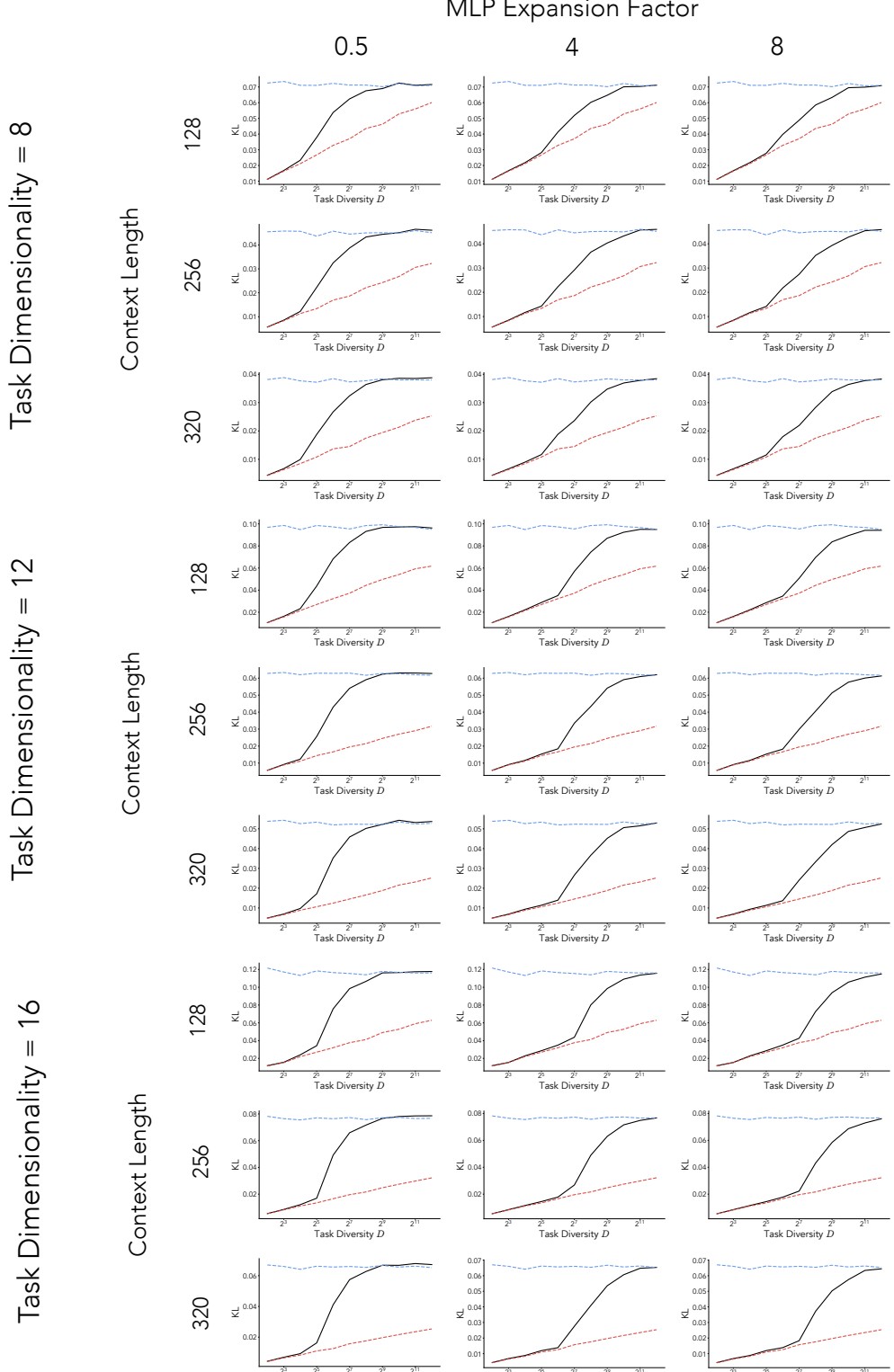

Figure 14: **Task Diversity Effects Across Balls & Urns Conditions.** Red dashed line indicates the memorizing solution $M$, blue dashed line indicates the generalizing solution $G$, and black solid line indicates Transformer behavior at the end of training (100K steps).

### H.1.2 Linear Regression

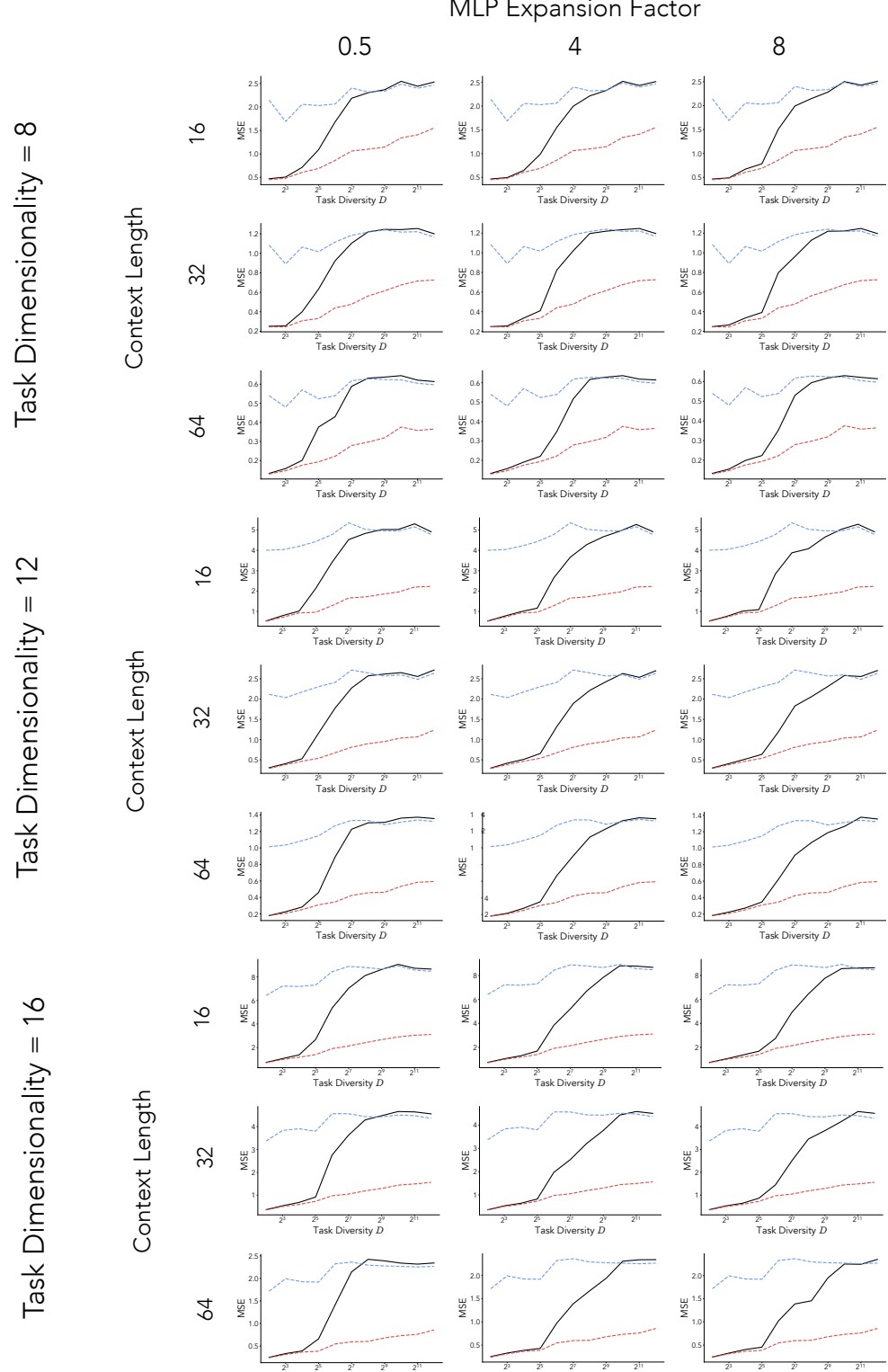

Figure 15: **Task Diversity Effects Across Linear Regression Conditions.** Red dashed line indicates the memorizing solution $M$, blue dashed line indicates the generalizing solution $G$, and black solid line indicates Transformer behavior at the end of training (100K steps).

### H.1.3 Classification

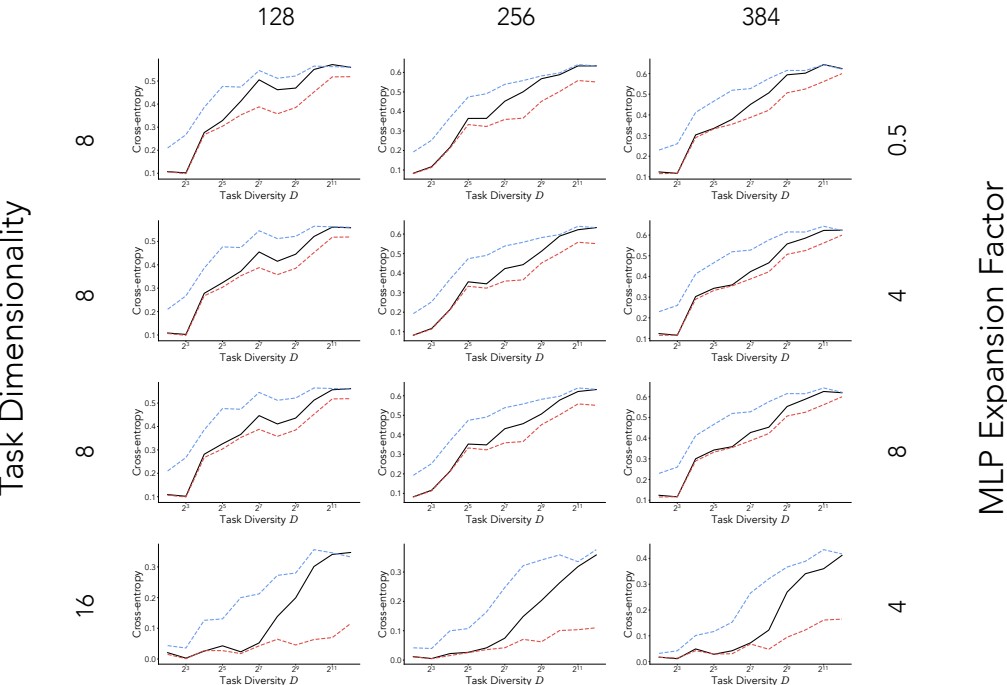

Figure 16: **Task Diversity Effects Across Classification Conditions.** Red dashed line indicates the memorizing solution $M$, blue dashed line indicates the generalizing solution $G$, and black solid line indicates Transformer behavior at the end of training (100K steps).

### H.2 Transience

Across settings and conditions, we also find the phenomenon of transience [24] to be consistent in moderate task diversity values. More specifically, in moderate task diversity values, we see the Transformer approach the generalizing solution in terms of OOD performance early in training, only to eventually begin memorizing and worsen in OOD performance. In the figures below, we show OOD performance of Transformers trained in different task diversity conditions, with low task diversity values showing immediate memorization, moderate task diversity values showing transience, and high task diversity values often continuing to generalize well throughout training. See results in following pages.

### H.2.1 Balls & Urns

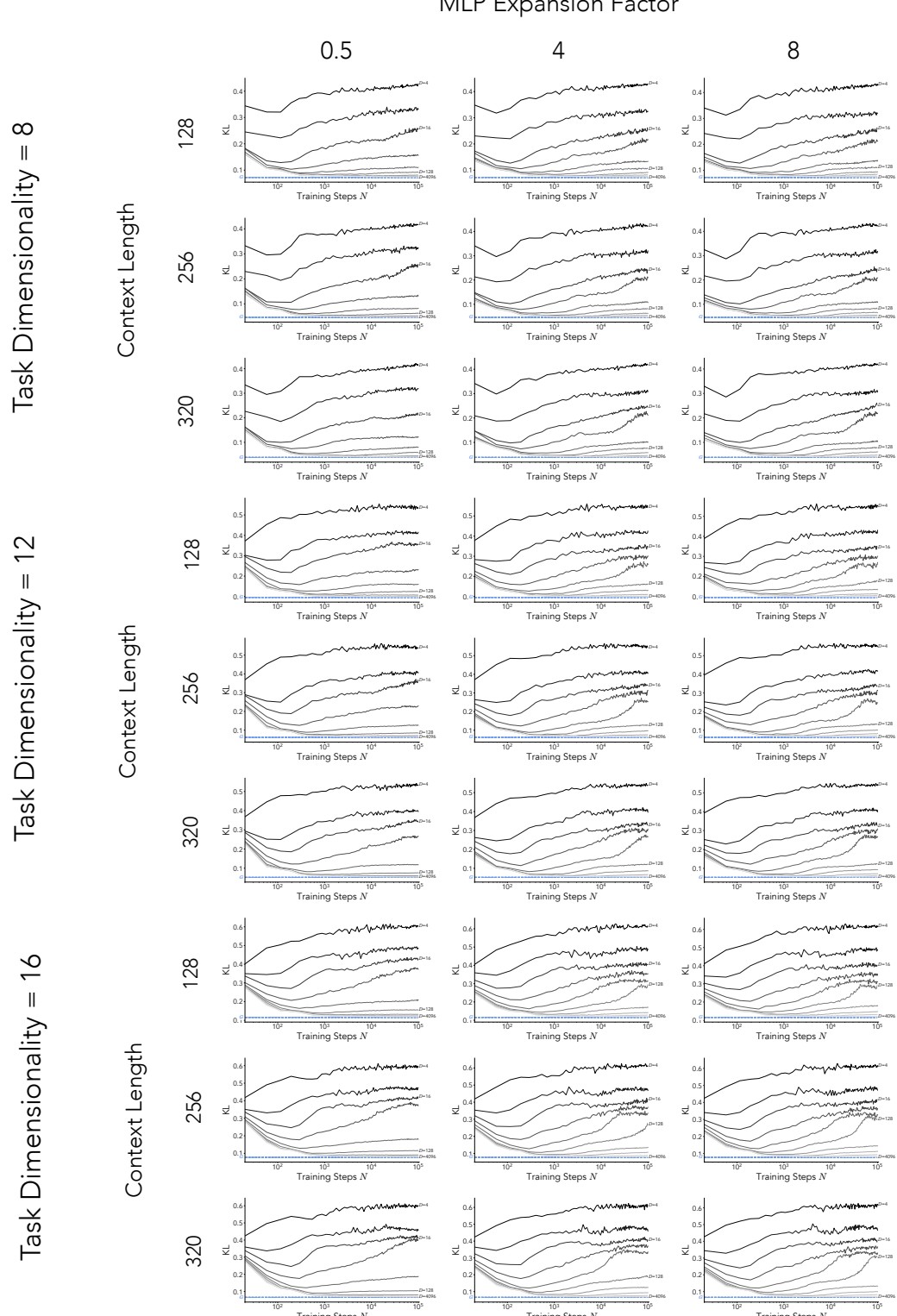

Figure 17: **Transience Across Balls & Urns Conditions.** OOD performance presented. Blue Dashed line indicates OOD performance of generalizing solution $G$.

### H.2.2 Linear Regression

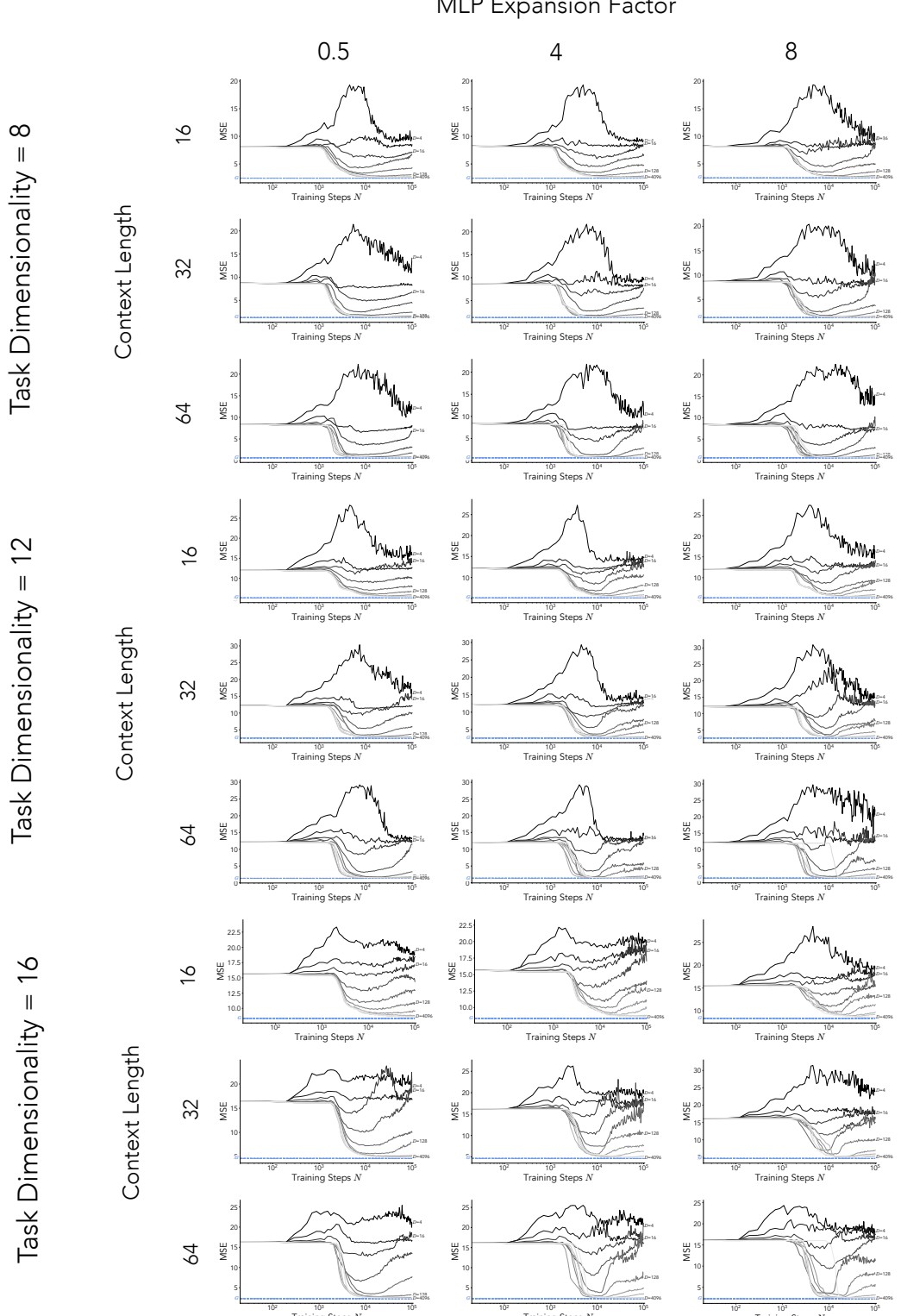

Figure 18: **Transience Across Linear Regression Conditions.** OOD performance presented. Blue Dashed line indicates OOD performance of generalizing solution $G$.

**H.2.3 Classification**

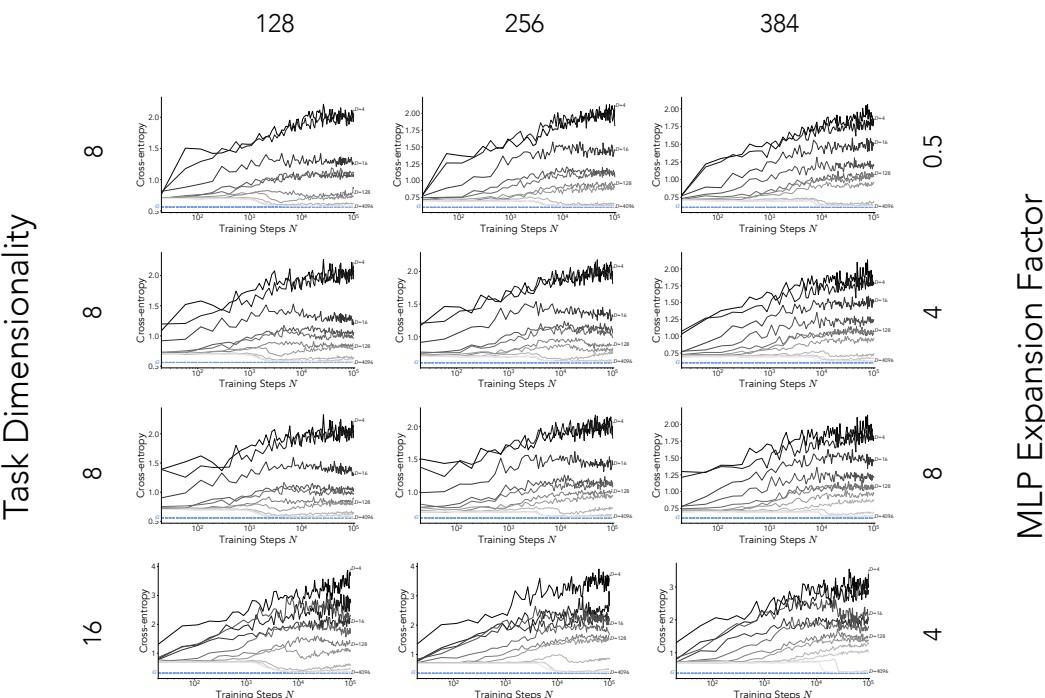

Figure 19: **Transience Across Classification Conditions.** OOD performance presented. Blue Dashed line indicates OOD performance of generalizing solution $G$. Note that in the case of classification, it is often the case that only one or two task diversity conditions show transient generalization, as can be seen more clearly from the absolute distance maps in the next section.

**H.3 Absolute Distance from Predictors**

We find that across settings and conditions, Transformers primarily learn and transition between behaving like two predictors: a generalizing solution $G$, which consists of the Bayesian posterior predictive distribution over the true task distribution $\mathcal{T}_{\text{true}}$ and a memorizing solution $M$ which consists of the Bayesian posterior predictive distribution over the training task distribution $\mathcal{T}_{\text{train}}$. In the figures below, we display the absolute distance from each of these predictors, as well the relative distance in the background (ranging from red indicating closeness to $M$ to blue indicating closeness to $G$). See results in following pages.

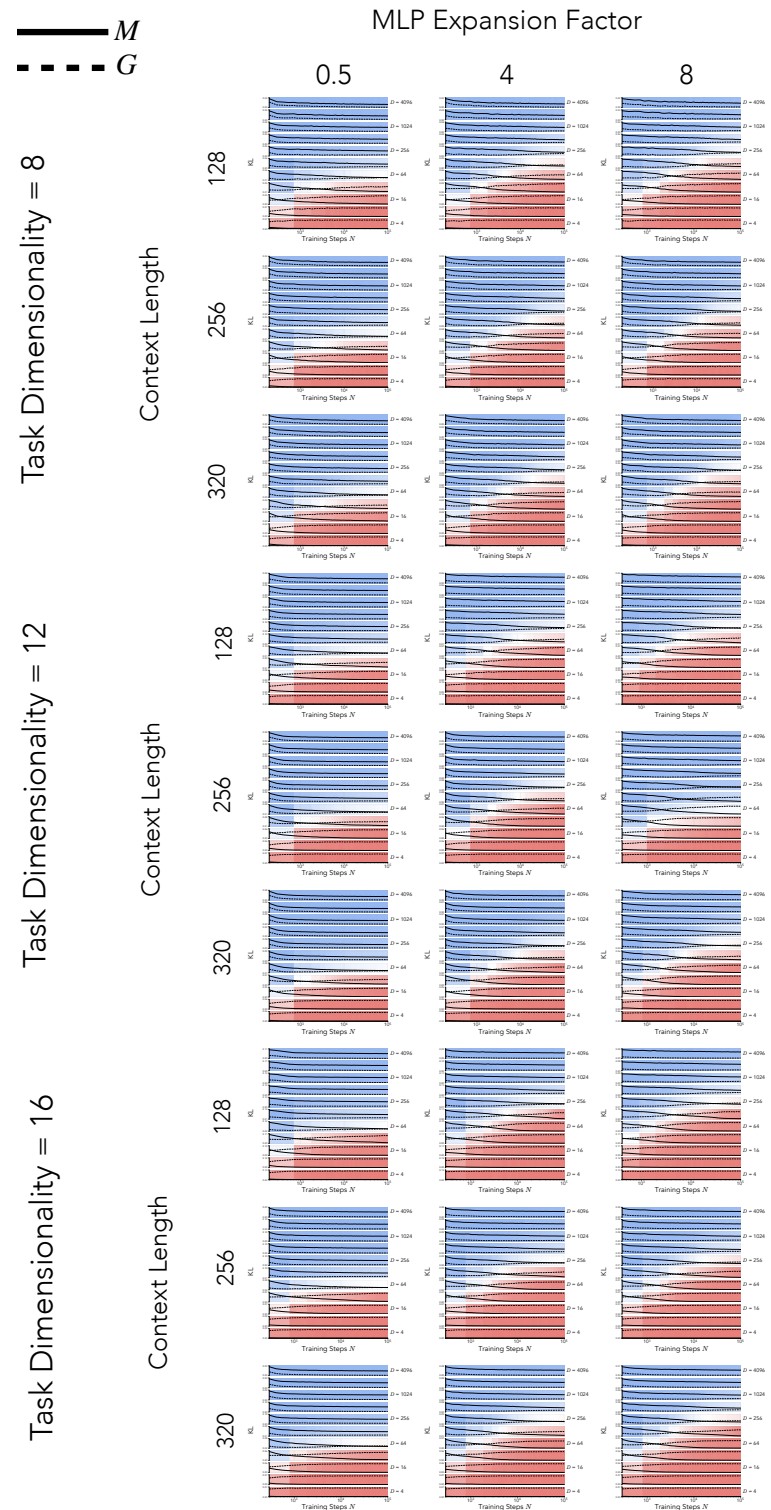

Figure 20: **Absolute and Relative Distance from Predictors Across Balls & Urns Conditions.** Distance from the generalizing solution $G$ shown in the dashed black line, while distance from the memorizing solution $M$ is shown in the solid line. KL indicates symmetrized KL divergence (average of forward and backward KL).

### H.3.2 Linear Regression

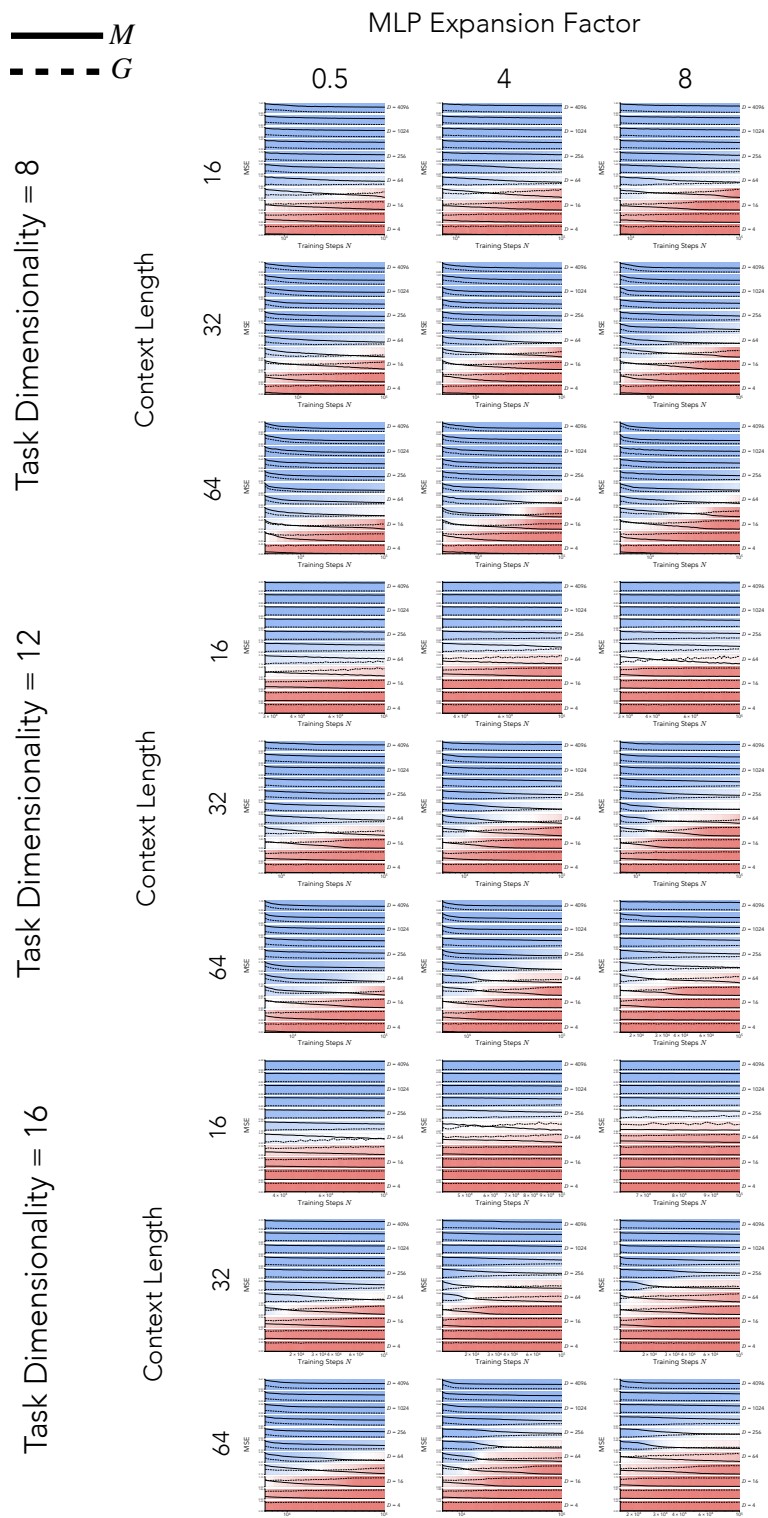

Figure 21: **Absolute and Relative Distance from Predictors Across Linear Regression Conditions.** Distance from the generalizing solution $G$ shown in the dashed black line, while distance from the memorizing solution $M$ is shown in the solid line.

### H.3.3 Classification

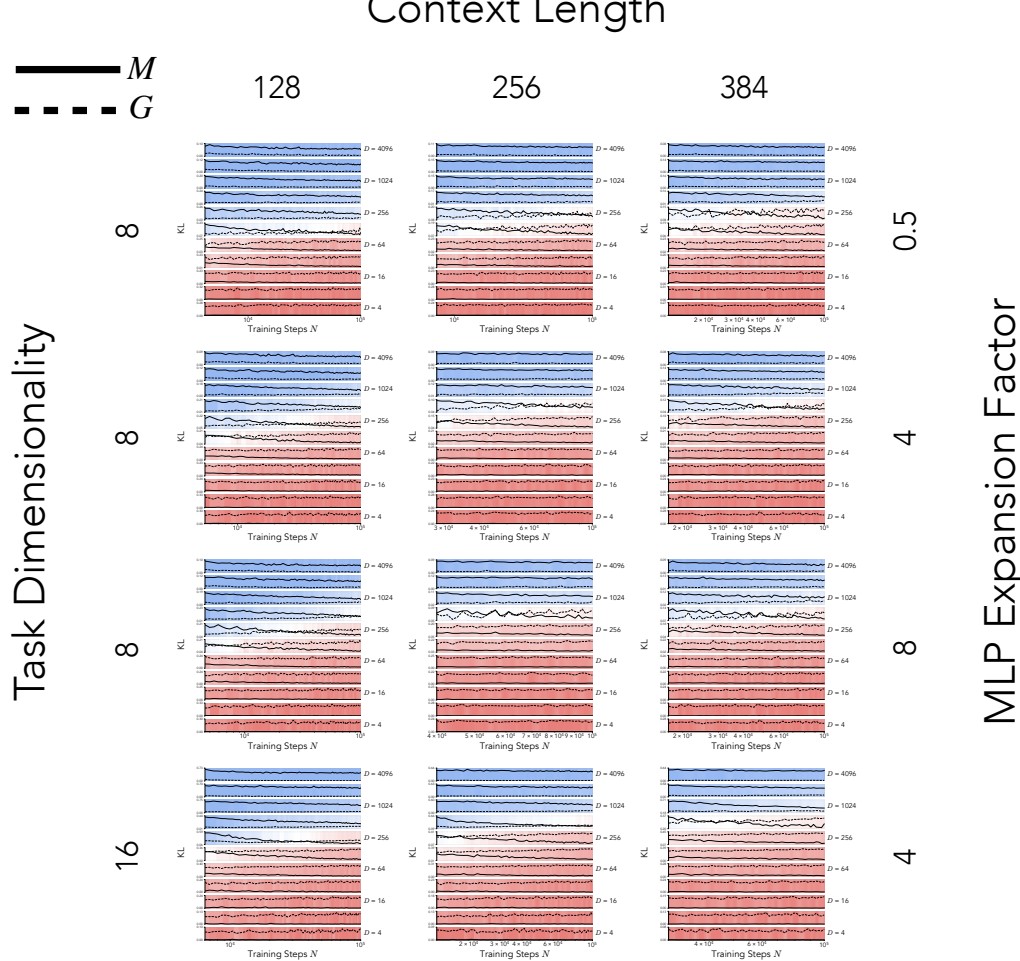

Figure 22: **Absolute and Relative Distance from Predictors Across Classification Conditions.** Distance from the generalizing solution $G$ shown in the dashed black line, while distance from the memorizing solution $M$ is shown in the solid line. KL indicates symmetrized KL divergence (average of forward and backward KL).

### H.4 Model Predictions

Across settings and training conditions, we find that our model's predictions consistently perform well both in estimating the next-token prediction behavior of the Transformer, as well as capturing its change in generalization behavior across conditions, as displayed by the relative distance from predictors $G$ and $M$. We conduct thorough stress-testing of our model across 3 settings and 72 $(N, D)$ maps, each containing 11 different training runs, and find that our model consistently performs well across maps, thus providing robust evidence for the predictive and explanatory power of our account. See results in following pages.

### H.4.1 Balls & Urns

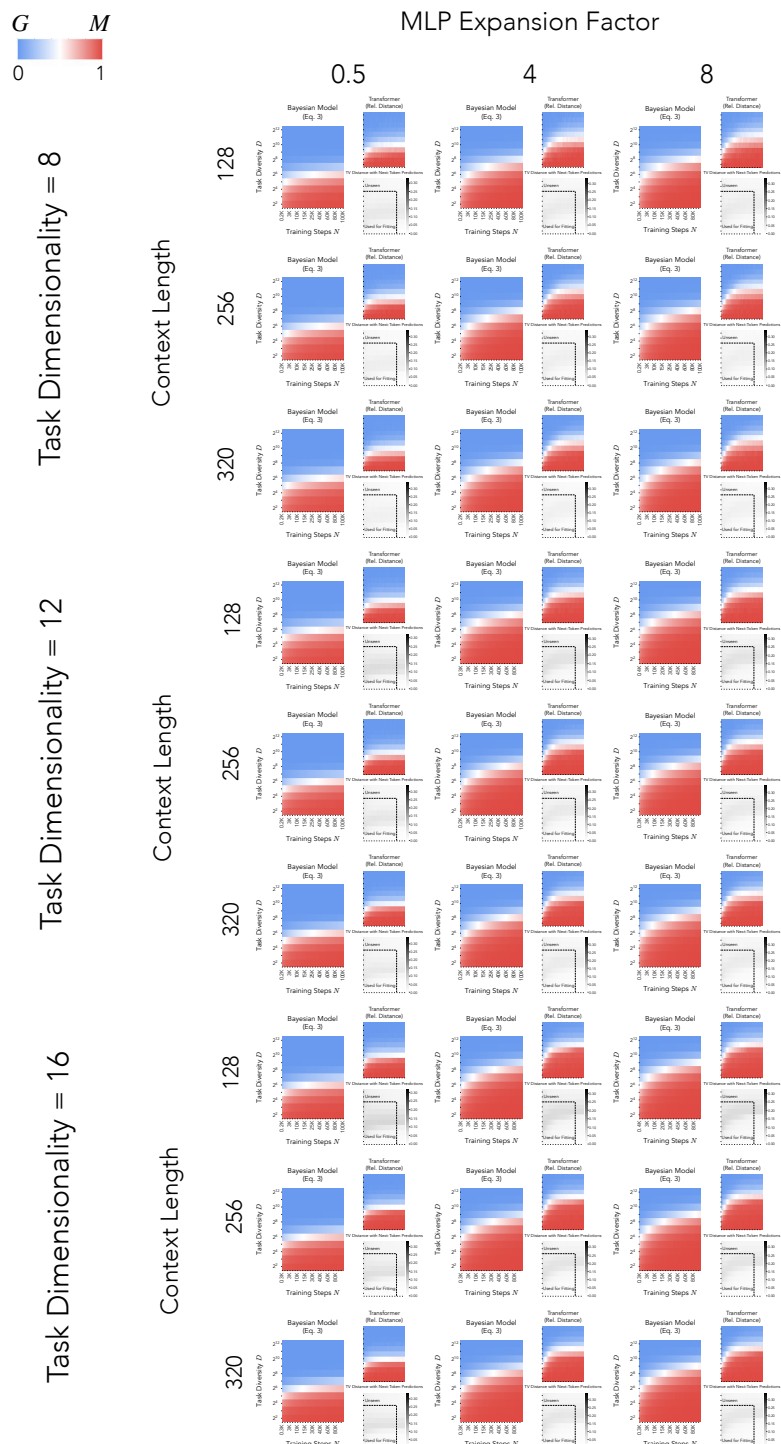

Figure 23: **Bayesian Model Predictions Across Balls & Urns Conditions.** Red indicates closeness to memorizing predictor $M$, while blue indicates closeness to generalizing predictor $G$. Shown is a comparison between the posterior probability of the memorizing solution $M$ given by our Bayesian model (left) and the relative distance from the Transformer (top right), as well as heatmaps indicating similarity with Transformer next-token predictions (bottom right). Max color bar value is determined by the performance of a baseline predictor that always outputs the mean of the distribution $\mathcal{T}_{\text{true}}$.

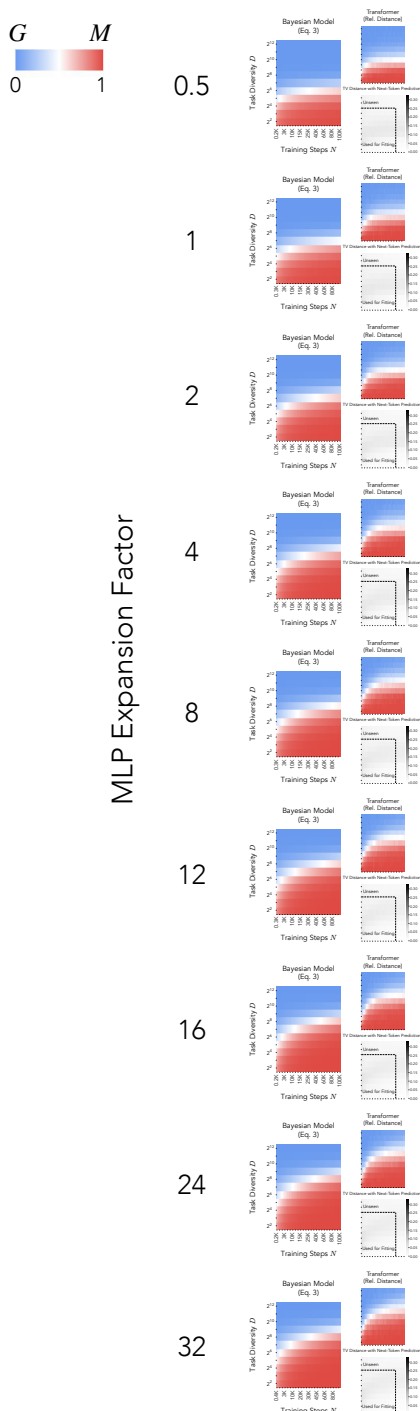

Figure 24: **Bayesian Model Predictions Across Balls & Urns Conditions with Varying MLP Expansion Factors.** Red indicates closeness to memorizing predictor $M$, while blue indicates closeness to generalizing predictor $G$. Shown is a comparison between the posterior probability of the memorizing solution $M$ given by our Bayesian model (left) and the relative distance from the Transformer (top right), as well as heatmaps indicating similarity with Transformer next-token predictions (bottom right). Context length is 128, task dimensionality is 8, and hidden size is 64 in all conditions shown. MLP width is given by hidden size times MLP expansion factor. Max color bar value is determined by the performance of a baseline predictor that always outputs the mean of the distribution $\mathcal{T}_{\text{true}}$.

### H.4.2 Linear Regression

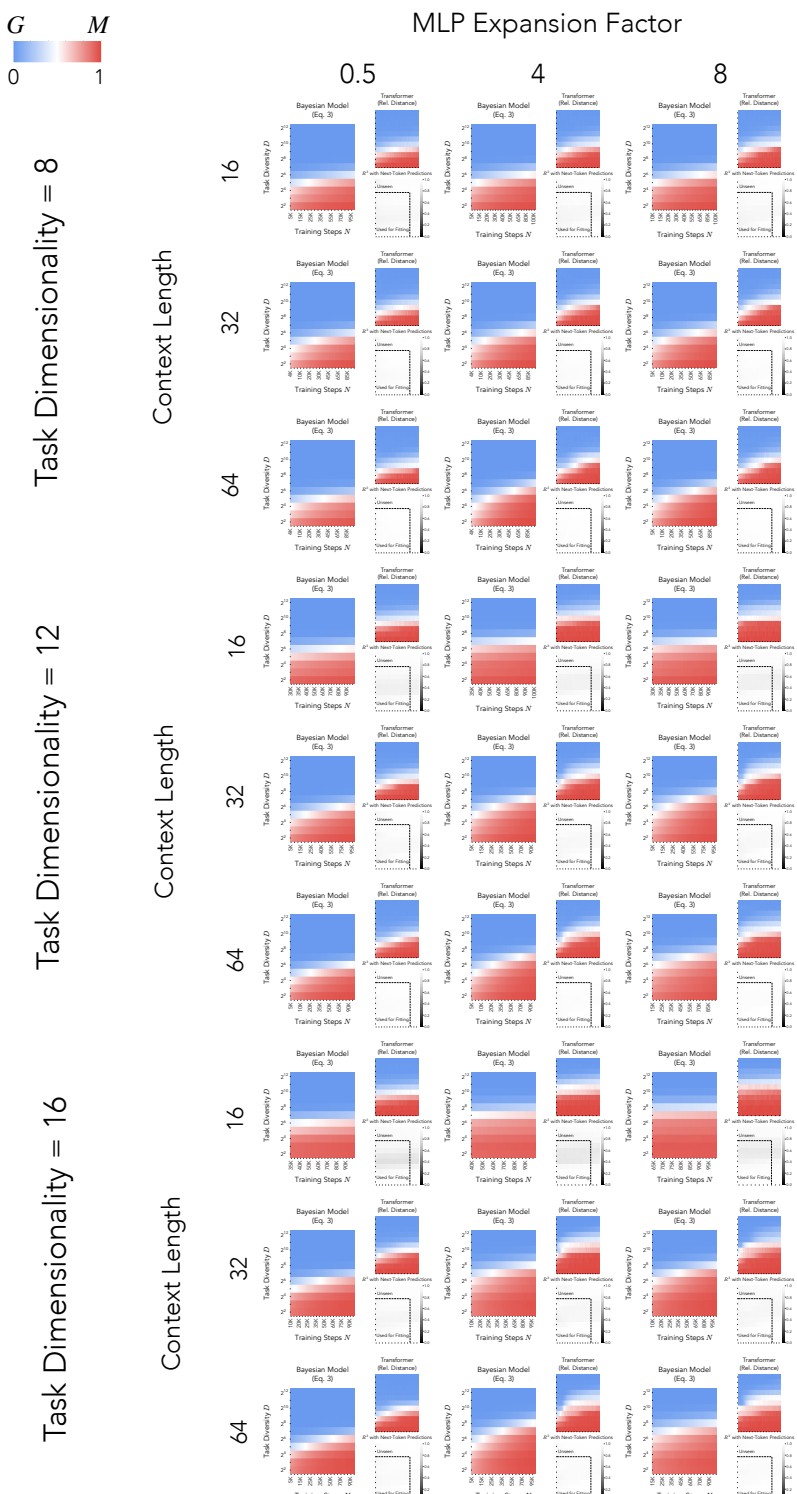

Figure 25: **Bayesian Model Predictions Across Linear Regression Conditions.** Red indicates closeness to memorizing predictor $M$, while blue indicates closeness to generalizing predictor $G$. Shown is a comparison between the posterior probability of the memorizing solution $M$ given by our Bayesian model (left) and the relative distance from the Transformer (top right), as well as heatmaps indicating similarity with Transformer next-token predictions (bottom right).

### H.4.3 Classification

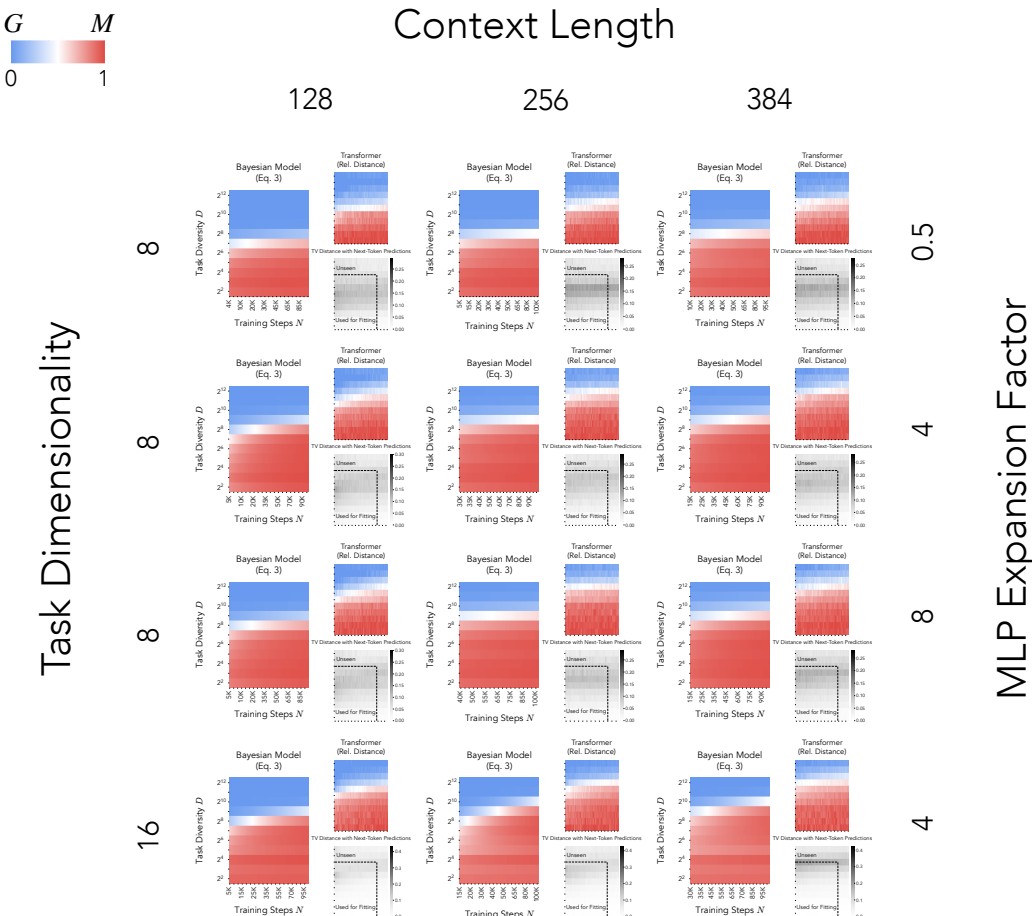

Figure 26: **Bayesian Model Predictions Across Classification Conditions.** Red indicates closeness to memorizing predictor $M$, while blue indicates closeness to generalizing predictor $G$. Shown is a comparison between the posterior probability of the memorizing solution $M$ given by our Bayesian model (left) and the relative distance from the Transformer (top right), as well as heatmaps indicating similarity with Transformer next-token predictions (bottom right). Max color bar value is determined by the performance of a baseline predictor that always outputs the mean of the distribution $\mathcal{T}_{\text{true}}$. The conditions where task dimensionality equals 16 in this setting (bottom row) reveal a limitation of our complexity measure: since the memorizing and generalizing predictors are very close in performance in low task diversities for these conditions, the loss term does not strongly bias the Transformer towards the memorizing predictor. However, the Transformer, is very close to the memorizing predictor for low task diversities, which would indicate according to our framework that memorizing few items is substantially simpler than implementing a copy operation. However, this is not captured by our complexity measure, since the compressed size of the code for the memorizing and generalizing predictors is roughly the same, thus we are unable to capture the bias toward the memorizing predictor in low task diversity settings. To overcome this, in these 3 conditions only, we heuristically multiply the bit size of the code for the generalizing predictor by 5, and with that fix, we find good performance (though as can be seen, the model still under-weights the memorizing solution for some low task diversity conditions).

## H.5 Extension to Multiple Strategies

To examine the generalization of our model to multiple predictors, we include a constant mean predictor in the hypothesis space for the linear regression setting, as models are known to learn this solution early [70, 101]. As shown in Fig. 27, our model successfully captures the dynamics of multiple predictors, and the inclusion of the optimal constant predictor allows us to capture Transformer behavior very early in training (from 20 steps), which was not possible using only the memorizing and generalizing solutions (App. E). To compare the posterior probabilities given by our model to the relative distance, we compute a generalized version of the relative distance by fitting a convex combination of strategies to Transformer next-token predictions at each $(N, D)$ condition. We find a high average cosine similarity of $0.96 \pm 0.003$ (SE) between the vector of relative distances from each predictor and our model's posterior probabilities.

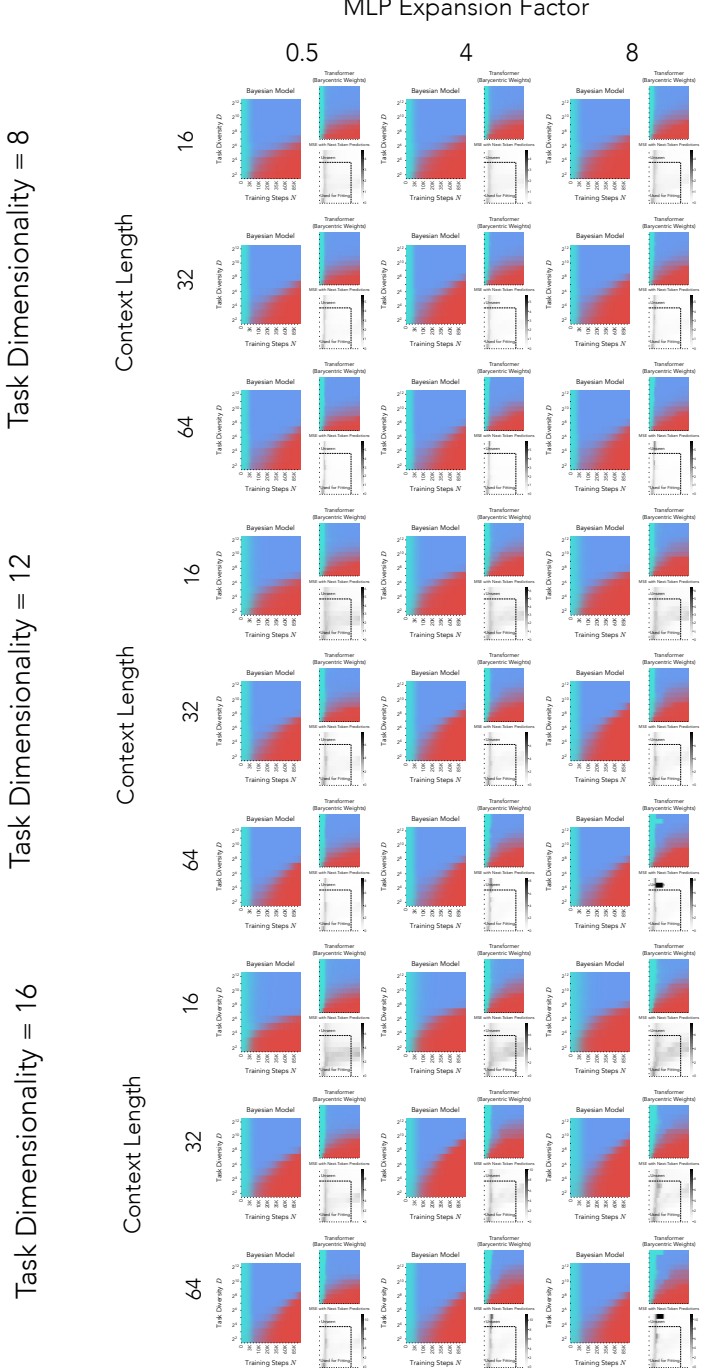

Figure 27: **Predictions of a Bayesian Model with Multiple Strategies Across Linear Regression Conditions.** Red indicates closeness to memorizing predictor $M$, blue indicates closeness to generalizing predictor $G$, and teal indicates closeness to the optimal constant solution $C$. We color the maps via a convex combination of posterior probabilities from our model for $M$, $G$, and $C$ (left), and use a similar method for coloring the generalized relative distance measure (top right). Finally, we show heatmaps indicating similarity with Transformer next-token predictions (bottom right). Max color bar value is determined by the average performance of the optimal constant solution across all conditions.

## H.6 Extension to In-Context Strategy Selection

We find that models evaluated in OOD settings behave more like the generalizing solution, a phenomenon we term *in-context strategy selection*. We extend our model to accommodate this behavior by allowing the posterior probability of a predictor to continue updating throughout the context (see D.4). We find that, for the Balls & Urns setting, this extension aids in capturing the shift toward the generalizing predictor seen during OOD evaluation (Fig. 28). Yet, particularly in higher task dimensionality, OOD behavior is not very well-captured by the model around the transition boundary, indicating the posterior updating throughout the context may lead other predictors to receive non-negligible posterior mass, or simply general instability at OOD evaluation (which is also seen in the linear regression setting). Additionally, we make a simplifying assumption when deriving the form for this extension, that the $\alpha$ and $\gamma$ parameters that govern the rate of update for the posterior during pretraining remain similar during in-context updating. Additional work is required for understanding the nature of updates from pretraining vs. ICL [73], and potentially incorporating these into Bayesian models of ICL.

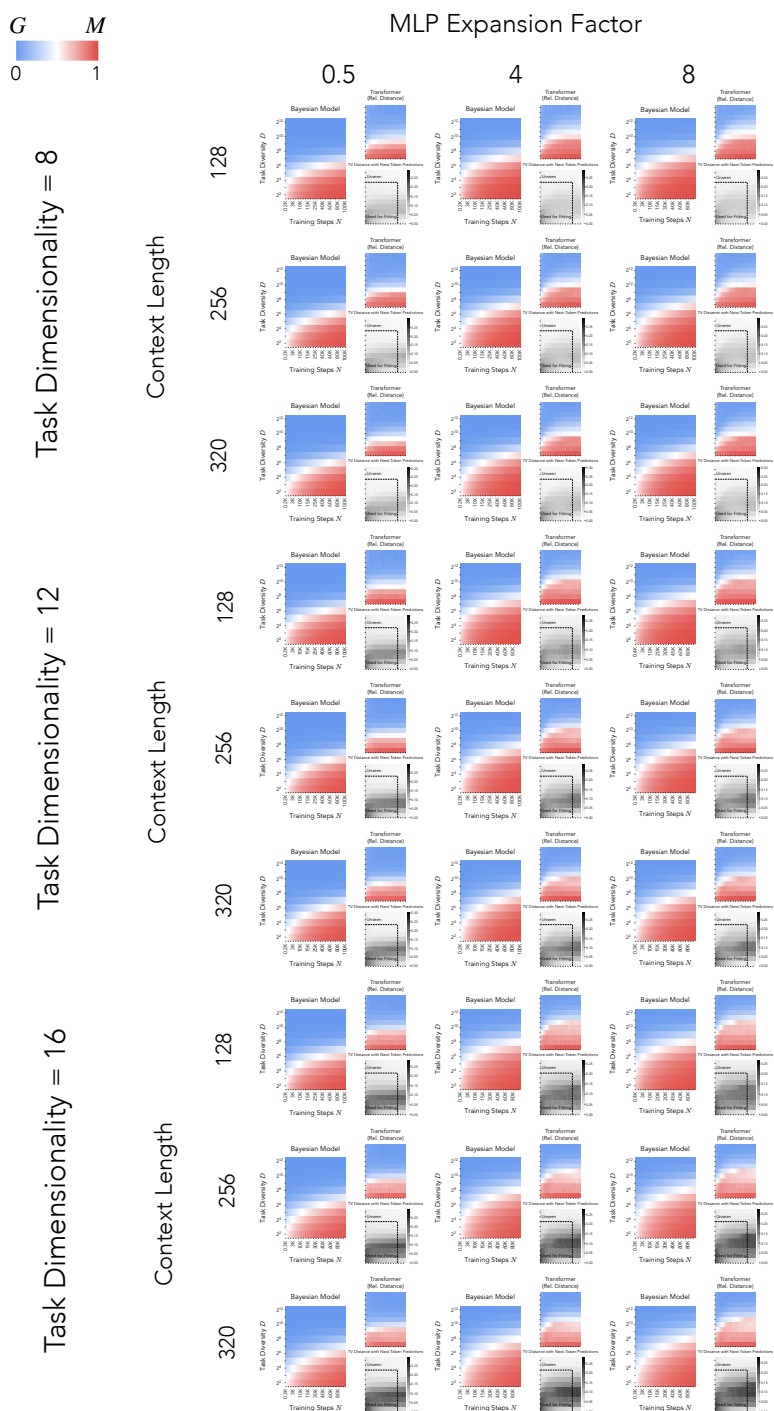

Figure 28: **Bayesian Model Predictions Across OOD Balls & Urns Conditions.** Red indicates closeness to memorizing predictor $M$, while blue indicates closeness to generalizing predictor $G$. All relative distance values are computed over OOD sequences drawn from $\mathcal{T}_{\text{true}}$. Shown is a comparison between the average posterior probability over a sequence of the memorizing solution $M$ given by our Bayesian model (left) and the relative distance from the Transformer (top right), as well as heatmaps indicating similarity with Transformer next-token predictions (bottom right).

# I Functional Form Ablations

Our functional form for the log posterior odds $\eta$ consists of 3 free parameters, $\alpha$, determining sublinear sample efficiency, $\beta$, a power law on the estimated Kolmogorov complexity $K$, and $\gamma$, a coefficient for the loss term. We find that each of these free parameters are necessary for the success of our model, since removing any of them results in a worsening of the model's ability to capture the phenomenology of ICL.

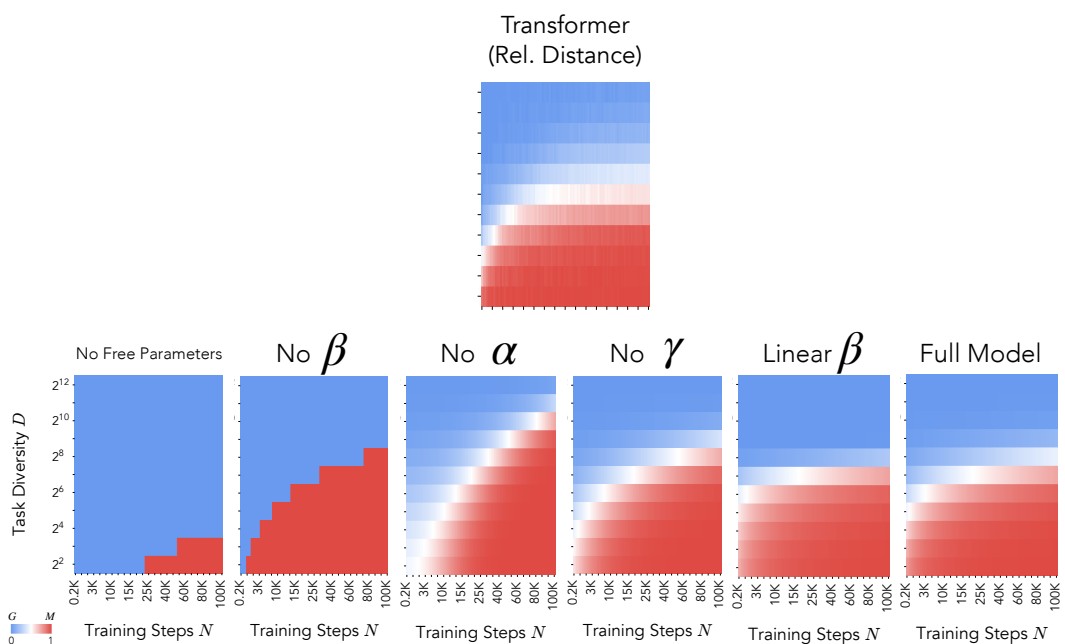

Figure 29: **Power laws over complexity measure and sample efficiency are necessary for explaining ICL phenomenology.** By ablating our functional form, we see that the free parameters $\alpha, \beta, \gamma$ are required for the performance of the model. In particular, the simplicity bias derived from $K$ without a $\beta$ term is much too sharp and over-penalizes complexity compared to the Transformer, and simply adding a linear $\beta$ term does not capture the contour of the transition or the gradual transience of generalization even in higher task diversity conditions. Additionally, memorization proceeds much too rapidly without the $\alpha$ term, pointing toward the necessity of assuming sub-linear sample efficiency for capturing Transformer training dynamics.

## J Memorization Continues to Increase After Task Diversity Threshold – Refutation of Raventós et al. [25]'s Claim

In Fig. 30, we show a refutation of the claim made by Raventós et al. [25], who claimed that after the task diversity threshold is reached, the Transformer will only continue to get closer to the generalizing solution throughout training, *regardless* of how long one trains. In making this claim, Raventós et al. [25] focused only on the *absolute* distance between the Transformer and the generalizing solution. However, when we considering a *relative* distance measure, we can show this claim to be false (see right side of Fig. 30): Even in conditions in which the task diversity threshold was reached and generalization is sustained throughout a reasonable amount of training (100K steps), we see that by absolute distance, the Transformer not only gets closer to the generalizing solution during training, it *also* continues to get closer to the memorizing solution (left side of Fig. 30). It seems that the rate at which the Transformer nears the memorizing solution is greater than that at which it nears the generalizing solution in task diversity settings such as $D = 512$ or $D = 256$, which is why the relative distance continues to grow even despite the transformer nearing the generalizing solution. Note also, that in settings that clearly show transience, e.g., $D = 64$, the distance from the generalizing solution shrinks until a certain critical point in which it begins to grow. It is likely that this point can be reached in all task diversity settings examined given the growth trend of the relative distance. However, it may require a very long training process to reach that point.

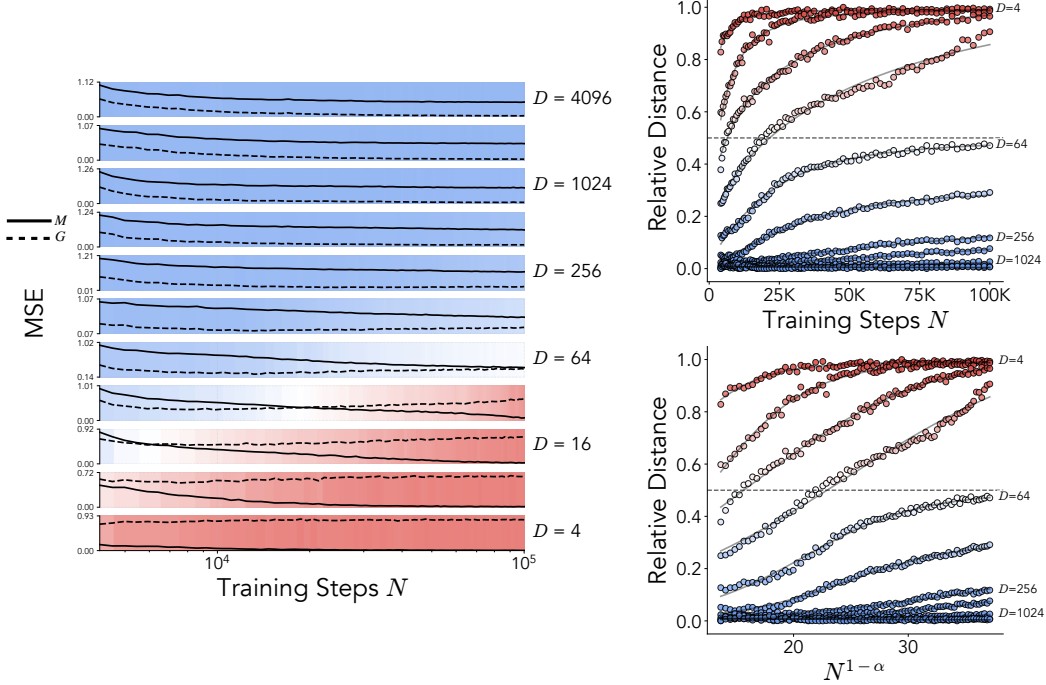

Figure 30: **Relative distance continues to rise throughout training, even after the task diversity threshold.** The figure displays relative distance, as well as absolute distance from the memorizing and generalizing solutions, for the linear regression setting with context length of 32, MLP expansion factor of 4, and 8 dimensions.

# K   Learning Rate Annealing Can Improve Adherence to Bayes-Optimal Trajectories

In our main experimental settings, in which we train with a constant learning rate, we find that relative distance trajectories follow a sigmoidal growth pattern with respect to $N^{1-\alpha}$ (see Fig. 6 and Fig. 31(a) top and bottom right). However, in contrast with our theory's predicted sigmoidal curves which plateau at 1 (i.e., *some* amount of training would eventually yield full adherence to the memorizing solution, when it has a lower loss than the generalizing solution), this does not seem to be the case with the trajectories displayed by Transformers, which appear to plateau *early* (see Fig. 31(a) top right $D = 256$ for a clear example). Indeed, fitting parameterized logistic curves to these trajectories yields plateau values different from 1. To explain this, we turn to foundational work from Geman and Geman [102], who showed that a slow (logarithmic) temperature cooling schedule for Gibbs sampling substantially increases the likelihood of convergence to the global minimum (MAP estimate). Drawing a very rough parallel to the case of deep learning, it is reasonable to assume that a learning rate annealing schedule of some form is required to converge to the MAP estimate (which is the memorizing solution, in cases where it has lower loss than the generalizing solution). To test this, we repeated experiments in the Balls & Urns and Linear Regression settings and used warm-up and inverse-squared learning rate decay. Surprisingly, we indeed find that adding learning rate annealing can increase adherence to Bayes-optimal trajectories (Fig. 31(a)). However, we also find this effect is highly sensitive to training conditions: training for longer, even in conditions that yield the effect for a smaller number of training steps, can lead to plateau (Fig. 31(b), top), and slight changes in training conditions, in this case number of warm-up steps, can substantially reduce the effect (Fig. 31(b), bottom). It should also be noted that in this case, adherence to the Bayes-optimal trajectory is actually *negative* for generalization, since it means the model will converge to the memorizing solution quicker. However, this is not necessarily the case in more realistic training regimes, where the ability to overcome a simplicity bias and adopt more complex solutions is likely beneficial.

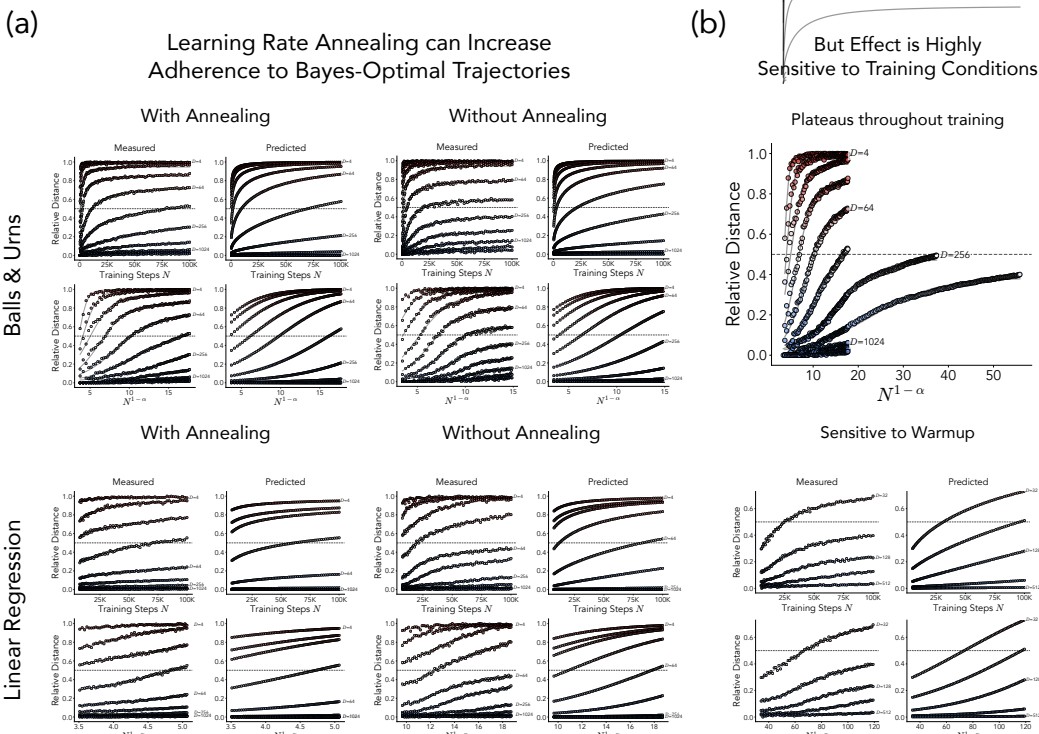

Figure 31: **Learning Rate Annealing Can Increase Adherence to Bayes-Optimal Trajectories.** (a) Balls & Urns setting uses context length of 128, MLP width of 256, and 8 dimensions. Linear regression uses similar variables except a context length of 16. (b) Effect is highly sensitive to training conditions: e.g., Training in the Linear regression setting with 5000 warmup steps failed, but succeeds with 500 warmup steps (the number used for the experiment in panel (a)).

## L Code Availability

All code used to run the experiments and analysis, as well as evaluation metrics for all settings, is available at: https://github.com/DanielWurgaft/rational-icl

