# OpenReview forum: "In-Context Learning Strategies Emerge Rationally"
_NeurIPS.cc/2025/Conference — NeurIPS 2025 poster_

### Official Review · Reviewer_PqQ7 · 2025-06-20

**Clarity:** 3
**Significance:** 3
**Originality:** 2
**Rating:** 5
**Confidence:** 4

**Summary:**

The paper studies the generalization behaviour of in-context learning to new tasks given a fixed number of training tasks and training steps. They propose a rigorous theoretical model of in-context learning as Bayesian inference over two solutions—memorization and generalization—with a prior for low-complexity hypotheses. They show that their model accounts for fine-grained empirical data, reproducing the precise tradeoff points where memorization transitions to generalization and vice-versa.

**Questions:**

1. It is possible that the thing that is important for characterizing the generalization behaviour of the model is not just the task diversity (i.e., the number of training tasks seen), but rather **the complexity of the task distribution**. This shifts the focus from talking about the “true” vs. “empirical” task distribution to talking about “simple” vs. “complex” task distributions, and a sequence model’s tendency to prefer simple ones. Let me explain. The empirical distributions in the regime of few training tasks is simple (a model just needs to memorize a handful of tasks), and in the regime of many training tasks that distribution is complex (a model would need to memorize far more tasks). Being a neural network, the sequence model prefers a simpler prior over tasks in the later case, which happens to coincide with what you’ve called the “true” task distribution. But what if you actually make the “true” task distribution complex! For instance, you can sample $N$ possible task parameters from a distribution $T_{simple}$(in the same way that you currently define $T_{true}$) and then convolve them with a Gaussian, so that you have a very complex multi-modal “true” task distribution $T_{true}$. I bet that even if you sampled $|T_{train}| \gg N$ training tasks from this “true” distribution, the model would **not** learn a prior that corresponds to $T_{true}$, but would instead learn a prior that corresponds to $T_{simple}$. Again, this shifts the focus from “true” vs. “empirical” or “generalization” vs. “memorization” to now being something about “simple” vs. “complex” task distributions, which need not be defined w.r.t. some true underlying distribution. My questions are:
    1. Do you agree with this framing and the hypothesized finding?
    2. Can you run this experiment and present the results in at least an Appendix section, along with a discussion of their implications for how one should precisely interpret your theory and results?
2. You write your universal prior $p(Q) = 2^{-K(Q)^\beta}$ in a way that suggests it is a valid distribution, but it is not. The universal prior is a semi-measure, not a valid measure. Did you normalize this distribution such that $p(G) + p(M) = 1$, in which case you probably meant $p(Q) \propto 2^{-K(Q)^\beta}$ instead?
3. Why do you have to “estimate a loose upper bound” for $K(Q)$ using “compression algorithms to the code and data required for describing the predictor” (L.203)? Don’t you know $K(G) = L(T_{true})$ is the length of the function describing your true distributions $T_{true}$ (which are all easy to figure out, given that you know the code needed to efficiently define something like a categorical or noise distribution), and that $K(M) = L(T_{true}) - \log_2 p_{T_{true}}(T_{train})$ where the second term comes from Huffman encoding in a 2-part code? Clearly for the way you’ve defined $T_{train}$, a 2-part code using $T_{true}$ as the model would be the optimal strategy, and even if you misestimate $K(T_{true})$ by a couple bits here and there you still have a free parameter $\beta$ in your universal prior that can correct for this.
    1. Note that if you don’t trust that this is the best compressor for both distributions, you can simply check that it gives a lower bound that what you currently have with the other compressors you’ve used, since the codes above are definitely valid ways to compress the two hypotheses.
4. Equation 2 and L.216, you expand $\log [P(M) / P(G)] \implies -\Delta K(D)^\beta$, and then you define $\Delta K(D)^\beta = \log 2(K_M(D)^\beta - K_G(D)^\beta)$. I’m a bit lost on several fronts here:
    1. What is $\log2$? I’m assuming it has to be $\log_2$ based on everything else, but you’ve written it in a way that means $\log(2)$. For subsequent follow-up questions right below, I’m going to assume you meant $\log_2$.
    2. What is $K_M(D)$ or $K_G(D)$? You haven’t defined these terms anywhere. Did you mean $K(M)$ and $K(D)$?
    3. I think the derivation is wrong. It should be $\log_2 [P(M) / P(G)] = \log_2[2^{-K(M)^\beta}] - \log_2[2^{-K(G)^\beta}] = K(G)^\beta -K(M)^\beta$, and so you should be defining $\Delta K(D)^\beta = K(M)^\beta -K(G)^\beta$ for Eq. 2 to be valid. Am I missing something? Why do you have the $\log_2$ in your definition for $\Delta K(D)^\beta$?
    4. You using a notation for the complexity term $\Delta K(D)^\beta$ that has $D$ in it, but this is initially confusing because $K(M)$ and $K(G)$ are not denoted in terms of $D$. The trick is that $M$ depends on $D$ implicitly because it has to memorize it, but $M$ doesn’t depend on $D$ (please correct me if I’m wrong). To clarify things, can you switch to denoting $M \rightarrow M_D$ and correspondingly $K(M) \rightarrow K(M_D)$?

**Ethical Concerns:**

["NO or VERY MINOR ethics concerns only"]

**Final Justification:**

The authors have addressed my main concerns, and I maintain that this is a strong paper on an important topic, deserving acceptance so that it has better visibility in the ICL community.

**Limitations:**

Yes.

**Paper Formatting Concerns:**

None.

**Quality:**

3

**Strengths And Weaknesses:**

**Strengths**

1. The problem studied is important and remains unsolved in the field. While I think many in the field already had intuitions that were aligned with the theory presented in this work, it lays those intuitions out quite rigorously.
2. The proposed model fits the empirical data extremely well, which inspires great confidence in the authors’ theory.
3. The paper is largely clearly written and pedagogical (although see my comments below on parts of Section 4).
4. The figures are excellent at illustrating the hypotheses, methodology, and results. The figures and captions together largely tell a self-contained story, and somebody could get a good understanding of this work just by reading them alone, which is always nice.
5. The future work suggested is thought-out and genuinely interesting.
6. Overall, my impression of this paper is quite positive. I have to admit that I went into it feeling like many of the statements being made early on in the abstract and introduction were quite obvious and well-known, but the precision and rigour of the proposed model is very nice to see and I feel like I now have a better understanding of ICL.

**Weaknesses**

1. In line with my comment just above, I think the biggest weakness is one of perception, especially early on in the paper. The problem is set up on L.30-35 in a way that is not specific to ICL. Questions like “why does an ability to learn novel tasks emerge, given that memorizing the training distribution almost always leads to better performance” apply to *all* of deep learning as a whole, not just ICL. If one just thinks of the sequence model as a neural network taking sequence-structure inputs (and it is that!), then many of the stated questions simply reduce to “why do deep neural networks generalize?”, and this is a question that has been asked many times before. In fact, the answers that have been proposed look very much like the one here, in that they show deep neural network training can be seen as Bayesian inference where there is a prior for simple solutions (Wilson 2025 “Deep Learning is Not So Mysterious or Different”, for instance). Again by the time I got to the end of the paper it was clear that it differs from the general deep learning question (e.g., you consider memorization of the sequence *generative parameters*, rather than memorization of the *sequences themselves*, hence your ID evaluation) and that the proposed model does have something interesting to offer, but I think you should make some effort to rework the abstract and introduction in order to head off these kinds of sceptical reactions. How to best do that, I’m not sure unfortunately.
2. Section 4.0 needs some cleaning up, both in terms of math correctness (I think) and notation. It can also be clarified, especially near the end of Section 4.0 where it is very definition-heavy and readers might get lost in notation that was defined earlier on. I had to scramble around to find what terms like $L_G(S_{\mathcal{T}_{train}}(\infty, D))$ referred to, for instance.
3. Fig. 5 should appear before Fig. 4, as it gives intuition for why we see the patterns in Fig. 4. Similarly, the contents of Section 4.1 should be described before the paragraph on “Validating the Model”, as it is there to provide intuition for the model.
4. I would rename and reframe Section 4.2 from “novel predictions” to “features explained by the model”. While I can technically see how these are “novel predictions”, really you’re not using your theory to predict the results of new experiments, but rather are using it to explain why the empirical data you’ve already presented looks the way it does. Your model is explaining observed data, not making new predictions for novel experiments. You can then consolidate the paragraph on “Validating the Model” into Section 4.2 Features Explained by the Model (see my point directly above).

---

> ### Author Rebuttal · Authors · 2025-07-31
>
> We are deeply grateful to the reviewer for their thorough and insightful review. We thank them for supporting our work for acceptance, and were especially encouraged that they highlighted the “precision and rigour of the proposed model” and found that the “proposed model fits the empirical data extremely well, which inspires great confidence.” We were also happy to hear that the reviewer found our figures to be “excellent at illustrating the hypotheses, methodology, and results” and that our paper helped them develop “a better understanding of ICL.”
>
> The feedback was particularly valuable to us because the suggestions for improving the paper’s narrative and clarity align with and validate motivations behind a significant revision we recently prepared. Before addressing specific points, we wanted to briefly outline the narrative of the revised manuscript, as we believe it addresses some high-level suggestions.
> * Recent work analyzing ICL has identified a broad set of strategies that describe model behavior in different experimental conditions. We aim to unify these findings by asking *why* a model learns these disparate strategies in the first place as training conditions are varied.
> * We observe that when trained to learn a mixture of tasks, models transition between two Bayesian predictors, a memorizing predictor and a generalizing predictor.
> * This observation leads us to adopt a normative Bayesian account for modeling Transformer behavior. Under assumptions of simplicity bias and power-law scaling, our account is highly predictive of model behavior and yields an explanatory account of a loss-complexity tradeoff as core to ICL phenomena.
> * This narrative, which focuses on why distinct strategies are learned and transitions between them occur, allows us to focus on ICL phenomena as outcomes of a single, underlying loss-complexity dynamic. We believe this revised framing makes the paper's contribution clearer.
>
> With that context, we now address specific comments and questions in detail.
>
> -----
>
> >  **I think the biggest weakness is one of perception, especially early on in the paper… Again by the time I got to the end of the paper it was clear that it differs from the general deep learning question … and that the proposed model does have something interesting to offer, but I think you should make some effort to rework the abstract and introduction in order to head off these kinds of sceptical reactions.**
>
> We thank the reviewer for the valuable and constructive feedback! We agree with the general comments made, which is why, as stated above, we reworked the abstract and introduction to focus more specifically on ICL from the start, framing our investigation around the question of why models learn distinct strategies for performing ICL.  Yet, we note that we do in fact agree with the reviewer’s observation that our account can apply, as phrased in the current intro, to neural networks in general! In this paper we indeed focus on ICL, and so the hypothesis space consists of different predictors; however, in a more standard neural network case, the hypothesis space can consist of specific weights (e.g., in a mixture of linear regression tasks), and we are actively working on demonstrating our account can predict general neural network training dynamics and scaling phenomena! With that said, this paper indeed focuses on ICL, hence we changed the intro accordingly. Yet, we do want to highlight that, as opposed to works such as Wilson’s “Deep Learning is Not So Mysterious or Different”, we believe the strength of our account is that beyond offering intuitions or a descriptive lens, it is also highly predictive of model behavior throughout training, which we see as crucial for a successful account of deep learning.
>
> ------
>
> > **Section 4.0 needs some cleaning up, both in terms of math correctness (I think) and notation. It can also be clarified…**
>
> We thank the reviewer for the constructive comments, and we certainly agree! In our rewrite, we revised the notation and attempted to improve the clarity of section 4. For example, we define $\overline{L_G}(S_{\mathcal{T}_{train}}(D))$
>
> as the average loss of a predictor G on a dataset of sequences sampled from $\mathcal{T}_{\text{train}}$,
>
> and maintain this notation in the derivation, thereby avoiding confusing terminology such as $L_G(S_{\mathcal{T}_{\text{train}}}(\infty, D))$.
>
> We also revised several other elements in section 4, which we detail in the sections relevant to the reviewer’s specific questions.
>
> ------
>
> > **Fig. 5 should appear before Fig. 4…**
>
>  We agree and will revise accordingly!
>
> ------
>
> > **It is possible that the thing that is important...is not just the task diversity...but rather the complexity of the task distribution...Do you agree with this framing and the hypothesized finding? Can you run this experiment...?**
>
> This is a brilliant question, and we agree with your framing! The core axis is indeed the complexity of implementing each solution, which depends on the complexity of memorizing the generative parameters of each distribution. Indeed, in works such as [1], it is shown that more complex data can indeed drive learning of better generalizing predictors in the case where the true distribution (in the case of the paper, the hierarchical structure of language) is indeed complex. Your proposed experiment—making the "true" distribution more complex than the empirical one—is an excellent way to disentangle these concepts.
>
> We have actually run experiments in a similar spirit using a different task domain (learning Boolean functions). Preliminary results support your hypothesis: when tasks comprising the “true” distribution cannot be compressed into a simple generative process, we do not see the standard task diversity effect, as memorizing only the tasks in the training data is indeed simpler than learning the generalizing solution! In contrast, when the true distribution comes from a single “true” input-output map, which is “noised” to create slightly different maps for the training distribution, the Transformer does indeed transition to learning the true map with increased task diversity! This aligns with your idea that the model's inductive bias is for *simple generative processes*, not necessarily the "true" one. We will add a discussion of such findings and their implications to the appendix in the final version. Thank you for the thought-provoking suggestion.
> [1] https://arxiv.org/abs/2412.04619
>
> ------
>
> > **You write your universal prior …  in a way that suggests it is a valid distribution, but it is not.**
>
> You are absolutely correct. Thank you for this important technical correction. We have corrected the prior to be written as $p(Q) \propto 2^{-K(Q)^{\beta}}$, and we indeed meant that to be the case in our paper, but wrote it using an equality sign by mistake.
>
> ------
>
> > **Why do you have to “estimate a loose upper bound” for K(Q) using “compression algorithms"...? Don’t you know K(M) is the length of the function describing your true distributions...and that K(G) \approx D \times K(w_d)...?**
>
> This is an excellent and deeply insightful question. Our reasoning for using general-purpose compressors is subtle but important. The complexity term $K(Q)$ in our model is meant to capture the cost of a Transformer *implementing the predictor algorithm Q*. For example:
> * For the **Generalizing predictor (G)** in Balls & Urns, the algorithm is "count token occurrences in the context and add a pseudo-count".
> * For the **Memorizing predictor (M)**, the algorithm is "compute the likelihood of the context under each of the D stored urns and compute a weighted average".
>
> The 2-part code you describe is an optimal way to compress the *data* (the urn parameters $w_d$). However, it doesn't directly capture the algorithmic complexity of *implementing the inference procedure itself* (e.g., computing a histogram or Bayesian averaging). Our approach, by compressing the Python code that implements the full predictor, is an attempt to create a proxy for this algorithmic implementation cost.
>
> That said, your suggestion is very compelling. We suspect it might work well for settings like Balls & Urns, in which computational procedures are shared: both strategies require computing a histogram, but in the case of the memorizing solution, this histogram is compared with learned distributions, whereas the generalizing solution simply adds a 1 pseudo-count. However, we expect it might be less complete for settings like Linear Regression, where the complexity seems to lie more in implementing the ridge regression algorithm itself, compared to Bayesian averaging. We would be very interested in any further thoughts you have on this.
>
> ------
>
> > **Equation 2 and L.216, you expand…**
>
> We are grateful for your meticulous reading of our derivations. You were right to be confused—our notation in the original draft was unclear and contained typos. We sincerely apologize for this. As mentioned in our opening, we have thoroughly revised Section 4 and Appendix D to address every point you raised:
>
> 1. $\log2 (K_M(D)^{\beta} - K_G(D)^{\beta})$ should be written as $(K_M(D)^{\beta} - K_G(D)^{\beta}) \cdot \log2$ to avoid confusion.
> 2. We indeed meant $K(M)$ and $K(D)$ and will adopt your notational suggestion from part 4. to prevent confusion.
> 3. We include the $\log2$ term since we use the natural log rather than $\log_2$ in our derivation, therefore a $\log_{{\rm e}}2$ constant appears in the term for $\Delta K(D)^{\beta}$. Also, we define $\Delta K(D)^{\beta} = (K_M(D)^{\beta} - K_G(D)^{\beta}) \cdot \log2$ rather than $\Delta K(D)^{\beta} = (K_G(D)^{\beta} - K_M(D)^{\beta}) \cdot \log2$ to highlight the tradeoff between the loss term and the complexity term in the equation for $\eta$, since defining $\Delta K(D)^{\beta}$ in former way yields a minus sign between the two terms in the equation for $\eta$, whereas defining it the latter yields a plus sign between the terms.

---

> > ### Comment · Reviewer_PqQ7 · 2025-08-04
> >
> > Thank you very much for taking all of my feedback into serious consideration and addressing it. I think that the rewrite of the intro/abstract that you've described will go a long way towards preparing readers for the specific contributions made in this paper, and that the fixes to Section 4 will help people follow the methods more smoothly --- I look forward to reading the updated draft!
> >
> > I'm going to maintain my current score as that is where I think the paper stands: I think that this is strong work that will be interesting to the field of ICL and deserves acceptance. I will advocate as such in the discussion with other reviewers and the AC.
> >
> > *Other points in the rebuttal*
> > - Happy to hear that you've already looked into disentangling the notion of the "true" distribution vs. the "simplest" one. I was not necessarily suggesting, though, that the memorizing solution would be preferred to the true Bayesian one in cases where the prior over tasks is complex (although I can imagine cases in which that would happen). What I had in mind was that the "generalizing" solution would learn a simpler approximation of the complex true prior over tasks (e.g., something smoother with fewer scattered modes). I'm still curious if you see this happen in the case I described in my original review, but either way I look forward to reading your Appendix section on this topic.
> > - Regarding your point about needing to compress the inference solutions rather than the data-generating process, I see your point. I forgot what confused me in the first place, but compressing the python code implementing the solution sounds like a very reasonable way to tightly upper-bound the complexities of these solutions.

---

### Official Review · Reviewer_Zg8o · 2025-06-29

**Clarity:** 2
**Significance:** 2
**Originality:** 2
**Rating:** 4
**Confidence:** 2

**Summary:**

This paper provides a framework to explain the effects of in-context learning in Transformers by training a lightweight Bayesian model. They define two components: the memorizer predictor and the generalizer predictor -- these aim to predict the performance across three tasks: (1) balls and urns, (2) linear regression, and (3) binary classification.

**Questions:**

I might have missed where these details are included, but the experimentation details are quite unclear to me:
1. Do the authors use a specific Transformer-based model, or just experiment with their own stack of Transformer blocks?
2. Where is the data from the 3 problem settings obtained from?

Additionally,
1. What are the chosen $\alpha, \beta, \gamma$ values learned for the experiments?
2. Is there an intuitive interpretation to $\alpha, \beta,$ and  $\gamma$?

I'm happy to raise my scores!

**Ethical Concerns:**

["NO or VERY MINOR ethics concerns only"]

**Final Justification:**

As I mentioned in the response to the authors, I had misunderstood some aspects of their paper. I also read through the other reviews and responses -- I did not have any additional questions or concerns.

**Limitations:**

Yes

**Paper Formatting Concerns:**

None, the formatting looks good.

**Quality:**

2

**Strengths And Weaknesses:**

Strengths:
- The paper tackles an interesting topic, with a unique approach
- It is a well-organized paper
- The mathematical formulation is laid out clearly throughout the paper

Weaknesses:
- Scope of chosen problems (ball and urns, linear regression, binary classification) are more of toy problems than real, large-scale problem statements.

---

> ### Author Rebuttal · Authors · 2025-07-31
>
> We thank the reviewer for their feedback! We are glad that they found our paper “tackles an interesting topic, with a unique approach”, that it is “well-organized”, and that the “mathematical formulation is laid out clearly throughout the paper.” Below, we respond to specific comments.
>
> -----
>
> > **Scope of chosen problems (ball and urns, linear regression, binary classification) are more of toy problems than real, large-scale problem statements.**
>
> Fair point! While we agree that our experiments use synthetic tasks, we emphasize that these tasks are *precisely* the ones that have been studied in the bulk of the literature to understand ICL [1-12]. We thus argue that any theory targeted at explaining ICL must at least show predictive power over this spectrum of simple, controlled, but phenomenologically very rich set of tasks that form the focus of prior work. Our hierarchical Bayesian framework meets this standard, offering both a predictive account of Transformer behavior throughout training and explanations for several past results on ICL in a unified manner.
>
> Having achieved this result, we now strongly think that the stage is set for analyzing more naturalistic settings. We have in fact already begun preliminary explorations on this front, analyzing the phenomenon of many-shot jailbreaking wherein safety fine-tuned models, when offered in-context sufficiently many exemplars of some undesirable behavior (e.g., producing violent speech), start to engage in said behavior. We believe our framework can offer a predictive account of this phenomenon with realistic models, but a detailed analysis warrants its own explicit study!
>
> [1] arxiv.org/abs/2205.05055
>
> [2] arxiv.org/abs/2312.03002
>
> [3] arxiv.org/abs/2412.00104
>
> [4] arxiv.org/abs/2311.08360
>
> [5] www.jmlr.org/papers/v25/23-1042.html
>
> [6] arxiv.org/abs/2310.08391
>
> [7] arxiv.org/abs/2503.05631
>
> [8] arxiv.org/abs/2402.11004
>
> [9] www.pnas.org/doi/abs/10.1073/pnas.2502599122
>
> [10] arxiv.org/abs/2211.15661
>
> [11] arxiv.org/abs/2306.15063
>
> [12] arxiv.org/abs/2412.01003
>
> [13] www.anthropic.com/research/many-shot-jailbreaking
>
> -----
> -----
>
> **Questions about experimental details**
>
> > **Do the authors use a specific Transformer-based model, or just experiment with their own stack of Transformer blocks?**
>
> We refer the reviewer to experimental details mentioned on page 30. In brief, we note that for all settings, we use the GPT-NeoX architecture sourced from Huggingface [70]. While the number of layers / blocks in the model depends on the specific experimental setting (as reported in that section), we use only 1 attention head per layer and follow a sequential residual stream architecture across all settings. In the Balls and Urns and Classification settings, we use one layer. In the Linear Regression setting, we use eight layers to ensure the model can express the ridge regression solution.
>
>
> > **Where is the data from the 3 problem settings obtained from?**
>
> Thank you for the question that offers us a chance to clarify our experimental setup! As we mentioned on pages 3-4 in the main text and detailed much more extensively on pages 32-34 in the appendix, we use an online learning procedure for sampling data from the training distribution. From page 3: “During training, each input sequence of length $C$ is generated by randomly sampling a function $f(w, \cdot) \in \mathcal{T}_{\text{train}}$” and “Batches consist of independently generated sequences.”  The appendix offers several more details, with visualizations clarifying the sampling protocol in further detail.
>
>
> -----
>
> #### **Questions regarding free parameters**
>
> Thank you for the questions regarding the free parameters, which are an important part of our model!
>
>
> > **What are the chosen $\alpha$, $\beta$, $\gamma$ values learned for the experiments?**
>
> We believe there may be a slight misunderstanding here and take this opportunity to clarify the same. In particular, we note that the free parameters of our framework, i.e., $\alpha$, $\beta$, $\gamma$, are fitted to a trained model’s next-token predictions. Specifically, we minimize the cross entropy between our predictor (Equation 3) and the trained Transformer’s next-token predictions, yielding the values of free parameters. We strongly emphasize that these same three free parameters show almost perfect generalization to next-token predictions of an order of 100 experimental settings, while also capturing the empirical relative distance maps from Figure 3! That is, our hierarchical Bayesian model offers predictive power at the level of trained Transformer’s phenomenology, not just its behavior (i.e., next-token predictions).
> In case the reviewer finds it useful, we are also happy to add a detailed table describing the precise values of the three free parameters to the paper’s appendix!
>
> > **Is there an intuitive interpretation to $\alpha$, $\beta$, and $\gamma$?**
>
> Indeed! We provide a brief summary next.
>
> - **Rate of Evidence Accumulation ($\alpha$):** This adjusts the power on the effective sample size $N^{1-\alpha}$ and can hence be seen as modulating the sample efficiency of the Transformer. The term emerges from the scaling exponent involved in power-law scaling of neural network loss with dataset size [1], which we assume to be an axiomatic property of neural network training for deriving our framework (see page 27 for the derivation). In the language of Bayesian inference, one can also deem $\alpha$ as a measure of the rate at which a Transformer accumulates evidence from the data seen during training. As we discuss on page 8, learned values of $\alpha$ turn out to be less than 1, indicating sublinear sample efficiency of Transformers compared to an optimal Bayesian learner.
>
>
> - **Complexity Penalty ($\beta$):** This parameter captures the extent to which a Transformer has an inductive bias towards implementing simpler predictors. If we had an objective measure of complexity of implementing a predictor in a Transformer, we would not need to model this free parameter. However, given that that is not the case, we use (an upper bound for) the Kolmogorov complexity of the predictor by compressing its python implementation via off-the-shelf compression algorithms. This leads to the definition of $\beta$: it captures how much more costly implementation of the predictor would be in a Transformer, a suboptimal computational system, compared to a universal Turing machine! Specifically, given that $\beta$ adjusts the penalizing of complexity by the prior for a predictor, we can thus interpret it as a ‘complexity penalty’ on page 9, and show that our model accommodates the increase in memorization with expanding MLP width via a decrease in the ‘complexity penalty’ $\beta$.
>
>
> - **Effects of initial loss ($\gamma$):** On page 7, we note that $\gamma$ is a constant related to the term A from the scaling laws assumption. More specifically, in the derivation on page 27, it can be seen that the free parameter $\gamma$ encapsulates $\frac{A}{L(Q)}$, with $A$ corresponding to the loss of a random predictor on the dataset, and $L(Q)$ being the loss of the optimal Bayesian predictor (irreducible error). We only fit a single free parameter $\gamma$ across different task diversity settings in order to simplify our resulting functional form.
>
> [1] arxiv.org/abs/2102.06701
>
> -----
> -----
>
> **Summary:** We thank the reviewer for their helpful comments that helped us demonstrate the generality of our framework and contextualize the intuitive interpretation of variables involved within it. We hope our responses have appropriately addressed the reviewer’s concerns and, if so, hope that they will consider raising their score to support the acceptance of our work!

---

> > ### Comment · Reviewer_Zg8o · 2025-08-01
> >
> > Thank you to the authors for their detailed response. It seems I had misunderstood some aspects of their work, but now they are clearer to me. I will update my score accordingly.

---

### Official Review · Reviewer_JDTE · 2025-07-02

**Clarity:** 4
**Significance:** 2
**Originality:** 3
**Rating:** 3
**Confidence:** 3

**Summary:**

This is a methodology-experimental paper, which explores in-context learning in a general stylized model, and 3 different specific settings. The paper defines memorization (M) (resp. generalization, G) as the case in which the test prediction is based on Bayesian weightening of the predictors according to their empirical appearance in the training set (resp. in the population distribution). The paper empirically validates that two main phenomena occur for transformer-based training in all 3 settings: (1) transition from memorization to generalization when the training task diversity increases. (2) A deterioration of the test (out-of-distribution) performance when the length of the training increases. Then, it is proposed to model the predictor as a Bayesian mixture of a memorizing predictor and a generalizing predictor. The log-likelihood of the posterior odds is parameterized as a sum of a loss term and a complexity term. The model is used to predict the distance to M and G predictors as the function of the number of samples, that the change between them is rapid, and that it increases with task diversity.

**Questions:**

In parallel to the weaknesses above:

[Q1] What is an evident for the uniqueness of your explanation in the space of various possible explanations? How does the unified view proposed in this paper contributes to the validity of your explanation?

[Q2] What is an empirical evident that a *specific* complexity term is the one to accurately predict the experimental results?

[Q3] What makes the sub-linear growth a prediction, and what can be said on its exponent?

[Q4] What is different in this context from the well-established phenomenon that long training prevents generalization?

[Q5] How could this serve as an estimate for this transience time in new tasks?

**Ethical Concerns:**

["NO or VERY MINOR ethics concerns only"]

**Final Justification:**

I accept the authors answer that the paper clarifies reported results from previous works on the subject, which were misleading.

In previous works, simple toy models were suggested, to obtain some intuition of the ICL phenomenon. The authors take these as indispensable and aim to provide a unifying theory.  The main intuition of transitioning from M to G is apparent in these works.  For example, I don't think that the loss-complexity trade-of is implicit in these works. This is a completely standard ML phenomenon.
From my perspective, the authors empirically observe a very high level phenomena, and match it with a basic model.
I find the prior assumption of two-hypothesis to be restricting, and the new addition of a third hypothesis does not ameliorate that. The goal is to show that a model, free from any prior restriction, actually transitions between such predictors.

Regarding the predictive power: from my perspective, since the hypothesized models M and G do not take into account any aspect of the model's architecture, they do not allow any intuition on why the specific architecture works and how to build a better one. It just explains what it currently does, and as said, in a high-level manner (a general loss term, a general complexity term).


I thus maintain my score.

**Limitations:**

Yes

**Quality:**

2

**Strengths And Weaknesses:**

**Strengths:**

[S1] The paper addresses a timely and central topic: In-context learning is one of the most important aspects of the recent advances in ML, and the question on under what circumstances it arises is important. The step towards unification of various models proposed and phenomena observed in the literature is useful.

[S2] The conclusion is based on extensive empirical evidence, and the graphs displayed are convincing.

[S3] Overall, the paper is well written and easy to follow.

**Weaknesses:**

[W1] The main weakness is that, from my perspective, the paper is fully derived from the empirical observation of the transitions from memorization (M) to generalization (G). This was already observed in previous works [23,27, maybe more]. From that point onward, the paper reads like a **tautology**. Specifically: (1) It is obvious that if the empirical predictor is very close to either the M or G predictor, and sharply transitions between them, then the only the posterior weights of these predictors will be close to 1, and otherwise tend to zero. (2) It is obvious that if the prior is modeled by a complexity term and the likelihood by a loss term, then the posterior would be comprised of them, and would appear as a loss-complexity trade-off. The paper aims to answer why ICL emerges, but I think that instead it just fits a model to the observed transition between M and G. In other words, the paper observes a transition between M and G, then models the prior as complexity terms, and then explains that the transition is due to balance between complexity and loss. If the prior was modeled in a different way, by a term of different nature, then the “why” explanation would have changed accordingly.

[W2] In continuation to the above, I believe it is agreed that the prior is some sort of **a** complexity term. The weakness of the paper is that it does not address the main question of **which** complexity term? The paper mentions Kolmogorov complexity, but it is well known that this universal complexity term is uncomputable. Any compression algorithm may indeed lead to a measure of complexity, but again, the questions is which complexity term characterizes the ICL phenomena.

[W3] The prediction of sub-linear growth is weak in the sense that a reader cannot judge the order of events. It is possible that you have observed a sub-linear growth, and then decided on the modeling assumption A1. An expectation from a theory is not just to claim for a sub-liner growth, but rather to quantify the exponent $\alpha$ in some way.

[W4] The paper mentions that transient generalization phenomenon as an important observation, but it is well established that long training prevent generalization (hence, early stopping techniques).

[W5] The scaling of time to transience with task-diversity $N^*(D)$ depends on the complexity of the M and G models, as well as their losses, which, of course, unknown a priori.

---

> ### Author Rebuttal · Authors · 2025-07-31
>
> We thank the reviewer for their detailed and thoughtful feedback. We are glad that they found that our paper “addresses a timely and central topic” and that “the step towards unification of various models proposed and phenomena observed in the literature is useful.” Further, we are grateful the reviewer's note that our “conclusion is based on extensive empirical evidence, and the graphs displayed are convincing”.
>
> Below, we respond to specific comments. Since the questions map on to comments in the weakness section, we bundle them up and respond to both together.
>
> -----
>
> > **(W1) & (Q1): The main weakness is that...the paper is fully derived from the empirical observation of the transitions from memorization (M) to generalization (G)...From that point onward, the paper reads like a tautology...I think that instead it just fits a model to the observed transition…**
>
> We *respectfully* disagree with the reviewer that our analysis is tautological or involves obvious implications derived from the transitions between memorizing and generalizing predictors observed in prior work. To clarify our position, we respond to two specific points that we think summarize the reviewer’s comment.
>
> - **Grounding of our analysis in the transitions between M and G.** While the reviewer is correct that transitions between different predictors have been observed in prior work, *we emphasize that our characterization of these transitions is both substantially richer and highly indicative of a unifying theory of ICL*—to our knowledge, our unifying formalization and predictive modeling of transitions between different predictors has not been proposed in prior work. More specifically:
>    - Prior work focuses on demonstrating these transitions with respect to a single control variable, i.e., task diversity or training time, but we in fact show this phenomenon involves a rich interplay between the two variables, yielding distinct phases. Thus, given that several past findings on ICL have characterized these predictors behaviorally, the intuition driving our work is that explaining *transitions* between different predictors learned by a model is the critical bottleneck towards understanding ICL.
>    - Our highly predictive model *builds upon* the observation of distinct phases characterized by each predictor, rather than directly stemming from it. Our model builds upon this observation by considering the two predictors as the model’s hypothesis space. Yet, it goes beyond it by characterizing model inference-time behavior as a posterior-weighted average of predictors, indicating the model weighs preference for each predictor throughout training, and interpolates between predictors at inference time. We believe the finding that a transformer appears to interpolate between different solutions in its response patterns is a non-trivial finding that cannot be directly inferred from the observation of transitions.
>
> - **Mere fitting of some assumed model to the transitions / different models would yield different explanations.** We first note that our hierarchical Bayesian model, which is highly predictive of a trained Transformer’s behavior, emerges from two well-known and generally observed empirical properties of neural networks: scaling laws and simplicity bias. While the reviewer is correct that a different model with different assumptions would have led to different explanations, we strongly emphasize that this empirically does not work out: in Appendix I (functional form ablations), we explicitly attempted several variants of our assumptions and found them to lead to models that are unable to capture the transitions between different predictors. We thus disagree with the reviewer’s claim that our results may be obvious if one merely models the fact that there are fast transitions occurring between two predictors. Instead, we claim our *highly parsimonious functional form (Eq. 3) that quantitatively predicts and explains **why** transition phenomena occur* is a core contribution of our work. More broadly, we believe that the fact that our simple, 3-parameter model accurately predicts the behavior of ~900 checkpoints across 3 diverse settings is strong evidence against the tautology claim made by the reviewer. If the reviewer has a different model in mind, which they believe may fit the data well and hence offer an explanation, we are happy to run experiments with it and assess its validity!
>
> -----
>
> > **(W2) & (Q2): The weakness of the paper is that it does not address the main question of which complexity term?**
>
> Great question! In this work, we *intentionally* take a top-down, normative perspective to explain Transformer behavior. That is, we examine whether we can abstract away architectural details and explain Transformer behavior by simply asking what would be the optimal behavior under a particular set of solutions (M and G), and given minimal computational constraints. Thus, we emphasize our goal was not to find the *exact* complexity measure a Transformer uses, but to perform a top-down analysis to test if a **general, computational-level notion of complexity** could explain ICL. The striking result here is that even a **crude upper bound on Kolmogorov complexity**—approximated via standard compression algorithms—is sufficient to generate highly accurate predictions! That a rough, architecture-agnostic measure works so well is strong empirical evidence for our core claims. This robustness suggests that the underlying loss-complexity dynamic is fundamental and not overly sensitive to the specific proxy used.
>
> [1] arxiv.org/abs/1805.08522
>
> [2] www.nature.com/articles/s41467-024-54813-x
>
> [3] arxiv.org/abs/2308.12108
>
> -----
>
> > **(W3) & (Q3) The prediction of sub-linear growth is weak...it is possible that you have observed a sub-linear growth, and then decided on the modeling assumption A1**
>
> Thanks for raising this concern! We should have been more specific when giving the title for the prediction, and we apologize for the lack of clarity. As discussed in the caption for figure 6 and the beginning of page 8, our framework not only predicts a sub-linear growth of relative distance with $N$, but also that the relative distance between M and G follows a **sigmoidal trend with respect to $N^{1-\alpha}$**. We can assure the reviewer that our sigmoidal growth prediction came after the modeling assumptions—in fact, before making an assumption of power-law scaling, we could not have plotted the relative distance with respect to  $N^{1-\alpha}$! Consequently, our claim that the relative distance between M and G will follow a **sigmoidal trend with respect to $N^{1-\alpha}$** was truly a predictive statement, and we will change the title of the section to reflect this.
>
> ----
>
> > **(W3) An expectation from a theory is... to quantify the exponent $\alpha$**
>
> We emphasize $\alpha$ is a free parameter related to power law scaling of neural networks’ loss with dataset size. Predicting the precise exponent $\alpha$ *a priori* is an extremely difficult, open problem in the entire field of scaling laws, and we believe it is beyond the scope of our paper to solve it.
>
> -----
>
> >**(W4) & (Q4) The paper mentions that transient generalization phenomenon as an important observation, but it is well established that long training prevent generalization (hence, early stopping techniques)...What is different in this context...**
>
> We emphasize there is a crucial distinction between "transient generalization" and “overfitting”; the latter motivates early stopping and what we assume the reviewer has in mind whilst asking this question. While the reviewer is correct that standard overfitting occurs from training for a long time, i.e., for several epochs, on a *fixed* dataset, our experiments are conducted in an **online learning regime** where every training batch consists of **fresh, independently sampled sequences**. That is, the learner is not allowed to (or is at least extremely unlikely to) see the same datapoint twice. Therefore, "overfitting" in the traditional sense is not applicable in this setting. Instead, what we see is the model transitioning from a simpler, OOD-generalizing solution (G) to a more complex, ID-specialized memorizing solution (M) that achieves lower loss on the training *distribution*. This is a fundamentally different phenomenon, and our framework provides a quantitative explanation for why it occurs.
>
> ----
>
> > **(W5) & (Q5) The scaling of time to transience...depends on the complexity of the M and G models, as well as their losses, which, of course, unknown a priori...How could this serve as an estimate for this transience time in new tasks?**
>
> We believe there may be a slight misunderstanding here and take this opportunity to clarify. Specifically, given that we can write the predictors M and G in a closed-form, we emphasize that, unlike what the reviewer said, we can compute their loss and complexity a priori! In fact, our fitting of the Bayesian model strongly relies on this. More specifically, given sequences sampled from a data-generating distribution of a setting, we can compute the loss of each predictor on these sequences autoregressively, as well as compute their complexity by applying compression algorithms to the code and taking the lowest size in bits.
>
> This thus offers us all variables necessary to estimate the time to transience ($N^{*}$), yielding a non-trivial prediction that is pre-registered before the corresponding experiment is executed! As we show empirically in **Fig. 6(c)**, the predicted $N^*(D)$ from our theory aligns remarkably well with the observed crossover points from G to M, even as we compare with a training run with over *10 million steps*!
>
> -----
> -----
>
> **Summary:** We sincerely thank the reviewer for their detailed feedback and interesting questions! We hope our responses help address the reviewer’s concerns and, if so, they will consider increasing their score to support the acceptance of our work!

---

> ### Comment · Reviewer_JDTE · 2025-08-04
> **Reply to authors**
>
> Thank you very much for your detailed answers. Below are additional comments/questions.
>
> # (W1) & (Q1)
> ---
> 1. You say that "explaining transitions between different predictors learned by a model is the critical bottleneck towards understanding ICL." but I believe that this is known. You characterize the joint dependency on training steps and task diversity. What are the challenges and unexpected behavior that stems from this joint characterization? Moreover, how this agrees with the next part of your answer, in which you say that your observation on transformer interpolation "cannot be directly inferred from the observation of transitions."
> 2. Imposing the model’s hypothesis space to be either of two predictors is, in my opinion, is circular reasoning, as it ignores all other potential predictors.
> 3. What does it mean for "a predictive model to stem from the observation of distinct phases characterized by each predictor"? I don't follow.
> 4.  I believe that the claim is that in ICL the model learns the actual prior of the task distribution rather than interpolation between pretraining tasks.
> 5. The fact that a simple model explains a simple curve is desirable and anticipated. Appendix I is about further simplification of the model, whereas I am concerned about the details of the model. The claim that the transition from M to G is based on loss term and complexity term is well rooted in classic machine learning theory. The question is why exactly the loss term behaves as this power law, and why the complexity term is proportional to Kolmogorov complexity.
> ---
> # (W2) & (Q2)
> 6. Why is so surprising that a complexity term controls Memorization-Generalization trade-off? This is standard, and becomes interesting if the actual complexity term is identified (e.g., margin, various norms of DNN parameters).
> ---
> # (W3) & (Q3)
>
> My comment is not to doubt the integrity of the research process. However, the fact that it is even possible to easily do so, i.e., observe a sigmoidal behavior and fit a curve to it, weakens the predictive quality of the model.
> ---
>
> # (W5) & (Q5)
>
> Yes, but these are in a synthetic setting. Can the theory predict in a unseen problem that a predictor has transitioned from M to G?
> ---

---

> ### Author Response · Authors · 2025-08-06
> **Response to follow-up (1/3)**
>
> Thank you for the continued engagement with our responses! We also apologize for the delay in responding to your latest batch of comments due to some unforeseen circumstances.
>
> ---------
>
> > **You say that "explaining transitions between different predictors learned by a model is the critical bottleneck towards understanding ICL." but I believe that this is known.**
>
> We *respectfully* disagree with the reviewer that prior work has offered an explanation for why transitions between different predictors are observed in ICL. Instead, as we note in Sec. 1 (introduction), prior work has primarily focused on *eliciting* such transitions by varying experimental conditions (i.e., task diversity and training time) in *specific* settings. To our knowledge, an explanation, and a predictive model to validate it, for what drives these transitions across settings has been missing—our paper aims to fill this gap by (i) demonstrating transitions with respect to task diversity and training time across all well-studied settings can be unified in the language of Bayesian predictors and (ii) identifying an implicit loss–complexity tradeoff as the source of transitions between these predictors, and validating this explanation via a highly predictive model of transformer ICL behavior throughout training.
>
> That said, in the case the reviewer is aware of specific papers that offer a similar analysis to ours and that we failed to compare our work with, we are happy to both discuss and update our related work!
>
> ---------
>
> > **You characterize the joint dependency on training steps and task diversity. What are the challenges and unexpected behavior that stems from this joint characterization?**
>
> Great question! Consider prior work analyzing effects of training time (aka transience) and task-diversity. In both cases, prior work *unintentionally* controlled for one variable while characterizing the other. This often led to confusion for when transience or effects of task diversity are observed: e.g., following a very similar experimental protocol as studied by Singh et al. [1], who were the first to propose transience, Reddy [2] found transience does not occur in their setup. *The joint characterization helps us clarify this confusion:* one sees that transience occurs as a function of the given data diversity. Similarly, Raventos et al. [3], who were the first to propose the effects of data diversity, claimed that when data-diversity is increased and a model enters the ridge regression phase, it never goes back to dMMSE. We demonstrate this to be not true: transitions with respect to data diversity depend on the amount of training, and hence the authors’ claim was a consequence of inadequate training (see Appendix J for a precise statement and our analysis).
>
> On unexpected behavior: we emphasize the joint characterization elicits a precise phase diagram that has a universal structure across rather disparate settings---a sequence modeling task, regression on a continuous target, and classification with discrete labels. We argue this structure could not have been preemptively predicted unless the precise sigmoidal rate at which transience and task diversity effects modulate model behavior were known.
>
> [1] https://arxiv.org/abs/2311.08360
>
> [2] https://arxiv.org/abs/2312.03002
>
> ---------
> > **Moreover, how this agrees with the next part of your answer, in which you say that your observation on transformer interpolation "cannot be directly inferred from the observation of transitions."**
>
> We note that transitions are best described as a phenomenon observed in the limit, i.e., when a control variable is taken to $0$ or $\infty$. That is, one does not have any precise explanations for the system’s behavior when the control is in some intermediate range. Accordingly, merely observing the transitions does not suffice to say the model is *interpolating* between the two end states of the transition. To see this concretely, let us consider a generalized form of the reviewer’s quoted statement.
>
> Consider a control variable $n$ (e.g., time), such that a model $f(n)$’s behavior is observed to transition between predictors denoted $Q_1$ and $Q_2$ as $n$ varies from $0$ to $\infty$. That is, if $d(.,.)$ defines the distance between two functions (e.g., MSE over some dataset), then we have $d \left(f(n), Q_1 \right) \to 0$ when $n \to 0$, and $d \left(f(n), Q_2 \right) \to 0$ when $n \to \infty$. If the rate at which these distances change is nonlinear, there may be multiple explanations to the observed dynamics.
>
> **(response continued in next comment)**

---

> ### Author Response · Authors · 2025-08-06
> **Response to follow-up (2/3)**
>
> **(continuation of response)**
>
> For example, there may exist intermittent solutions learned by a model that have some constant distance with the two predictors $Q_1$ and $Q_2$ for any input $x$; this would yield a flat (or at least slowly changing) regions in a diagram measuring relative distance with respect $Q_1$ and $Q_2$ (we observe several such regions in our diagrams). Thus, to confirm whether the interpolation is in fact valid, one must compute a predictor $w(n) * Q_1 + (1 - w(n)) * Q_2$, where $w(n)$ is the weight associated with predictor $Q_1$ at control variable $n$, and compute its distance with the model: $d\left(f(n), w(n) * Q_1 + (1 - w(n)) * Q_2\right)$. If one somehow identifies the optimal value of weights $w(n)$ and finds the distance between model and interpolated predictor is always zero, despite the value of $n$, only then the statement that the model is an interpolation of the two predictors becomes justified. Meanwhile, if the distance is non-zero, there is likely some part of the model that the current predictors in our hypothesis space cannot be used to explain—that is, the hypothesis space is incomplete. When we assume a model is interpolating between the two predictors, and find this assumption to hold well when we analyze distance between model predictions and our hierarchical Bayesian model, we claim the assumption becomes justified.
>
> ---------
>
> > **Imposing the model’s hypothesis space to be either of two predictors is, in my opinion, is circular reasoning, as it ignores all other potential predictors.**
>
> While we agree our hierarchical Bayesian model **assumes** the hypothesis space is comprised of two predictors, we emphasize this **assumption is exhaustively validated** by demonstrating predictive power over both model next-token predictions and phase diagrams in order of 100s experimental settings (different training hyperparameters, context size, vocabulary size, and model size). In the scenario these predictions are imperfect, we agree with the reviewer there is a possibility of an alternative predictor that could be required to explain the model behavior, but given that this situation rarely arises in our experiments (especially in balls and urns and linear regression), we feel comfortable claiming the two predictors assumption is sufficient.
>
> On a related note, we take this opportunity to highlight our new results added in response to reviewer yreR’s comments on how our work would generalize to the case of multiple predictors. In our analysis, we generally start training time $N$ at some value sufficiently greater than $0$ (which we term the “Two-Hypotheses Threshold”). Close to initialization, as we mention in footnote 1, the distance between the interpolating predictor and Transformer turns out to be significantly greater than zero. This is arguably expected, for one is otherwise saying the model started with the ability to generalize. However, looking at the model outputs, one finds them to be equal to a constant value. For example, for the linear regression setting, this value equals the mean of the target variables. We find that when we accommodate this mean predictor into our hypothesis space, a new phase early in training is associated with this predictor in the relative distance maps. Moreover, the absolute distance between our model vs. the interpolated predictor (now defined over three predictors) goes to zero for that new phase!
>
> ---------
>
> > **What does it mean for "a predictive model to stem from the observation of distinct phases characterized by each predictor"? I don't follow.**
>
> Apologies for the confusion! To quote our precise phrase, we wrote “Our highly predictive model builds upon the observation of distinct phases characterized by each predictor, rather than directly stemming from it.” We thus meant to say that (i) we make the observation that there are continuous ranges of experimental settings (defined by data diversity and training time) wherein the model’s outputs are better (not perfectly) explained by *one of* the predictors (M or G), and (ii) we build upon this observation to define a hierarchical Bayesian model that has predictive power, i.e., it can predict next-token probabilities (without assuming access to model weights) and describe experimental conditions in which a specific predictor will better explain the model behavior. In step (ii), we are making the *assumption* that the model is interpolating between the two predictors, which is why we argue we are building upon our observation in step (i), instead of our model just stemming out of it (i.e., being a direct consequence of it).

---

> ### Author Response · Authors · 2025-08-06
> **Response to follow-up (3/3)**
>
> ---------
> > **The claim that the transition from M to G is based on loss term and complexity term is well rooted in classic machine learning theory. The question is why exactly the loss term behaves as this power law, and why the complexity term is proportional to Kolmogorov complexity.**
>
> We do *partially* agree with this. Specifically, the derivation of our hierarchical Bayesian model is akin to how generalization bounds are derived in machine learning (decompose validation loss into fit to training data and complexity of the implemented solution). However, in classical machine learning theory, the hypothesis space is defined over parameters of the model and hence the complexity measures are generally parametric in nature (e.g., given the final model, what is the norm or rank of weights). In contrast, our analysis is functional in nature: we assume specific functions the model will implement (M and G) and compute the complexity of these functions—not of the model! This is closer to the way derivations of generalization bounds would work in algorithmic learning theory (ALT), causing our analysis to deviate from classical machine learning theory. Even with ALT, we note the specific predictors we identified and the rate at which the model changes its behavior between them is not a prediction that would directly emerge from prior results in ALT—our findings are novel (and likely specific to the case of ICL).
>
> ---------
>
> > **Why is so surprising that a complexity term controls Memorization-Generalization trade-off? This is standard, and becomes interesting if the actual complexity term is identified (e.g., margin, various norms of DNN parameters).**
>
> Subjectively speaking, the surprise in our analysis lies in the predictions: the fact that without assuming access to models parameters (as would be done by the complexity terms suggested by the reviewer, e.g., norms, margin, or other measures like rank), we are able to almost perfectly predict a model’s outputs on both a per-datapoint and population level. To our knowledge, despite significant advances in classical machine learning theory to enable non-vacuous generalization bounds, such highly accurate predictions have not been produced (sans some very recent work [1]). We do however agree the specific form of the complexity term is critical (as shown by our experiments in the appendix), but for now our goal was the empirical demonstration of how *a crude approximation* of Kolmogorov complexity enables a predictive account of ICL, instead of deriving from scratch why this form is the right one.
>
> [1] https://proceedings.mlr.press/v162/lotfi22a.html
>
> ---------
>
> > **My comment is not to doubt the integrity of the research process. However, the fact that it is even possible to easily do so, i.e., observe a sigmoidal behavior and fit a curve to it, weakens the predictive quality of the model.**
>
> While we thank the reviewer for this comment, we emphasize that most scientific predictions tend to involve simple, smooth expressions, akin to our sigmoidal one. Moreover, it should be noted that without fitting our model under the scaling-law assumption and getting an $\alpha$ parameter, it would not be possible to observe this sigmoidal behavior with respect to $N^{1-\alpha}$, hence we could have only plotted such a curve after fitting our model. Moreover, in our opinion, the predictive quality of a model arises from how it relates some meaningful variables to each other, *prompting an experiment* to check whether the relationship empirically holds. Said another way, if one knew what experiment to run, i.e., what measure to plot against what control variable, we agree with the reviewer that one could likely guess what the relationship between them is and fit a curve to it. However, knowing what this experiment is, i.e., what variables one should plot and with respect to what, leads to the predictive nature of a claim. We believe our analysis of the sigmoidal trend meets that criterion.
>
> ---------
>
> > **Yes, but these are in a synthetic setting. Can the theory predict in a unseen problem that a predictor has transitioned from M to G?**
>
> Thank you for the question! We are unsure what the reviewer means by an “unseen problem”. In the case they mean an experimental setting beyond the ones studied in prior work, we note our Balls and Urns task satisfies that criterion: we introduced it in this paper and found high predictive power over whether the model has transitioned from M to G (or vice versa). In case the reviewer meant a more realistic, natural language setting, as we noted before, we are now working on an exciting follow-up paper that aims to demonstrate predictive power of our model in the natural task of many-shot jailbreaking with off-the-shelf LLMs. The experiments involved in that work have been quite involved and we hence believe its analysis is out-of-scope for the narrative of our current paper.

---

> > ### Comment · Reviewer_JDTE · 2025-08-07
> >
> > Dear authors,
> > I will try my best to go over it within the given rebuttal time, though your response is unreasonably long.
> > Reviewer JDTE

---

> > > ### Author Response · Authors · 2025-08-07
> > >
> > > Sincere apologies about that! We wanted to make sure we hit all points raised in your comments. Hopefully the responses feel like mere expansions on arguments from our rebuttals, since that was our intention, and hence easy to skim through!

---

### Official Review · Reviewer_yreR · 2025-07-02

**Clarity:** 3
**Significance:** 2
**Originality:** 3
**Rating:** 4
**Confidence:** 4

**Summary:**

This paper introduces a unified Bayesian framework to study the transition of transformers between memorization and generalization in In-Context Learning. The authors model the phenomena of ICL in Transformers as a Bayesian model comparison process.

**Questions:**

How would the framework fit to tasks with more than the two relevant predictors?

**Ethical Concerns:**

["NO or VERY MINOR ethics concerns only"]

**Final Justification:**

After a consideration according to the answers and other reviews, I would like to maintain my current score.

**Limitations:**

The dependence on the assumptions of simplicity bias and power law scaling may not hold in all cases.

**Quality:**

2

**Strengths And Weaknesses:**

Pros:
1. The paper show the tradeoff between memorizing predictors and generalizing predictors as a function of task diversity and training time.
2. The authors conduct experimental exploration under three different settings, including sequence modeling, linear regression and classification.
3. The paper uses an interpretable Bayesian framework to explain several In-Context Learning phenomena.

Cons:
1. It is advised to add a computational cost analysis of Bayesian model fitting process.
2. The generalizability of simplified assumptions with two predictors to complex scenarios remain uncertain.
3. The experiments are conducted in synthetic tasks with well defined task boundaries. The applicability of the framework to real cases with real-world settings is a concern.

---

> ### Author Rebuttal · Authors · 2025-07-31
>
> We thank the reviewer for their valuable feedback! We also thank the reviewer for supporting the acceptance of our work, and highlighting the strength of the paper in providing an “interpretable Bayesian framework to explain several In-Context Learning phenomena.” We wish to highlight that, in addition to explaining ICL phenomena, our account shows excellent quantitative fit and predictive power: quantitatively, across 72 different maps with varying MLP width, context length, and task dimensionality, we find our model is highly predictive of Transformer next-token predictions, with a mean $R^2$ of 0.97 in Linear Regression, a mean agreement of 0.92 in Classification, and a mean Spearman rank correlation of 0.97 in Balls & Urns. Additionally, we find very strong correlations of 0.99, 0.98, and 0.99 between our model’s posterior probabilities and the relative distance values given by the Transformer in the Linear Regression, Classification, and Balls & Urns settings, respectively.
>
> Below, we respond to specific comments by the reviewer.
>
> -----
>
>
> > **It is advised to add a computational cost analysis of Bayesian model fitting process.**
>
> While we do not address this directly in the paper, we note the fitting process of our Bayesian model is very fast! This is because for a given heatmap of predictions (e.g., Figure 4a), we only fit **3 free parameters** across *all* combinations of task diversity and training times. The process of fitting takes less than a minute to complete. This shows that our model is not only interpretable and predictive of Transformer behavior throughout training, but also quick to fit. We will make sure to add fitting details to the paper!
>
> ------
>
> > **The generalizability of simplified assumptions with two predictors to complex scenarios remains uncertain... How would the framework fit to tasks with more than the two relevant predictors?**
>
> Great question, and one that is crucial to the generalizability of our account! While we demonstrate our main results for two predictors, our framework can be easily extended to more predictors. Below, we first provide the **derivation** showing how our framework generalizes to more than two predictors. Then, we describe **new results** from an experiment in the linear regression setting, which provides a proof-of-concept that our framework generalizes to more than two predictors.
>
> #### **Mathematical derivation:**
> Our predictive equation for transformer behavior is:
> $$h^{pred}(s_i|s_{i-1:1},S_{\mathcal{T}})=p(M|S_{\mathcal{T}})M(s_i|s_{i-1:1})+p(G|S_{\mathcal{T}})G(s_i|s_{i-1:1})$$,
> which is simply a posterior-weighted average over the predictors (which in our case are $M$ and $G$).
> Therefore, in the case of $H$ predictors $Q_i \in \{Q_1, ..., Q_H\}$ we can simply write: $$h^{pred}(s_i|s_{i-1:1},S_{\mathcal{T}})=\sum_{i=1}^Hp(Q_i|S_{\mathcal{T}})Q_i(s_i|s_{i-1:1}).$$
>
> How should one compute the posterior in the case of $H$ predictors? In our current analysis, given our assumptions of prior and likelihood, we write the log posterior odds as:
> $$\eta(N,D)=\log p(M|S_{\mathcal{T}})-\log p(G|S_{\mathcal{T}})=\gamma N^{1-\alpha}\Delta L(D)-K(D)^{\beta}$$ and convert this to the posterior for the memorized solution via $p(Q|S_\mathcal{T})=\sigma (\eta(N,D))$.
> However, we can just as easily write this in a way that generalizes to $H$ predictors: given our assumptions of prior and likelihood, we can write: $$\log f(M|S_\mathcal{T})=-\gamma N^{1-\alpha}\bar{L_M}(S_\mathcal{T})-K(M)^{\beta}$$ with $\log f(M|S_{\mathcal{T}})$ being the unnormalized log probability. Then we get the posterior probability for $M$ by merely applying Softmax: $$p(M|S_\mathcal{T})=\frac{\exp(\log(f(M|S_{\mathcal{T}})))}{\exp(\log(f(M|S_{\mathcal{T}}))) + \exp(\log(f(G|S_{\mathcal{T}})))} = softmax(\log f(M|S_{\mathcal{T}}))$$
>
> Similarly, for a predictor $Q_i \in \{Q_1, ..., Q_H\}$, we have $\log f(Q_i|S_\mathcal{T})=-\gamma N^{1-\alpha}\bar{L_{Q_i}}(S_\mathcal{T})-K(Q_i)^{\beta}$, and hence:
> $$p(Q_i|S_\mathcal{T})=\frac{\exp(\log f(Q_i|S_{\mathcal{T}}))}{\sum_{j=1}^H \exp(\log f(Q_j|S_{\mathcal{T}}))} = softmax(\log f(Q_i|S_{\mathcal{T}})).$$
> Putting this into the equation above gives: $$h^{pred}(s_i| s_{i-1:1}, S_{\mathcal{T}})=\sum_{i=1}^H softmax(\log f(Q_i|S_{\mathcal{T}}))Q_i(s_i|s_{i-1:1}),$$ which is a general form of our model that works for $H$ predictors.
>
> #### **New results:**
> Beyond the derivation above, given the reviewer’s question, we note we have now run experiments to study the generalization of our results to more than 2 predictors. Specifically, we conduct an experiment in the linear regression setting, where, as we note in the discussion of our paper (Section 5), we find an early-appearing predictor that we call the *optimal-constant solution* ($C$). This solution always predicts the mean of the distribution (see [1], [2] for further description) and appears before either the generalizing ($G$) or memorizing ($M$) predictors, which qualitatively aligns with our framework’s perspective that simpler solutions appear earlier and are later replaced by better-fitting but more complex ones.
>
> While we are unable to provide heatmaps similar to Figure 4 due to this year’s NeurIPS rebuttals guidelines, we emphasize the results are very similar in nature. In this analysis, we start our model’s predictions from just 20 training steps, yielding insight very early into training. We also note that the model with 3 predictors captures transition dynamics, yielding a strong correlation with a version of the relative distance measure generalized to three predictors. Overall, this result provides a proof-of-concept that our framework can be easily generalized to capturing complex dynamics with more than two predictors, as well as capturing phenomena that emerge in settings with multiple predictors, such as the rapid emergence of in-context learning (seen early in the linear regression setting).
>
> Overall, we again thank the reviewer for this great question as it helped us further generalize our claims. We will make sure these results are added to the final version of the paper!
>
> [1] arxiv.org/abs/2406.17467
>
> [2] arxiv.org/abs/2402.02364
>
> ------
>
> > **The experiments are conducted in synthetic tasks with well defined task boundaries. The applicability of the framework to real cases with real-world settings is a concern.**
>
> Fair point! While we agree that our experiments use synthetic tasks, we emphasize that these tasks are *precisely* the ones that have been studied in the bulk of the literature to understand ICL [1-12]. We thus argue that any theory targeted at explaining ICL must at least show predictive power over this spectrum of simple, controlled, but phenomenologically very rich set of tasks that form the focus of prior work. Our hierarchical Bayesian framework meets this standard, offering both a predictive account of Transformer behavior throughout training and explanations for several past results on ICL in a unified manner.
> Having achieved this result, we now strongly think that the stage is set for analyzing more naturalistic settings. We have in fact already begun preliminary explorations on this front, analyzing the phenomenon of many-shot jailbreaking wherein safety fine-tuned models, when offered in-context sufficiently many exemplars of some undesirable behavior (e.g., producing violent speech), start to engage in said behavior. We believe our framework can offer a predictive account of this phenomenon with realistic models, but a detailed analysis warrants its own explicit study!
>
> [1] arxiv.org/abs/2205.05055
>
> [2] arxiv.org/abs/2312.03002
>
> [3] arxiv.org/abs/2412.00104
>
> [4] arxiv.org/abs/2311.08360
>
> [5] www.jmlr.org/papers/v25/23-1042.html
>
> [6] arxiv.org/abs/2310.08391
>
> [7] arxiv.org/abs/2503.05631
>
> [8] arxiv.org/abs/2402.11004
>
> [9] www.pnas.org/doi/abs/10.1073/pnas.2502599122
>
> [10] arxiv.org/abs/2211.15661
>
> [11] arxiv.org/abs/2306.15063
>
> [12] arxiv.org/abs/2412.01003
>
> [13] https://www.anthropic.com/research/many-shot-jailbreaking
>
> -----
> -----
>
> **Minor Comments**
>
> > **The dependence on the assumptions of simplicity bias and power law scaling may not hold in all cases.**
>
> While we agree with the reviewer that we rely on assumptions of simplicity bias and power law scaling, these assumptions have been robustly shown to hold across many studies of neural networks as well as practical applications. For power law scaling, there is extensive evidence that neural networks reduce loss at a rate corresponding to power laws as dataset size increases ([1-2]). This evidence has also strongly driven model development in industry, which has shown power laws to hold across many orders of magnitude. With regards to simplicity bias, works examining training dynamics ([3-5]), as well as directly sampling from initialized neural networks ([6,7]), have consistently revealed a simplicity bias in neural network training dynamics and their parameter-function map, respectively. Thus, we see our two main assumptions as robust and not substantially impacting the generalizability of our work.
>
>
> [1] arxiv.org/abs/2001.08361
>
> [2] arxiv.org/abs/2203.15556
>
> [3] arxiv.org/abs/1905.11604
>
> [4] arxiv.org/abs/2006.14599
>
> [5] arxiv.org/abs/2405.05847
>
> [6] arxiv.org/abs/2304.06670
>
> [7] arxiv.org/abs/1805.08522
>
>
> -----
> -----
>
> **Summary:** We sincerely thank the reviewer for their constructive comments and suggestions that have helped us demonstrate the generality of our framework, while also offering the opportunity to show its efficiency for explaining ICL. We hope our comments address the reviewer’s concerns, and, if so, they would consider championing the acceptance of our work!

---

> > ### Comment · Reviewer_yreR · 2025-08-08
> >
> > Thank you very much for your detailed response. After a consideration according to the answers and other reviews, I would like to maintain my current score.

---

### Note · Authors · 2025-08-12

We thank all reviewers for their detailed and constructive feedback. We are encouraged by the agreement that our work addresses a "timely and central topic" (JDTE) with a "unique approach" (Zg8o), providing a useful "step towards unification" (JDTE). Reviewers found our paper "clearly written and pedagogical" (PqQ7) as well as “well written and easy to follow” (JDTE), our mathematical formulation “laid out clearly” (Zg8o), and our figures “excellent at illustrating the hypotheses, methodology, and results” (PqQ7).

With regards to our method, results, and conclusion, reviewers specifically praised our "interpretable Bayesian framework" (yreR) and the "precision and rigour of the proposed model" (PqQ7), noting that it "fits the empirical data extremely well, which inspires great confidence” (PqQ7). Even reviewer JDTE, who had the most concerns regarding our work, noted that our “conclusion is based on extensive empirical evidence, and the graphs displayed are convincing.” We were particularly happy to read that Reviewer PqQ7 noted “I feel like I now have a better understanding of ICL” after reading the paper.

The review process has been invaluable. Per Reviewer yreR's suggestion, we demonstrated our framework's generality by extending it to more than two predictors, providing a mathematical derivation and new proof-of-concept experiments that address their main concern. We also incorporated valuable suggestions aligned with Reviewer PqQ7 to improve the paper's narrative. The reviewer was highly satisfied with our responses and noted that this is a “strong work that will be interesting to the field of ICL and deserves acceptance”. We are also pleased that our clarifications resolved Reviewer Zg8o's concerns, leading them to raise their score.

Our core contribution is a parsimonious, 3-parameter Bayesian model that is highly predictive of a Transformer ICL behavior throughout training and across diverse settings (e.g., mean $R^2= 0.97$ in linear regression). Our model also offers a unifying explanation for multiple ICL phenomena by positing a loss-complexity tradeoff impacting training dynamics. We believe the productive discussions have significantly strengthened the paper, and we hope the reviewers' positive assessments will lead you to consider our work for acceptance.

---

### Decision · Program_Chairs · 2025-09-17

**Decision:**

Accept (poster)

**Comment:**

This paper develops a hierarchical Bayesian framework that unifies two well-known ICL phenomena—task diversity effects and transient generalization—into a single predictive model based on a loss–complexity tradeoff. Unlike prior studies focusing on one factor, this work jointly models the roles of task diversity and training time, offering a coherent explanation and generating novel predictions. The empirical validation is thorough and shows strong alignment between the proposed model and Transformer behavior. While some predictions align with existing intuitions, the unified and predictive nature of the framework, along with extensive experiments, makes this a valuable contribution.